# Conditional Variational Diffusion Models

**Gabriel della Maggiora[1,2,∗], Luis Alberto Croquevielle[3,∗]**

**Nikita Deshpande[1,2], Harry Horsley[4], Thomas Heinis[3], Artur Yakimovich[1,2,5]**

[1] Center for Advanced Systems Understanding (CASUS), Görlitz, Germany
[2] Helmholtz-Zentrum Dresden-Rossendorf e. V. (HZDR), Dresden, Germany
[3] Department of Computing, Imperial College London, London, United Kingdom
[4] Bladder Infection and Immunity Group (BIIG), UCL Centre for Kidney and Bladder Health,
   Division of Medicine, University College London,
   Royal Free Hospital Campus, London, United Kingdom
[5] Institute of Computer Science, University of Wrocław, Wrocław, Poland

correspondence: g.della-maggiora-valdes@hzdr.de, aac622@ic.ac.uk, a.yakimovich@hzdr.de

## Abstract

Inverse problems aim to determine parameters from observations, a crucial task in engineering and science. Lately, generative models, especially diffusion models, have gained popularity in this area for their ability to produce realistic solutions and their good mathematical properties. Despite their success, an important drawback of diffusion models is their sensitivity to the choice of variance schedule, which controls the dynamics of the diffusion process. Fine-tuning this schedule for specific applications is crucial but time-consuming and does not guarantee an optimal result. We propose a novel approach for learning the schedule as part of the training process. Our method supports probabilistic conditioning on data, provides high-quality solutions, and is flexible, proving able to adapt to different applications with minimum overhead. This approach is tested in two unrelated inverse problems: super-resolution microscopy and quantitative phase imaging, yielding comparable or superior results to previous methods and fine-tuned diffusion models. We conclude that fine-tuning the schedule by experimentation should be avoided because it can be learned during training in a stable way that yields better results[1].

## 1 Introduction

Inverse problems deal with the task of finding parameters of interest from observations. Formally, for a mapping $A : \mathcal{Y} \to \mathcal{X}$ and $\mathbf{x} \in \mathcal{X}$ (the *data*), the inverse problem is to find an input $\mathbf{y} \in \mathcal{Y}$ such that $A(\mathbf{y}) = \mathbf{x}$. Examples abound in computer science. For instance, single-image super-resolution can be formulated in this setting. Inverse problems are usually *ill-posed*, which means that observations underdetermine the system or small errors in the data propagate greatly to the solution.

Deep networks trained with supervised learning have recently gained popularity for tackling inverse problems in contexts where input-data samples are available. Depending on the application, the optimization might target the $L_1$ or $L_2$ norm, or pixel-wise distance measures in image-based tasks. These methods are effective in many scenarios, but the supervised learning approach can introduce undesired artifacts (Lehtinen et al., 2018) and it does not yield uncertainty estimates for the solutions.

Since most inverse problems are ill-posed, it makes sense to model the uncertainty explicitly. This can be done by considering $\mathbf{y}$ and $\mathbf{x}$ as realizations of random variables $Y$ and $X$, respectively, and learning a conditional probability distribution $\mathbb{P}_{Y|X}$. Several methods are available to learn a distribution from pairs of samples (Isola et al., 2016; Peng & Li, 2020; Sohl-Dickstein et al.,

---

[∗]Equal contribution
[1]The code is available on https://github.com/casus/cvdm.

2015; Kohl et al., 2018). In this work, we use diffusion models, a likelihood-based method with state-of-the-art generation capabilities (Dhariwal & Nichol, 2021; Saharia et al., 2021).

Diffusion models produce high quality samples and offer stable training (Ho et al., 2020; Kingma et al., 2023). Despite these advantages, they have a few drawbacks, like the number of steps required to generate samples (Song et al., 2022) and their sensitivity to the choice of variance schedule (Saharia et al., 2021; Nichol & Dhariwal, 2021). The variance schedule controls the dynamics of the diffusion process, and it is usually necessary to fine-tune it with a hyperparameter search for each application. This is time-consuming and leads to suboptimal performance (Chen et al., 2020).

In this work, we introduce the Conditional Variational Diffusion Model (CVDM), a flexible method to learn the schedule that involves minimum fine-tuning. Our detailed contributions are:

- Following Kingma et al. (2023) we learn the schedule as part of the training, extending their approach to the conditioned case. Furthermore, we allow for learning a different schedule for each element in the output (e.g., a pixel-wise for images). These extensions require several technical novelties, including a separation-of-variables strategy for the schedule.

- We prove that the rate of convergence of the discrete-time diffusion loss to the continuous-time case depends strongly on the derivatives of the schedule. This shows that the finding in Kingma et al. (2023) that continuous-time diffusion loss is invariant under choice of schedule may not hold in practice. Based on this result, we introduce a novel regularization term that proves to be critical for the performance of the method in our general formulation.

- We implement the schedule by replacing the architecture in Kingma et al. (2023) with two networks, one required to be positive for the conditioning variable and one monotonic-convolutional network. This allows us to test our model with inputs of different resolutions without retraining. Moreover, thanks to our clean formulation, our method does not need the post-processing of the schedule that Kingma et al. (2023) introduces, nor the preprocessing of the input. This further contributes to streamlining our implementation.

We test CVDM in three distinct applications. For super-resolution microscopy, our method shows comparable reconstruction quality and enhanced image resolution compared to previous methods. For quantitative phase imaging, it significantly outperforms previous methods. For image super-resolution, reconstruction quality is also comparable to previous methods. CVDM proves to be versatile and accurate, and it can be applied to previously unseen contexts with minimum overhead.

## 2 RELATED WORK

Diffusion probabilistic models (DPMs) (Sohl-Dickstein et al., 2015) are a subset of variational autoencoders methods (Kingma & Welling, 2022; Kingma et al., 2023). Recently, these models have demonstrated impressive generation capabilities (Ho et al., 2020; Dhariwal & Nichol, 2021), performing better than generative adversarial networks (Isola et al., 2016) while maintaining greater training stability. Fundamentally, diffusion models operate by reversing a process wherein the structure of an input is progressively corrupted by noise, according to a variance schedule.

Building on the DPM setup, conditioned denoising diffusion probabilistic models (CDDPMs) have been introduced to tackle challenges like image super-resolution (Saharia et al., 2021), inpainting, decompression (Saharia et al., 2022), and image deblurring (Chen et al., 2023). All the cited works use different variance schedules to achieve their results, which is to be expected, because fine-tuning the schedule is usually a prerequisite to optimize performance (Saharia et al., 2021).

Efforts have been made to understand variance schedule optimization more broadly. Cosine-shaped schedule functions (Nichol & Dhariwal, 2021) are now the standard in many applications (Dhariwal & Nichol, 2021; Croitoru et al., 2023), showing better performance than linear schedules in various settings. Beyond this empirical observation, Kingma et al. (2023) developed a framework (Variational Diffusion Models, or VDMs) to learn the variance schedule and proved several theoretical properties. Their work supports the idea that learning the schedule during training leads to better performance.

Specifically, Kingma et al. (2023) formulate a Gaussian diffusion process for unconditioned distribution sampling. The latent variables are indexed in continuous time, and the forward process diffuses the input driven by a learnable schedule. The schedule must satisfy a few minimal conditions, and its

parameters are defined as a monotonic network. By forcing the schedule to be contained within a range, the model can be trained by minimizing a weighted version of the noise prediction loss. Their work also introduces Fourier features to improve the prediction of high-frequency details.

In this work, we extend VDMs to the conditioned case. To achieve this, we define the schedule as a function of two variables, time $t$ and condition $\mathbf{x}$. This requires careful consideration because the schedule should be monotonic in $t$ and not necessarily with respect to $\mathbf{x}$. To solve this issue, we propose a novel factorization of the schedule and derive several theoretical requirements for the schedule functions. By incorporating these requirements into the loss function, we can train the model in a straightforward way and dispense with the schedule post-processing of Kingma et al. (2023). We adapt the framework proposed by Saharia et al. (2021), which learns a noise prediction model that takes the variance schedule directly as input. To streamline the framework, we eliminate the noise embedding and directly concatenate the output of our learned schedule to the noise prediction model.

## 3 METHODS

### 3.1 CONDITIONED DIFFUSION MODELS AND INVERSE PROBLEMS

For a brief introduction to inverse problems, see Appendix A. For a formulation of non-conditioned DPMs, see Ho et al. (2020); Kingma et al. (2023). In conditioned DPMs we are interested in sampling from a distribution $\mathbb{P}_{Y|X=\mathbf{x}}(\mathbf{y})$, which we denote $p(\mathbf{y}|\mathbf{x})$. The samples from this distribution can be considered as solutions to the inverse problem.

Following Kingma et al. (2023), we use a continuous-time parametrization of the schedule and latent variables. Specifically, the schedule is represented by a function $\gamma(t, \mathbf{x})$ and for each $t \in [0, 1]$ there is a latent variable $\mathbf{z}_t$. We formulate the diffusion process using a time discretization, with continuous-time introduced as a limiting case. This formulation is better for the introduction of key concepts like the regularization strategy in Section 3.4. To start with, for a finite number of steps $T$, let $t_i = i/T$ for $i \in \{0, 1, \ldots, T\}$, and define the forward process by

$$q(\mathbf{z}_{t_i}|\mathbf{y}, \mathbf{x}) = \mathcal{N}\left(\sqrt{\gamma(t_i, \mathbf{x})}\mathbf{y}, \sigma(t_i, \mathbf{x})\mathbf{I}\right). \tag{1}$$

We use the variance-preserving condition $\sigma(t_i, \mathbf{x}) = \mathbf{1} - \gamma(t_i, \mathbf{x})$ but sometimes write $\sigma(t_i, \mathbf{x})$ to simplify notation. The whole process is conditioned on $\mathbf{y}$: it starts at $\mathbf{z}_{t_0} = \mathbf{y}$ and it gradually injects noise, so that each subsequent $\mathbf{z}_{t_i}$ is a noisier version of $\mathbf{y}$, and $\mathbf{z}_{t_T}$ is almost a pure Gaussian variable. The forward process should start at $\mathbf{y}$ in $t = 0$, so for all $\mathbf{x}$ we enforce the condition $\gamma(0, \mathbf{x}) = \mathbf{1}$.

We follow Kingma et al. (2023) in learning the schedule instead of fine-tuning it as a hyperparameter, with some key differences. If $\mathbf{y}$ has a vector representation, we learn a different schedule for each component. For example, in the case of images, each pixel has a schedule. This means that functions of $\gamma$ apply element-wise and expressions like $\sqrt{\gamma(t_i, \mathbf{x})}\mathbf{y}$ represent a Hadamard product.

Similarly to the non-conditioned case, the diffusion process is Markovian and each step $0 < i \leq T$ is characterized by a Gaussian distribution $q(\mathbf{z}_{t_i}|\mathbf{z}_{t_{i-1}}, \mathbf{x})$ (see Appendix B.1). The posterior $q(\mathbf{z}_{t_{i-1}}|\mathbf{z}_{t_i}, \mathbf{y}, \mathbf{x})$ also distributes normally, and the parameters ultimately depend on the function $\gamma$ (see Appendix B.2). The reverse process is chosen to take the natural shape

$$p_\nu(\mathbf{z}_{t_{i-1}}|\mathbf{z}_{t_i}, \mathbf{x}) = q(\mathbf{z}_{t_{i-1}}|\mathbf{z}_{t_i}, \mathbf{y} = \hat{\mathbf{y}}_\nu(\mathbf{z}_{t_i}, t_i, \mathbf{x}), \mathbf{x}), \tag{2}$$

such that it only differs from the posterior in that $\mathbf{y}$ is replaced by a deep learning prediction $\hat{\mathbf{y}}_\nu$. The forward process should end in an almost pure Gaussian variable, so $q(\mathbf{z}_{t_T}|\mathbf{y}, \mathbf{x}) \approx \mathcal{N}(\mathbf{0}, \mathbf{I})$ should hold. Hence, for the reverse process we model $\mathbf{z}_{t_T}$ with

$$p_\nu(\mathbf{z}_{t_T}|\mathbf{x}) = \mathcal{N}(\mathbf{0}, \mathbf{I}) \tag{3}$$

for all $\mathbf{x}$. The reverse process is defined by equations (2) and (3) without dependence on $\mathbf{y}$, so we can use them to sample $\mathbf{y}$. Specifically, we can sample from $p_\nu(\mathbf{z}_{t_T}|\mathbf{x})$ and then use relation (2) repeatedly until we reach $\mathbf{z}_{t_0} = \mathbf{y}$. All the relevant distributions are Gaussian, so this procedure is computationally feasible.

In other words, equations (2) and (3) completely define the reverse process, such that we can sample any $\mathbf{z}_{t_i}$ conditioned on $\mathbf{x}$, including $\mathbf{z}_{t_0} = \mathbf{y}$. If we are able to learn the $p_\nu(\mathbf{z}_{t_{i-1}}|\mathbf{z}_{t_i}, \mathbf{x})$, we should have a good proxy for $p(\mathbf{y}|\mathbf{x})$ from which we can sample.

## 3.2 Defining the Schedule

We now describe in more detail our approach to learning the schedule. Recall that the latent variables $\mathbf{z}_t$ are indexed by a continuous time parameter $t \in [0, 1]$, and we introduced the diffusion process using a time discretization $\{t_i\}_{i=0}^T$. We now consider the non-discretized case, starting with the forward process. In this case, the mechanics described by equation (1) can be extended straightforwardly to the continuous case $q(\mathbf{z}_t | \mathbf{y}, \mathbf{x})$.

The continuous-time version of the forward Markovian transitions and the posterior distribution are more complicated. Consider the forward transitions, which in the discretized version we denote by $q(\mathbf{z}_{t_i} | \mathbf{z}_{t_{i-1}}, \mathbf{x})$. To extend this idea to the continuous case, Kingma et al. (2023) consider $q(\mathbf{z}_t | \mathbf{z}_s, \mathbf{x})$ for $s < t$. We use a different approach and focus on the infinitesimal transitions $q(\mathbf{z}_{t+dt} | \mathbf{z}_t, \mathbf{x})$. This idea can be formalized using stochastic calculus, but we do not need that framework here. From Appendix B.1, the forward transitions are given by

$$q(\mathbf{z}_{t_i} | \mathbf{z}_{t_{i-1}}, \mathbf{x}) = \mathcal{N} \left( \sqrt{1 - \hat{\beta}_T(t_i, \mathbf{x})} \mathbf{y}, \hat{\beta}_T(t_i, \mathbf{x}) \mathbf{I} \right).$$

where $\hat{\beta}_T(t_i, \mathbf{x}) = 1 - \gamma(t_i, \mathbf{x})/\gamma(t_{i-1}, \mathbf{x})$. As defined, the values $\hat{\beta}_T(t_i, \mathbf{x})$ control the change in the latent variables over a short period of time. Now, consider the continuous-time limit $T \to \infty$. The intuition is that there is a function $\beta$ such that the change of the latent variables over an infinitesimal period of time $dt$ is given by $\beta(t, \mathbf{x})dt$. In the discretization, this becomes $\hat{\beta}_T(t_i, \mathbf{x}) = \beta(t_i, \mathbf{x})/T$. This idea leads to the following relation between $\gamma$ and $\beta$ (details in Appendix F):

$$\frac{\partial \gamma(t, \mathbf{x})}{\partial t} = -\beta(t, \mathbf{x})\gamma(t, \mathbf{x}). \tag{4}$$

In view of this relation, our approach is as follows. First, we make the assumption that $\beta$ can be decomposed into two independent functions, respectively depending on the time $t$ and the data $\mathbf{x}$. We write this as $\beta(t, \mathbf{x}) = \tau_\theta(t)\lambda_\phi(\mathbf{x})$, where both $\tau_\theta$ and $\lambda_\phi$ are learnable positive functions. This assumption takes inspiration from many separable phenomena in physics, and it proves to be general enough in our experiments. Moreover, $\gamma(t, \mathbf{x})$ should be decreasing in $t$, since the forward process should start at $\mathbf{y}$ in $t = 0$ and gradually inject noise from there. This monotony condition is much simpler to achieve in training if the $t$ and $\mathbf{x}$ variables are separated in the schedule functions. Replacing this form of $\beta$ in equation (4) and integrating with initial condition $\gamma(0, \mathbf{x}) = 1$, we get

$$\gamma(t, \mathbf{x}) = e^{-\lambda_\phi(\mathbf{x}) \int_0^t \tau_\theta(s)ds}. \tag{5}$$

Equation (5) could be used to compute $\gamma$ during training and inference, but we follow a different approach to avoid integration. Since $\tau_\theta$ is defined to be positive, its integral is an increasing function. This motivates the parametrization of $\gamma$ as $\gamma(t, \mathbf{x}) = e^{-\lambda_\phi(\mathbf{x})\rho_\chi(t)}$, where $\rho_\chi$ is a learnable function which is increasing by design. Summarizing, we parametrize $\beta$ and $\gamma$ such that they share the same function $\lambda_\phi(\mathbf{x})$ and separate time-dependent functions. *A priori*, $\rho_\chi$ and the integral of $\tau_\theta$ could not coincide. So, to ensure that equation (4) holds, we use the Deep Galerkin Method (Sirignano & Spiliopoulos, 2018) by including the following term as part of the loss function:

$$\mathcal{L}_\beta(\mathbf{x}) = \mathbb{E}_{t \sim U([0,1])} \left[ \left\| \frac{\partial \gamma(t, \mathbf{x})}{\partial t} + \beta(t, \mathbf{x})\gamma(t, \mathbf{x}) \right\|_2^2 \right] + \|\gamma(0, \mathbf{x}) - \mathbf{1}\|_2^2 + \|\gamma(1, \mathbf{x}) - \mathbf{0}\|_2^2,$$

where $U(A)$ represents the continuous uniform distribution over set $A$. The last two terms codify the soft constraints $\gamma(0, \mathbf{x}) = \mathbf{1}$ and $\gamma(1, \mathbf{x}) = \mathbf{0}$, which help to ensure that the forward process starts at $\mathbf{y}$ (or close) and ends in a standard Gaussian variable (or close). Noise injection is gradual because $\gamma$ is defined by a deep learning model as a smooth function (see details about architecture in Appendix G). In the rest of Section 3, we provide the remaining details about the loss function.

## 3.3 Non-regularized Learning

The schedule functions $\gamma, \beta$ define the forward process and determine the form of the posterior distribution and the reverse process (Appendix B). On the other hand, $\hat{\mathbf{y}}_\nu$ helps to define the reverse process $p_\nu$. To learn these functions we need access to a dataset of input-data pairs $(\mathbf{y}, \mathbf{x})$. The

standard approach for diffusion models is to use these samples to minimize the Evidence Lower Bound (ELBO), given in our case by (details in Appendix C)

$$\mathcal{L}_{\text{ELBO}}(\mathbf{y}, \mathbf{x}) = \mathcal{L}_{\text{prior}}(\mathbf{y}, \mathbf{x}) + \mathcal{L}_{\text{diffusion}}(\mathbf{y}, \mathbf{x}).$$

The term $\mathcal{L}_{\text{prior}} = D_{\text{KL}}\left(q(\mathbf{z}_{t_T}|\mathbf{z}_{t_0}, \mathbf{x}) \,||\, p_\nu(\mathbf{z}_{t_T}|\mathbf{x})\right)$ helps to ensure that the forward process ends in an almost pure Gaussian variable. As described in Appendix D, $\mathcal{L}_{\text{diffusion}}(\mathbf{y}, \mathbf{x})$ is analytic and differentiable, but is computationally inconvenient. To avoid this problem, this term can be rewritten as an expected value $\mathcal{L}_T(\mathbf{y}, \mathbf{x})$ which has a simple and efficient Monte Carlo estimator (Appendix E.1). For reasons outlined in the following section, we work in the continuous-time case, i.e. the $T \to \infty$ limit. Under certain assumptions, we get the following form of the ELBO when $T \to \infty$

$$\hat{\mathcal{L}}_{\text{ELBO}}(\mathbf{y}, \mathbf{x}) = D_{\text{KL}}\left(q(\mathbf{z}_1|\mathbf{y}, \mathbf{x}) \,||\, p_\nu(\mathbf{z}_1|\mathbf{x})\right) + \mathcal{L}_\infty(\mathbf{y}, \mathbf{x}).$$

In the above expression, $\mathbf{z}_1$ represents the latent variable at time $t = 1$, such that $p_\nu(\mathbf{z}_1|\mathbf{x})$ is a standard Gaussian and $q(\mathbf{z}_1|\mathbf{y}, \mathbf{x}) = \mathcal{N}(\sqrt{\gamma(1, \mathbf{x})}\mathbf{y}, \sigma(1, \mathbf{x})\mathbf{I})$. On the other hand, $\mathcal{L}_\infty$ is a continuous-time estimator for $\mathcal{L}_{\text{diffusion}}$ and takes an integral form. See Appendix E.2 for the full details.

In summary, $\hat{\mathcal{L}}_{\text{ELBO}}$ provides a differentiable and efficient-to-compute version of the ELBO, and we use it as the core loss function to learn the forward and reverse processes correctly. In the next section, we describe two important modifications we make to the loss function and the learning process.

## 3.4 REGULARIZED DIFFUSION MODEL

As mentioned above, we work with a diffusion process that is defined for continuous time. The schedule $\gamma$ and the model prediction $\hat{\mathbf{y}}_\nu$ depend on a continuous variable $t \in [0, 1]$, and the latent variables $\mathbf{z}_t$ are also parametrized by $t \in [0, 1]$. We also derived a continuous-time version of the diffusion loss, $\mathcal{L}_\infty$. In our final implementation, we get better results by replacing $\mathcal{L}_\infty$ with

$$\hat{\mathcal{L}}_\infty(\mathbf{y}, \mathbf{x}) = -\frac{1}{2}\mathbb{E}_{\epsilon \sim \mathcal{N}(\mathbf{0}, \mathbf{I}), t \sim U([0,1])}\left[\|\epsilon - \hat{\epsilon}_\nu(\mathbf{z}_t(\epsilon), t, \mathbf{x})\|_2^2\right],$$

where $\hat{\epsilon}_\nu$ is a noise prediction model. See details in Appendix E.3. This provides a natural Monte Carlo estimator for the diffusion loss, by taking samples $\epsilon \sim \mathcal{N}(\mathbf{0}, \mathbf{I})$ and $t \sim U([0, 1])$.

As shown in Kingma et al. (2023), increasing the number of timesteps $T$ should reduce the diffusion loss, so it makes sense to work in the $T \to \infty$ limit. Also, a continuous-time setup is easier to implement. Importantly, Kingma et al. (2023) prove that the continuous-time diffusion loss is invariant to the choice of variance schedule. However, we argue this is not necessarily the case in practice. Since all computational implementations are ultimately discrete, we look for conditions on $\gamma$ that make the discrete case as close as possible to the continuous one.

As explained in Appendix E.2, one way of achieving this is to keep the Euclidean norm of $\text{SNR}''(t, \mathbf{x})$ low, where $\text{SNR}(t, \mathbf{x}) = \gamma(t, \mathbf{x})/\sigma(t, \mathbf{x})$. We use $f'(t, \mathbf{x})$ to represent the partial derivative of a function $f(t, \mathbf{x})$ with respect to the time $t$. A natural way of incorporating this condition would be to include a regularization term in the loss function, with a form like

$$\mathcal{L}_{\text{SNR}}(\mathbf{x}) = \left\|\text{SNR}''(\cdot, \mathbf{x})\right\|_{L_2([0,1])} \quad \text{where by definition} \quad \text{SNR}(t, \mathbf{x}) = \frac{\gamma(t, \mathbf{x})}{\sigma(t, \mathbf{x})} = \frac{\gamma(t, \mathbf{x})}{1 - \gamma(t, \mathbf{x})}.$$

From this, we can see that $\mathcal{L}_{\text{SNR}}$ can be complicated to implement. It involves a fraction and a second derivative, operations that can be both numerically unstable. Moreover, as we have mentioned before, for $t = 0$ it should hold that $\gamma(t, \mathbf{x}) \approx 1$, which makes SNR more unstable around $t = 0$. Since $\text{SNR}(t, \mathbf{x}) \approx \gamma(t, \mathbf{x})$ for values of $t$ closer to 1, we replace $\mathcal{L}_{\text{SNR}}$ with the more stable

$$\mathcal{L}_\gamma(\mathbf{x}) = \mathbb{E}_{t \sim U([0,1])}\left[\|\gamma''(t, \mathbf{x})\|_2^2\right].$$

This regularization term is actually key for the performance of our method. To see this, recall that a variable $\mathbf{z}_t \sim q(\mathbf{z}_t|\mathbf{y}, \mathbf{x})$ can be reparametrized as $\mathbf{z}_t = \sqrt{\gamma(t, \mathbf{x})}\mathbf{y} + \sqrt{\sigma(t, \mathbf{x})}\epsilon$ with $\epsilon \sim \mathcal{N}(\mathbf{0}, \mathbf{I})$. This means that $\gamma \equiv \mathbf{0}, \sigma \equiv \mathbf{1}$ make $\mathbf{z}_t = \epsilon$, so that the noise prediction model $\hat{\epsilon}_\nu(\mathbf{z}_t(\epsilon), t, \mathbf{x}) = \mathbf{z}_t$ can perfectly predict $\epsilon$ and make $\hat{\mathcal{L}}_\infty = 0$. Now, $\gamma \equiv \mathbf{0}$ is not compatible with the $\mathcal{L}_\beta$ loss term, but any function $\gamma$ that starts at $\mathbf{1}$ for $t = 0$ and then abruptly drops to $\mathbf{0}$ is permitted. $\mathcal{L}_\gamma$ prevent this type of undesirable solution. Once we include this term, the full loss function takes the form

$$\mathcal{L}_{\text{CVDM}} = \mathbb{E}_{(\mathbf{y}, \mathbf{x}) \sim p(\mathbf{y}, \mathbf{x})}\left[\mathcal{L}_\beta(\mathbf{x}) + D_{\text{KL}}\left(q(\mathbf{z}_1|\mathbf{y}, \mathbf{x}) \,||\, p_\nu(\mathbf{z}_1|\mathbf{x})\right) + \hat{\mathcal{L}}_\infty(\mathbf{y}, \mathbf{x}) + \alpha\mathcal{L}_\gamma(\mathbf{x})\right], \quad (6)$$

where $\alpha$ controls the weight of the regularization term and $p(\mathbf{y}, \mathbf{x})$ is the joint distribution of $\mathbf{y}$ and $\mathbf{x}$. We optimize a Monte Carlo estimator of $\mathcal{L}_{\text{CVDM}}$ by using the available $(\mathbf{y}, \mathbf{x})$ samples. For the KL divergence term, we optimize the analytical form of the KL divergence between two Gaussian distributions with the log-variance (Kingma & Welling, 2022; Nichol & Dhariwal, 2021).

## 4 EXPERIMENTS AND RESULTS

We assess the performance of our model on three distinct benchmarks. First, we evaluate the model's ability to recover high-spatial frequencies using the BioSR super-resolution microscopy benchmark (Qiao et al., 2021). Second, we examine the model's effectiveness in retrieving the phase of a diffractive system with synthetic data and real brightfield image stacks from a clinical sample assessed by two experienced microscopists. The last benchmark is image super-resolution on ImageNet 1K (Deng et al., 2009) . For the first two, performance is measured using two key metrics: multi-scale structural similarity index measure (MS-SSIM) (Wang et al., 2003) and Mean Absolute Error (MAE), both detailed in Appendix G. In the case of ImageNet, performance is measured using SSIM and peak signal-to-noise ratio (PSNR). For BioSR, the resolution of the reconstruction (a metric related to the highest frequency in the Fourier space) is additionally evaluated as per Qiao et al. (2021), using a parameter-free estimation (Descloux et al., 2019). We evaluate against methods developed for each benchmark, as well as CDDPM. In the case of CDDPM, we follow the implementation shown in Saharia et al. (2021). The specific implementation of our model is described in Appendix G.

### 4.1 SUPER-RESOLUTION MICROSCOPY

Super-resolution microscopy aims to overcome the diffraction limit, which restricts the observation of fine details in images. It involves reconstructing a high-resolution image $\mathbf{y}$ from its diffraction-limited version $\mathbf{x}$, expressed mathematically as $\mathbf{x} = K * \mathbf{y} + \eta$, where $K$ is the point spread function (PSF) and $\eta$ represents inherent noise. Convolution of the PSF with the high-resolution image $\mathbf{y}$ leads to the diffraction-limited image $\mathbf{x}$, which complicates $\mathbf{y}$ recovery due to information loss and noise. In this context, we utilize the BioSR dataset (Qiao et al., 2021).

BioSR consists of pairs of widefield and structured illumination microscopy (SIM) images which encapsulate varied biological structures and signal-to-noise ratio (SNR) levels. The structures present in the dataset have varying complexities: clathrin-coated pits (CCPs), endoplasmic reticulum (ER), microtubules (MTs), and F-actin filaments, ordered by increasing structural complexity. Each image pair is captured over ten different SNR levels. Our results are compared with DFCAN, a regression-based method implemented as in Qiao et al. (2021), and CDDPM trained as in Saharia et al. (2021). For diffusion methods during inference, best results are found at $T = 500$, resulting in similar inference time for CVDM and CDDPM. For CDDPM, fine-tuning results in optimal performance for a linear schedule ranging from 0.0001 to 0.03. All models are trained for 400,000 iterations.

#### 4.1.1 RESULTS

Table 1 shows the enhanced resolution achieved in the BioSR benchmark, with our model surpassing other methods in the ER and F-actin structures. Our approach consistently delivers comparable or superior performance than CDDPM across all structures, and improves markedly over DFCAN in the resolution metric. Figure 6b facilitates a visual inspection of these achievements, comparing our reconstructions to those of DFCAN, and Figure 6a, which underscores the contrast in quality between our model and the CDDPM benchmark. Additional comparative insights are detailed in Appendix I.

### 4.2 QUANTITATIVE PHASE IMAGING

Quantitative phase imaging (QPI) has gained prominence in diverse applications, including bio-imaging, drug screening, object localization, and security scanning (Zuo et al., 2020). The Transport of Intensity Equation (TIE) method (Teague, 1983; Streibl, 1984) is a notable approach to phase retrieval, linking the diffraction intensity differential along the propagation direction to the lateral phase profile. This relationship, under the paraxial approximation, is formulated as

$$-k\frac{\partial I(x,y;z)}{\partial z} = \nabla_{(x,y)} \cdot (I(x,y;z)\nabla_{(x,y)}\varphi(x,y;z)),$$

Table 1: Performance on BioSR Structures. Bold represents a statistically significant best result. Underline represents a statistically significant second place. Statistical significance is determined by comparing the sample mean errors of two different methods over the dataset, using a hypothesis test.

(a) CCP structures.

| Metric / Model | DFCAN | CDDPM | CVDM (ours) |
| --- | --- | --- | --- |
| MS-SSIM ($\uparrow$) | 0.957 | 0.952 | 0.955 |
| MAE ($\downarrow$) | 0.006 | 0.007 | 0.007 |
| Resolution (nm) ($\downarrow$) | 107 | 100 | **96** |

(b) ER structures.

| Metric / Model | DFCAN | CDDPM | CVDM (ours) |
| --- | --- | --- | --- |
| MS-SSIM ($\uparrow$) | 0.928 | 0.920 | **0.934** |
| MAE ($\downarrow$) | 0.033 | 0.033 | 0.032 |
| Resolution (nm) ($\downarrow$) | 165 | 157 | **152** |

(c) MT structures.

| Metric / Model | DFCAN | CDDPM | CVDM (ours) |
| --- | --- | --- | --- |
| MS-SSIM ($\uparrow$) | **0.901** | 0.857 | 0.887 |
| MAE ($\downarrow$) | **0.033** | 0.042 | 0.04 |
| Resolution (nm) ($\downarrow$) | 127 | 101 | **97** |

(d) F-actin structures.

| Metric / Model | DFCAN | CDDPM | CVDM (ours) |
| --- | --- | --- | --- |
| MS-SSIM ($\uparrow$) | 0.853 | 0.831 | **0.863** |
| MAE ($\downarrow$) | 0.049 | 0.049 | 0.043 |
| Resolution (nm) ($\downarrow$) | 151 | 104 | **98** |

Table 2: Performance on QPI Data. Bold represents a statistically significant best result.

(a) Performance on QPI on synthetic HCOCO.

| Metric / Model | US-TIE | CDDPM | CVDM (ours) |
| --- | --- | --- | --- |
| MS-SSIM ($\uparrow$) | 0.907 | 0.881 | **0.943** |
| MAE ($\downarrow$) | 0.110 | 0.134 | **0.073** |

(b) MS-SSIM on QPI Brightfield images.

| Sample | US-TIE | CDDPM | CVDM (ours) |
| --- | --- | --- | --- |
| 1 | 0.740 | 0.863 | **0.892** |
| 2 | 0.625 | 0.653 | **0.742** |
| 3 | 0.783 | 0.823 | **0.851** |

where $k$ is the wavenumber, $I$ is the intensity, $\varphi$ is the phase of the image, and $x, y, z$ are the coordinates. In practice, the intensity derivative $\partial I(x, y; z)/\partial z$ at $z = 0$ is approximated via finite difference calculations, using 2-shot measurements at $z = d$ and $z = -d$:

$$\frac{\partial I(x, y; z)}{\partial z}\bigg|_{z=0} \approx \frac{I_{-d} - I_d}{2d}.$$

We train the model with a synthetic dataset created by using ImageNet to simulate $I_d$ and $I_{-d}$ from grayscale images. The process, informed by the Fresnel diffraction approximation, is expressed as

$$I_d = \left| \sqrt{I_0} e^{i\varphi} * \frac{e^{ikz}}{i\lambda z} e^{i\frac{k}{2z}(x^2 + y^2)} \right|^2,$$

with $z = d$, the defocus distance, fixed at $2\mu$m during the training phase. Noise was added using a $\mathcal{N}(\xi, \xi)$ distribution, where $\xi$ fluctuates randomly between 0 and 0.2 for every image pair to approximate Poisson noise.

To evaluate the effectiveness of our method, we compare against two baselines: CDDPM and US-TIE (Zhang et al., 2020), an iterative parameter-free estimation method for the QPI problem. For diffusion methods during inference, best results are found at $T = 400$, resulting in similar inference time for CVDM and CDDPM. For CDDPM, fine-tuning results in optimal performance for a linear schedule ranging from 0.0001 to 0.02. Both our model and CDDPM undergo 200,000 iterations of training.

### 4.2.1 RESULTS FOR THE SYNTHETIC QPI DATASET

We estimate the phase of images using the HCOCO dataset (Cong et al., 2019). These images are simulated at a distance of $d = \pm 2\mu$m. Table 2a presents the model's performance on the provided test split of HCOCO, which is conducted without noise. Figure 1a visually compares the methods. For more detailed comparisons, please refer to Appendix I.

### 4.2.2 RESULTS FOR QPI GENERATED FROM CLINICAL BRIGHTFIELD IMAGES

We evaluate our method on microscope brightfield images, consisting of three stacks with varying defocus distances, obtained from clinical urine microscopy samples. The phase ground truth is established by computing $\partial I(x, y; z)/\partial z$, using a 20th-order polynomial fitting for each pixel within the stack, following Waller et al. (2010). This fitting is performed at distances $d = \pm 2k\mu$m, with $k$

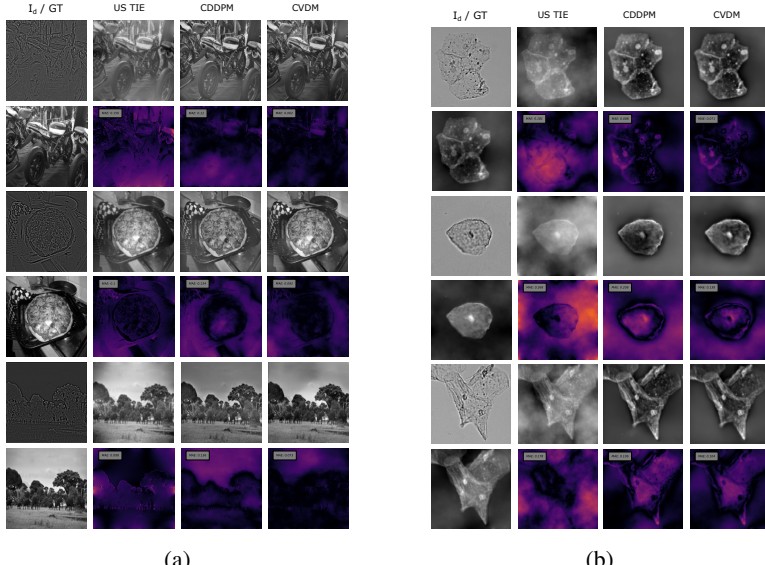

|  | $I_d$ / GT | US TIE | CDDPM | CVDM |  | $I_d$ / GT | US TIE | CDDPM | CVDM |

(a)                  (b)

Figure 1: QPI methods evaluated in the synthetic dataset (a) and brightfield clinical microscopy showing epithelial cells (b). From left to right: first column displays the defocused image at distance $d$ ($I_d$), with the respective ground truth (GT) situated directly below. Second, third, and fourth columns represent each a different method, with the reconstruction on top and the error image at the bottom.

ranging from 1 to 20, and the gradient of the polynomial is employed to apply the US-TIE method to generate ground truth data. All methods are evaluated using a 2-shot approach at $d = \pm 2\mu$m. The diffusion models are evaluated in a zero-shot scenario from the synthetic experiment. Figure 1b illustrates the reconstructions of all stacks, while Table 2b presents the MS-SSIM for each sample and method, with sample numbers corresponding to the figure rows.

### 4.3 ABLATIONS AND ADDITIONAL EXPERIMENTS

In addition to super-resolution microscopy and QPI, we also test our method for the problem of image super-resolution over ImageNet. For this task, we use the architecture of Saharia et al. (2021), equipped with our schedule-learning framework. Without any fine-tuning of the schedule, our results are comparable to Saharia et al. (2021) and slightly better than Kawar et al. (2022), another diffusion-based method. See Appendix L for more details and reconstruction examples.

We also evaluate the impact of our design decisions. First, the regularization term (Section 3.4) is critical in preventing the schedule from converging to a meaningless solution. See Appendix K for a detailed analysis. Second, the separation of variables for $\beta$ (Section 3.2) ensures that the monotonic behavior of $\gamma(t, \mathbf{x})$ with respect to $t$ is not extended to $\mathbf{x}$. This is key for achieving competitive performance, so we do not include a detailed ablation study. Finally, we compare learning a pixel-wise schedule to a single, global schedule using the synthetic QPI dataset. Our experiment shows that performance drops with a single learned schedule (see details in Appendix K).

### 5 ANALYSIS AND DISCUSSION

**Accuracy.** Our model competes favorably with DFCAN on the BioSR dataset, particularly excelling in image resolution (Table 1). It shows the versatility of diffusion models in generating realistic solutions across diverse phenomena. Additionally, it outperforms CDDPM in more complex biological structures, and it advances significantly in the QPI problem by overcoming noise challenges near the singularity (Wu et al., 2022). While not designed specifically for these tasks, our approach shows promise as a flexible tool for solving inverse problems with available input-data pair samples.

**Schedule.** As explained in Section 1, the schedule is a key parameter for diffusion models, so we aim to understand whether our method yields reasonable values. Based on the relation between $\gamma$ and $\beta$,

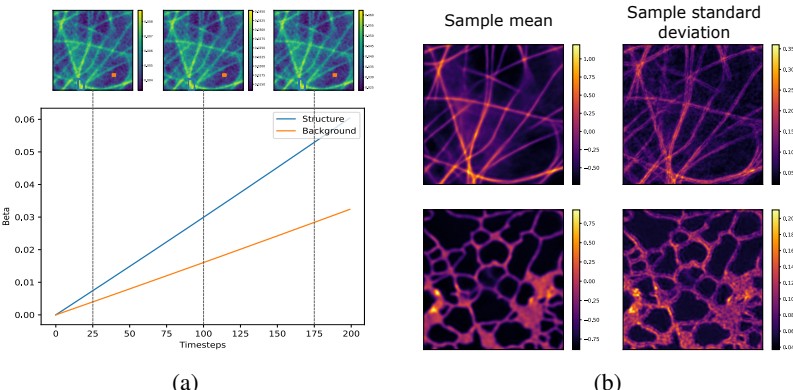

(a)             (b)

Figure 2: Schedules and sample mean and deviations for the represented images from the BioSR dataset. (a) Schedule ($\beta$) values for a microtubule image. The graph shows the average of the pixels in the respective region. (b) Mean and standard deviations for microtubule (top row) and endoplasmic reticulum (bottom row). The images were reconstructed using 20 samples obtained with CVDM.

the forward model can be parametrized as (details in Appendix J):

$$q(\mathbf{z}_t | \mathbf{y}, \mathbf{x}) = \mathcal{N}\left( e^{-\frac{1}{2}\int_0^t \beta(s, \mathbf{x})ds} \mathbf{y}, \left( \mathbf{1} - e^{-\int_0^t \beta(s, \mathbf{x})ds} \right) \mathbf{I} \right).$$

We can see that for large values of $\beta$, the latent variable $\mathbf{z}_t$ gets rapidly close to a $\mathcal{N}(\mathbf{0}, \mathbf{I})$ distribution. For small values of $\beta$, on the other hand, $\mathbf{z}_t$ remains closer to $\mathbf{y}$ for longer. In general, a steeper graph of $\beta$ results in a diffusion process that adds noise more abruptly around the middle timesteps, resulting in more difficult inversion of the process. In our image-based data applications, the pixel-wise dedicated schedule supports this analytical insight. In BioSR (Figure 2a), structure pixels (i.e., pixels with high-frequency information) are more difficult to denoise, which is reflected in the steeper $\beta$ graph. In contrast, background pixels (i.e., pixels with low-frequency information) are easier to resolve. This is consistent with the diffraction limit in optical microscopy, which mostly consists of low frequencies. Conversely, in QPI background pixels have a steeper $\beta$ graph, a phenomenon linked to the amplification of low-frequency noise around the singularity point in k-space (Wu et al., 2022).

**Uncertainty.** In our experiments, we measure the uncertainty of the reconstruction by the pixel-wise variance on the model predictions. This uncertainty is theoretically tied to $\beta$. As described in Song et al. (2021), the reverse diffusion process can be characterized by a stochastic differential equation with a diffusion coefficient of $\sqrt{\beta(t, x)}$ (in the variance-preserving case). Hence, higher values of $\beta$ introduce more diffusion into the reverse process over time, leading to more varied reconstructions. For BioSR, structure pixels exhibit higher values of $\beta$ and consequently higher reconstruction variance, as seen in Figure 2b. For QPI, the converse phenomenon is true (Figure 12). For further analysis and illustration of uncertainty, see Appendix M.

## 6 CONCLUSION

We introduce a method that extends Variational Diffusion Models (VDMs) to the conditioned case, providing new theoretical and experimental insights for VDMs. While theoretically, the choice of schedule should be irrelevant for a continuous-time VDM, we show that it is important when a discretization is used for inference. We test our method in three applications and get comparable or better performance than previous methods. For image super-resolution and super-resolution microscopy we obtain comparable results in reconstruction metrics. Additionally, for super-resolution microscopy we improve resolution by $4.42\%$ when compared against CDDPM and $26.27\%$ against DFCAN. For quantitative phase imaging, we outperform 2-shot US-TIE by $50.6\%$ in MAE and $3.90\%$ in MS-SSIM and improve the CDDPM benchmark by $83.6\%$ in MAE and $7.03\%$ in MS-SSIM. In the wild brightfield dataset, we consistently outperform both methods. In general, our approach improves over fine-tuned diffusion models, showing that learning the schedule yields better results than setting it as a hyperparameter. Remarkably, our approach produces convincing results for a wild clinical microscopy sample, suggesting its immediate applicability to medical microscopy.

## ACKNOWLEDGEMENTS

This work was partially funded by the Center for Advanced Systems Understanding (CASUS) which is financed by Germany's Federal Ministry of Education and Research (BMBF) and by the Saxon Ministry for Science, Culture, and Tourism (SMWK) with tax funds on the basis of the budget approved by the Saxon State Parliament.

The authors acknowledge the financial support by the Federal Ministry of Education and Research of Germany and by Sächsische Staatsministerium für Wissenschaft, Kultur und Tourismus in the programme Center of Excellence for AI-research "Center for Scalable Data Analytics and Artificial Intelligence Dresden/Leipzig", project identification number: ScaDS.AI.

The authors also acknowledge the support of the National Agency for Research and Development (ANID) through the Scholarship Program (DOCTORADO BECAS CHILE/2023 - 72230222).

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

## A  INVERSE PROBLEMS

We briefly describe a formalization of inverse problems, and mention some of the relevant ideas in this area. Consider two spaces $\mathcal{X}, \mathcal{Y}$ and a mapping $A : \mathcal{Y} \to \mathcal{X}$. For any particular $\mathbf{x} \in \mathcal{X}$ (usually called the *data*), we are interested in finding an input $\mathbf{y} \in \mathcal{Y}$ such that $A(\mathbf{y}) = \mathbf{x}$. In most interesting cases $A$ is non-invertible, and even when it is, it may be *ill-conditioned*, in the sense that the inverse operation will amplify the noise or measurements errors in the data (Fernández-Martínez et al., 2013).

Consider the case of solving $A\mathbf{y} = \mathbf{x}$, for an invertible matrix $A$. If the condition number of the matrix is large, even this relatively simple problem may be dominated by numerical errors in practice (Pyzara et al., 2011). The situation is much more numerically complex when the problem involves a mapping between infinite-dimensional spaces and there is a need for discretization. As a consequence, rather than inverting the operator $A$, inverse problems are more commonly solved with a *least-squares* approach (Lustig et al., 2007; Mousavi et al., 2020):

$$\min_{\mathbf{y} \in \mathcal{Y}} \|A(\mathbf{y}) - \mathbf{x}\|.$$

The solution of the above optimization problem defines a generalized inverse for $A$ (Korolev & Latz, 2021). However, if the problem is *ill-posed*, this approach will still be highly sensitive to the noise in the data. This has motivated the use of *regularization* (Calvetti & Somersalo, 2018; Benning & Burger, 2018), which changes the problem into

$$\min_{\mathbf{y} \in \mathcal{Y}} \|A(\mathbf{y}) - \mathbf{x}\|_2^2 + \alpha \mathcal{J}(\mathbf{y}),$$

where the regularization term usually takes the form $\mathcal{J}(\mathbf{y}) = \|\mathbf{y}\|$, and $\alpha$ determines how much weight is assigned to the second term (*prior knowledge of* $\mathbf{y}$) as compared to the first (*data fitting*). Depending on several conditions, regularization can guarantee a stable optimization problem (Korolev & Latz, 2021; Calvetti & Somersalo, 2018). Still, further methods for solving inverse problems have been explored. One reason for this is that the theoretical understanding of regularization is not complete for nonlinear mappings (Benning & Burger, 2018).

Moreover, as we mentioned already, one of the challenges for inverse problems is the presence of noise in the data. This is usually modeled as $\mathbf{x} = A(\mathbf{y}) + \eta$, where $\eta$ is the realization of a random variable. If the noise plays an important role, it would be desirable to provide uncertainty estimation for the solution. This has motivated the use of stochastic methods for inverse problems, one of the most prominent being the use of the Bayesian approach (Benning & Burger, 2018; Nickl, 2023).

## B  FORWARD PROCESS AND POSTERIOR DISTRIBUTION

### B.1  FORWARD PROCESS MARKOVIAN TRANSITIONS

Using the result in Appendix A of Kingma et al. (2023), for each step $0 < i \leq T$ we have

$$q(\mathbf{z}_{t_i} | \mathbf{z}_{t_{i-1}}, \mathbf{x}) = \mathcal{N}\left(\sqrt{\frac{\gamma(t_i, \mathbf{x})}{\gamma(t_{i-1}, \mathbf{x})}} \mathbf{z}_{t_{i-1}}, \left(\sigma(t_i, \mathbf{x}) - \frac{\gamma(t_i, \mathbf{x})}{\gamma(t_{i-1}, \mathbf{x})} \sigma(t_{i-1}, \mathbf{x})\right) \mathbf{I}\right).$$

We are working with the variance-preserving case, that is, $\sigma(t, \mathbf{x}) = \mathbf{1} - \gamma(t, \mathbf{x})$. Looking at the variance in the expression above, notice that

$$\sigma(t_i, \mathbf{x}) - \frac{\gamma(t_i, \mathbf{x})}{\gamma(t_{i-1}, \mathbf{x})} \sigma(t_{i-1}, \mathbf{x}) = (\mathbf{1} - \gamma(t_i, \mathbf{x})) - \frac{\gamma(t_i, \mathbf{x})}{\gamma(t_{i-1}, \mathbf{x})} (\mathbf{1} - \gamma(t_{i-1}, \mathbf{x}))$$

$$= \mathbf{1} - \gamma(t_i, \mathbf{x}) - \frac{\gamma(t_i, \mathbf{x})}{\gamma(t_{i-1}, \mathbf{x})} + \gamma(t_i, \mathbf{x})$$

$$= \mathbf{1} - \frac{\gamma(t_i, \mathbf{x})}{\gamma(t_{i-1}, \mathbf{x})}.$$

To simplify these expressions, we introduce the following notation:

$$\hat{\beta}_T(t_i, \mathbf{x}) := \mathbf{1} - \frac{\gamma(t_i, \mathbf{x})}{\gamma(t_{i-1}, \mathbf{x})}.$$

Notice that $\hat{\beta}_T$ depends on both $t_{i-1}$ and $t_i$, but we have only parametrized it in terms of $t_i$. We can do this because, for any fixed number of steps $T$, we can easily recover $t_{i-1}$ from $t_i$. Our notation will be convenient when we consider the limit case $T \to \infty$, as explained in Section 3.2. With this notation, we can restate the result in a more concise way:

$$q(\mathbf{z}_{t_i}|\mathbf{z}_{t_{i-1}}, \mathbf{x}) = \mathcal{N}\left(\sqrt{1 - \hat{\beta}_T(t_i, \mathbf{x})}\mathbf{z}_{t_{i-1}}, \hat{\beta}_T(t_i, \mathbf{x})\mathbf{I}\right).$$

### B.2 POSTERIOR DISTRIBUTION

By Bayes' theorem, we know that

$$q(\mathbf{z}_{t_{i-1}}|\mathbf{z}_{t_i}, \mathbf{y}, \mathbf{x}) = \frac{q(\mathbf{z}_{t_i}|\mathbf{z}_{t_{i-1}}, \mathbf{y}, \mathbf{x})q(\mathbf{z}_{t_{i-1}}|\mathbf{y}, \mathbf{x})}{q(\mathbf{z}_{t_i}|\mathbf{y}, \mathbf{x})} = \frac{q(\mathbf{z}_{t_i}|\mathbf{z}_{t_{i-1}}, \mathbf{x})q(\mathbf{z}_{t_{i-1}}|\mathbf{y}, \mathbf{x})}{q(\mathbf{z}_{t_i}|\mathbf{y}, \mathbf{x})}.$$

Notice that both the prior $q(\mathbf{z}_{t_{i-1}}|\mathbf{y}, \mathbf{x})$ and the likelihood $q(\mathbf{z}_{t_i}|\mathbf{z}_{t_{i-1}}, \mathbf{x})$ are Gaussian, which means that the posterior will be normally distributed too (MacKay, 2003). The computation of the parameters is somewhat involved, but it has already been done in the diffusion models literature. Following Kingma et al. (2023), we get

$$q(\mathbf{z}_{t_{i-1}}|\mathbf{z}_{t_i}, \mathbf{y}, \mathbf{x}) = \mathcal{N}\left(\mu_B(\mathbf{z}_{t_i}, t_i, \mathbf{y}, \mathbf{x}), \sigma_B(t_i, \mathbf{x})\mathbf{I}\right),$$

where

$$\mu_B(\mathbf{z}_{t_i}, t_i, \mathbf{y}, \mathbf{x}) = \sqrt{1 - \hat{\beta}_T(t_i, \mathbf{x})}\frac{\sigma(t_{i-1}, \mathbf{x})}{\sigma(t_i, \mathbf{x})}\mathbf{z}_{t_i} + \sqrt{\gamma(t_{i-1}, \mathbf{x})}\frac{\hat{\beta}_T(t_i, \mathbf{x})}{\sigma(t_i, \mathbf{x})}\mathbf{y}$$

$$= \sqrt{1 - \hat{\beta}_T(t_i, \mathbf{x})}\frac{1 - \gamma(t_{i-1}, \mathbf{x})}{1 - \gamma(t_i, \mathbf{x})}\mathbf{z}_{t_i} + \sqrt{\gamma(t_{i-1}, \mathbf{x})}\frac{\hat{\beta}_T(t_i, \mathbf{x})}{1 - \gamma(t_i, \mathbf{x})}\mathbf{y}$$

and

$$\sigma_B(t_i, \mathbf{x}) = \hat{\beta}_T(t_i, \mathbf{x})\frac{\sigma(t_{i-1}, \mathbf{x})}{\sigma(t_i, \mathbf{x})}$$

$$= \hat{\beta}_T(t_i, \mathbf{x})\frac{1 - \gamma(t_{i-1}, \mathbf{x})}{1 - \gamma(t_i, \mathbf{x})}.$$

Notice that, similar to $\hat{\beta}_T$, both $\mu_B$ and $\sigma_B$ require $t_{i-1}$ as input besides $t_i$. However, if the number of steps $T$ is fixed, then $t_i$ can be used to find $t_{i-1}$. So, same as $\hat{\beta}_T$, we write these functions as depending only on $t_i$, like $\hat{\beta}_T(t_i, \mathbf{x})$, rather than including $t_{i-1}$ as an extra parameter.

## C EVIDENCE LOWER BOUND

In this appendix, we derive the most important aspects of the loss function. Most of these details correspond to the derivations in Sohl-Dickstein et al. (2015); Ho et al. (2020) and other references, but we include them for completeness and to point out some key differences in our setup. For simplicity of notation, we denote $\mathbf{z}_{t_i}$ as $\mathbf{z}_i$ in most of this section. Let $\mathbf{x}$ be the data. We want the learned $p_\nu(\mathbf{y}|\mathbf{x})$ distribution to be as close as possible to the true distribution $p(\mathbf{y}|\mathbf{x})$. Hence, it makes sense to maximize the following log-likelihood term:

$$\mathbb{E}_{(\mathbf{y}, \mathbf{x}) \sim p(\mathbf{y}, \mathbf{x})}\left[\log p_\nu(\mathbf{y}|\mathbf{x})\right].$$

Actual implementations of diffusion models take the equivalent approach of minimizing the negative log-likelihood, so we focus on the following term:

$$\begin{aligned}
\mathcal{L} &= -\log p_\nu(\mathbf{y}|\mathbf{x}) \\
&= -\log p_\nu(\mathbf{z}_0|\mathbf{x}) \\
&= -\log \int p_\nu(\mathbf{z}_T|\mathbf{x})\prod_{i=1}^{T} p_\nu(\mathbf{z}_{i-1}|\mathbf{z}_i, \mathbf{x})d\mathbf{z}_1 \dots d\mathbf{z}_T \\
&= -\log \int p_\nu(\mathbf{z}_T|\mathbf{x})\frac{\prod_{i=1}^{T} p_\nu(\mathbf{z}_{i-1}|\mathbf{z}_i, \mathbf{x})}{\prod_{i=1}^{T} q(\mathbf{z}_i|\mathbf{z}_{i-1}, \mathbf{x})}\prod_{i=1}^{T} q(\mathbf{z}_i|\mathbf{z}_{i-1}, \mathbf{x})d\mathbf{z}_1 \dots d\mathbf{z}_T
\end{aligned}$$

Since the process is Markovian, $\prod_{i=1}^{T} q(\mathbf{z}_i|\mathbf{z}_{i-1}, \mathbf{x}) = q(\mathbf{z}_1, \ldots, \mathbf{z}_T|\mathbf{z}_0, \mathbf{x})$ and we get

$$= -\log \mathbb{E}_{(\mathbf{z}_1,\ldots,\mathbf{z}_T)\sim q(\mathbf{z}_1,\ldots,\mathbf{z}_T|\mathbf{z}_0,\mathbf{x})} \left[ p_\nu(\mathbf{z}_T|\mathbf{x}) \frac{\prod_{i=1}^{T} p_\nu(\mathbf{z}_{i-1}|\mathbf{z}_i, \mathbf{x})}{\prod_{i=1}^{T} q(\mathbf{z}_i|\mathbf{z}_{i-1}, \mathbf{x})} \right]$$

By Jensen's inequality:

$$\leq \mathbb{E}_{(\mathbf{z}_1,\ldots,\mathbf{z}_T)\sim q(\mathbf{z}_1,\ldots,\mathbf{z}_T|\mathbf{z}_0,\mathbf{x})} \left[ -\log \left( p_\nu(\mathbf{z}_T|\mathbf{x}) \frac{\prod_{i=1}^{T} p_\nu(\mathbf{z}_{i-1}|\mathbf{z}_i, \mathbf{x})}{\prod_{i=1}^{T} q(\mathbf{z}_i|\mathbf{z}_{i-1}, \mathbf{x})} \right) \right].$$

For simplicity of notation, in the appendices, we compute the loss terms before application of $\mathbb{E}_{(\mathbf{y},\mathbf{x})\sim p(\mathbf{y},\mathbf{x})} [\cdot]$. This means, for instance, that $\mathcal{L}$ depends on $(\mathbf{y}, \mathbf{x})$ and should be written as $\mathcal{L}(\mathbf{y}, \mathbf{x})$. Once again, in the interest of simplifying notation in the appendices, we write the loss terms without explicit dependence on $\mathbf{y}$ and $\mathbf{x}$. So, for instance, we write $\mathcal{L}_{\text{prior}}$ in the appendices, but in the main body of the paper, we use the more precise $\mathcal{L}_{\text{prior}}(\mathbf{y}, \mathbf{x})$.

In the following, we denote $\mathbb{E}_{(\mathbf{z}_1,\ldots,\mathbf{z}_T)\sim q(\mathbf{z}_1,\ldots,\mathbf{z}_T|\mathbf{z}_0,\mathbf{x})} [\cdot]$ by just $\mathbb{E}_{q|\mathbf{z}_0,\mathbf{x}} [\cdot]$. We continue by taking the logarithm of the whole product:

$$\mathcal{L} \leq \mathbb{E}_{q|\mathbf{z}_0,\mathbf{x}} \left[ -\log \left( p_\nu(\mathbf{z}_T|\mathbf{x}) \frac{\prod_{i=1}^{T} p_\nu(\mathbf{z}_{i-1}|\mathbf{z}_i, \mathbf{x})}{\prod_{i=1}^{T} q(\mathbf{z}_i|\mathbf{z}_{i-1}, \mathbf{x})} \right) \right]$$

$$= \mathbb{E}_{q|\mathbf{z}_0,\mathbf{x}} \left[ \sum_{i=1}^{T} \log \frac{q(\mathbf{z}_i|\mathbf{z}_{i-1}, \mathbf{x})}{p_\nu(\mathbf{z}_{i-1}|\mathbf{z}_i, \mathbf{x})} - \log p_\nu(\mathbf{z}_T|\mathbf{x}) \right]$$

$$= \mathbb{E}_{q|\mathbf{z}_0,\mathbf{x}} \left[ \sum_{i=2}^{T} \log \frac{q(\mathbf{z}_i|\mathbf{z}_{i-1}, \mathbf{x})}{p_\nu(\mathbf{z}_{i-1}|\mathbf{z}_i, \mathbf{x})} + \frac{q(\mathbf{z}_1|\mathbf{z}_0, \mathbf{x})}{p_\nu(\mathbf{z}_0|\mathbf{z}_1, \mathbf{x})} - \log p_\nu(\mathbf{z}_T|\mathbf{x}) \right]$$

$$= \mathbb{E}_{q|\mathbf{z}_0,\mathbf{x}} \left[ \sum_{i=2}^{T} \log \left( \frac{q(\mathbf{z}_{i-1}|\mathbf{z}_i, \mathbf{z}_0, \mathbf{x})}{p_\nu(\mathbf{z}_{i-1}|\mathbf{z}_i, \mathbf{x})} \frac{q(\mathbf{z}_i|\mathbf{z}_0, \mathbf{x})}{q(\mathbf{z}_{i-1}|\mathbf{z}_0, \mathbf{x})} \right) + \frac{q(\mathbf{z}_1|\mathbf{z}_0, \mathbf{x})}{p_\nu(\mathbf{z}_0|\mathbf{z}_1, \mathbf{x})} - \log p_\nu(\mathbf{z}_T|\mathbf{x}) \right]$$

$$= \mathbb{E}_{q|\mathbf{z}_0,\mathbf{x}} \left[ \sum_{i=2}^{T} \log \frac{q(\mathbf{z}_{i-1}|\mathbf{z}_i, \mathbf{z}_0, \mathbf{x})}{p_\nu(\mathbf{z}_{i-1}|\mathbf{z}_i, \mathbf{x})} + \sum_{i=2}^{T} \log \frac{q(\mathbf{z}_i|\mathbf{z}_0, \mathbf{x})}{q(\mathbf{z}_{i-1}|\mathbf{z}_0, \mathbf{x})} + \frac{q(\mathbf{z}_1|\mathbf{z}_0, \mathbf{x})}{p_\nu(\mathbf{z}_0|\mathbf{z}_1, \mathbf{x})} - \log p_\nu(\mathbf{z}_T|\mathbf{x}) \right]$$

$$= \mathbb{E}_{q|\mathbf{z}_0,\mathbf{x}} \left[ \sum_{i=2}^{T} \log \frac{q(\mathbf{z}_{i-1}|\mathbf{z}_i, \mathbf{z}_0, \mathbf{x})}{p_\nu(\mathbf{z}_{i-1}|\mathbf{z}_i, \mathbf{x})} + \log \frac{q(\mathbf{z}_T|\mathbf{z}_0, \mathbf{x})}{q(\mathbf{z}_1|\mathbf{z}_0, \mathbf{x})} + \frac{q(\mathbf{z}_1|\mathbf{z}_0, \mathbf{x})}{p_\nu(\mathbf{z}_0|\mathbf{z}_1, \mathbf{x})} - \log p_\nu(\mathbf{z}_T|\mathbf{x}) \right]$$

$$= \mathbb{E}_{q|\mathbf{z}_0,\mathbf{x}} \left[ \sum_{i=2}^{T} \log \frac{q(\mathbf{z}_{i-1}|\mathbf{z}_i, \mathbf{z}_0, \mathbf{x})}{p_\nu(\mathbf{z}_{i-1}|\mathbf{z}_i, \mathbf{x})} + \log \frac{q(\mathbf{z}_T|\mathbf{z}_0, \mathbf{x})}{p_\nu(\mathbf{z}_T|\mathbf{x})} - \log p_\nu(\mathbf{z}_0|\mathbf{z}_1, \mathbf{x}) \right]$$

$$= \mathcal{L}_{\text{prior}} + \mathcal{L}_{\text{reconstruction}} + \mathcal{L}_{\text{diffusion}}.$$

We take a look at each one of these terms in turn. To be consistent with the main body of the paper, we now return to the $\mathbf{z}_{t_i}$ notation instead of using $\mathbf{z}_i$. The *prior loss* term corresponds to

$$\mathcal{L}_{\text{prior}} = D_{\text{KL}} \left( q(\mathbf{z}_{t_T}|\mathbf{z}_{t_0}, \mathbf{x}) \, || \, p_\nu(\mathbf{z}_{t_T}|\mathbf{x}) \right),$$

and it helps ensure that the forward process ends with a similar distribution as to that with which the reverse process begins. In our setup, $p_\nu(\mathbf{z}_{t_T}|\mathbf{x})$ is fixed as a normal distribution for all $\mathbf{z}_T$ and has no trainable parameters. In many DPM setups (Ho et al., 2020; Nichol & Dhariwal, 2021) the schedule is fixed and hence $q(\mathbf{z}_{t_T}|\mathbf{z}_{t_0}, \mathbf{x})$ has no trainable parameters either, so $\mathcal{L}_{\text{prior}}$ is discarded altogether for training. However, in our framework we need the key flexibility of learning the schedule function $\gamma$, so we do not discard this term.

There are known ways to estimate and minimize $\mathcal{L}_{\text{prior}}$, since $p_\nu(\mathbf{z}_{t_T}|\mathbf{x})$ is a standard normal variable and $q(\mathbf{z}_{t_T}|\mathbf{z}_{t_0}, \mathbf{x})$ has a Gaussian distribution too. Hence, the KL divergence between the two has an analytical expression that is differentiable and can be minimized with standard techniques (see Section 3.4). On the other hand, consider the *reconstruction loss*, given by

$$\mathcal{L}_{\text{reconstruction}} = \mathbb{E}_{\mathbf{z}_{t_1} \sim q(\mathbf{z}_{t_1}|\mathbf{z}_{t_0}, \mathbf{x})} \left[ -\log p_\nu(\mathbf{z}_{t_0}|\mathbf{z}_{t_1}, \mathbf{x}) \right],$$

which helps ensure that the last step of the reverse process $p_\nu(\mathbf{z}_{t_0}|\mathbf{z}_{t_1}, \mathbf{x})$ gives the correct converse operation with respect to the forward process $q(\mathbf{z}_{t_1}|\mathbf{z}_{t_0}, \mathbf{x})$. In most DPM setups, this term is discarded or included in an altered form (Ho et al., 2020; Nichol & Dhariwal, 2021).

We choose to discard it for training, because its effect is very small when compared to $\mathcal{L}_{\text{diffusion}}$, and because the last step of the reverse process is not especially important. Consider the following: a diffusion process has to produce change in a gradual way, and in the end $p_\nu(\mathbf{z}_{t_1}|\mathbf{x})$ should be almost identical to $p_\nu(\mathbf{z}_{t_0}|\mathbf{x})$, especially when the number of steps $T$ is high or even infinite (we discuss the continuous-time case in Appendix D). The quality of $p_\nu$ is ultimately determined by the diffusion loss as a whole rather than by the last step.

Finally, consider the *diffusion loss*, given by

$$\mathcal{L}_{\text{diffusion}} = \sum_{i=2}^{T} \mathbb{E}_{(\mathbf{z}_{t_1},\ldots,\mathbf{z}_{t_T}) \sim q(\mathbf{z}_{t_1},\ldots,\mathbf{z}_{t_T}|\mathbf{z}_{t_0},\mathbf{x})} \left[ \log \frac{q(\mathbf{z}_{t_{i-1}}|\mathbf{z}_{t_i}, \mathbf{z}_{t_0}, \mathbf{x})}{p_\nu(\mathbf{z}_{t_{i-1}}|\mathbf{z}_{t_i}, \mathbf{x})} \right]$$

$$= \sum_{i=2}^{T} \underbrace{\mathbb{E}_{\mathbf{z}_{t_i} \sim q(\mathbf{z}_{t_i}|\mathbf{z}_{t_0},\mathbf{x})} \left[ D_{\text{KL}} \left( q(\mathbf{z}_{t_{i-1}}|\mathbf{z}_{t_i}, \mathbf{z}_{t_0}, \mathbf{x}) \,||\, p_\nu(\mathbf{z}_{t_{i-1}}|\mathbf{z}_{t_i}, \mathbf{x}) \right) \right]}_{\mathcal{L}_{t_i}}.$$

Each $\mathcal{L}_{t_i}$ term simplifies greatly, as we show in Appendix D. In the end, after discarding the reconstruction loss, the Evidence Lower Bound loss becomes

$$\mathcal{L}_{\text{ELBO}} = \mathcal{L}_{\text{prior}} + \mathcal{L}_{\text{diffusion}} = \mathcal{L}_{\text{prior}} + \sum_{i=2}^{T} \mathcal{L}_{t_i}.$$

## D    DIFFUSION LOSS AND TRAINING

From Appendix C, we know that one of the terms in the Evidence Lower Bound is

$$\mathcal{L}_{\text{diffusion}} = \sum_{i=2}^{T} \mathcal{L}_{t_i} = \sum_{i=2}^{T} \mathbb{E}_{\mathbf{z}_{t_i} \sim q(\mathbf{z}_{t_i}|\mathbf{z}_{t_0},\mathbf{x})} \left[ D_{\text{KL}} \left( q(\mathbf{z}_{t_{i-1}}|\mathbf{z}_{t_i}, \mathbf{z}_{t_0}, \mathbf{x}) \,||\, p_\nu(\mathbf{z}_{t_{i-1}}|\mathbf{z}_{t_i}, \mathbf{x}) \right) \right].$$

We now simplify this term, essentially following Kingma et al. (2023). We include this material for completeness and to highlight any differences that may come about from conditioning on $\mathbf{x}$. Each $\mathcal{L}_{t_i}$ includes the KL divergence between two Gaussian distributions, which has an analytic expression. Moreover, both distributions have the same variance. From Section 3.1, recall that by definition

$$p_\nu(\mathbf{z}_{t_{i-1}}|\mathbf{z}_{t_i}, \mathbf{x}) = q(\mathbf{z}_{t_{i-1}}|\mathbf{z}_{t_i}, \mathbf{z}_{t_0} = \hat{\mathbf{y}}_\nu(z_{t_i}, t_i, \mathbf{x}), \mathbf{x}),$$

Now, from Appendix B.2 recall that the variance $\sigma_B$ of $q(\mathbf{z}_{t_{i-1}}|\mathbf{z}_{t_i}, \mathbf{z}_{t_0}, \mathbf{x})$ only depends on $t_i$ and $\mathbf{x}$, and it does not depend on $\mathbf{y}$. Hence, as we mentioned before, both $q(\mathbf{z}_{t_{i-1}}|\mathbf{z}_{t_i}, \mathbf{z}_{t_0}, \mathbf{x})$ and $p_\nu(\mathbf{z}_{t_{i-1}}|\mathbf{z}_{t_i}, \mathbf{x})$ are Gaussian distributions with exactly the same variance. This means that the KL divergence between them simplifies to

$$D_{\text{KL}} \left( q(\mathbf{z}_{t_{i-1}}|\mathbf{z}_{t_i}, \mathbf{z}_{t_0}, \mathbf{x}) \,||\, p_\nu(\mathbf{z}_{t_{i-1}}|\mathbf{z}_{t_i}, \mathbf{x}) \right) = \frac{1}{2\sigma_B(t_i, \mathbf{x})} \|\mu - \mu_\nu\|_2^2,$$

where $\mu$ and $\mu_\nu$ are the means of $q(\mathbf{z}_{t_{i-1}}|\mathbf{z}_{t_i}, \mathbf{z}_{t_0}, \mathbf{x})$ and $p_\nu(\mathbf{z}_{t_{i-1}}|\mathbf{z}_{t_i}, \mathbf{x})$, respectively. Now, from Appendix B.2, we know that

$$\mu = \mu_B(\mathbf{z}_{t_i}, t_i, \mathbf{y}, \mathbf{x}) = \sqrt{1 - \hat{\beta}_T(t_i, \mathbf{x})} \frac{1 - \gamma(t_{i-1}, \mathbf{x})}{1 - \gamma(t_i, \mathbf{x})} \mathbf{z}_{t_i} + \sqrt{\gamma(t_{i-1}, \mathbf{x})} \frac{\hat{\beta}_T(t_i, \mathbf{x})}{1 - \gamma(t_i, \mathbf{x})} \mathbf{y},$$

$$\mu_\nu = \mu_B(\mathbf{z}_{t_i}, t_i, \mathbf{y}, \mathbf{x}) = \sqrt{1 - \hat{\beta}_T(t_i, \mathbf{x})} \frac{1 - \gamma(t_{i-1}, \mathbf{x})}{1 - \gamma(t_i, \mathbf{x})} \mathbf{z}_{t_i} + \sqrt{\gamma(t_{i-1}, \mathbf{x})} \frac{\hat{\beta}_T(t_i, \mathbf{x})}{1 - \gamma(t_i, \mathbf{x})} \hat{\mathbf{y}}_\nu$$

$$\implies \mu - \mu_\nu = \sqrt{\gamma(t_{i-1}, \mathbf{x})} \frac{\hat{\beta}_T(t_i, \mathbf{x})}{1 - \gamma(t_i, \mathbf{x})} (\mathbf{y} - \hat{\mathbf{y}}_\nu).$$

Since $\sigma_B(t_i, \mathbf{x}) = \hat{\beta}_T(t_i, \mathbf{x})(1 - \gamma(t_{i-1}, \mathbf{x}))/(1 - \gamma(t_i, \mathbf{x}))$, we can conclude that

$$
D_{\mathrm{KL}}\Big(q(\mathbf{z}_{t_{i-1}}|\mathbf{z}_{t_i}, \mathbf{z}_{t_0}, \mathbf{x}) \,\|\, p_\nu(\mathbf{z}_{t_{i-1}}|\mathbf{z}_{t_i}, \mathbf{x})\Big)
$$

$$
= \frac{1}{2\sigma_B(t_i, \mathbf{x})}\|\mu - \mu_\nu\|_2^2
$$

$$
= \frac{1 - \gamma(t_i, \mathbf{x})}{2\hat{\beta}_T(t_i, \mathbf{x})(1 - \gamma(t_{i-1}, \mathbf{x}))}\left\|\sqrt{\gamma(t_{i-1}, \mathbf{x})}\frac{\hat{\beta}_T(t_i, \mathbf{x})}{1 - \gamma(t_i, \mathbf{x})}(\mathbf{y} - \hat{\mathbf{y}}_\nu)\right\|_2^2
$$

$$
= \frac{\gamma(t_{i-1}, \mathbf{x})\hat{\beta}_T(t_i, \mathbf{x})}{2(1 - \gamma(t_{i-1}, \mathbf{x}))(1 - \gamma(t_i, \mathbf{x}))}\|\mathbf{y} - \hat{\mathbf{y}}_\nu\|_2^2
$$

$$
= \frac{\gamma(t_{i-1}, \mathbf{x})\left(1 - \frac{\gamma(t_i, \mathbf{x})}{\gamma(t_{i-1}, \mathbf{x})}\right)}{2(1 - \gamma(t_{i-1}, \mathbf{x}))(1 - \gamma(t_i, \mathbf{x}))}\|\mathbf{y} - \hat{\mathbf{y}}_\nu\|_2^2
$$

$$
= \frac{\gamma(t_{i-1}, \mathbf{x}) - \gamma(t_i, \mathbf{x})}{2(1 - \gamma(t_{i-1}, \mathbf{x}))(1 - \gamma(t_i, \mathbf{x}))}\|\mathbf{y} - \hat{\mathbf{y}}_\nu\|_2^2
$$

$$
= \frac{1}{2}\left(\frac{\gamma(t_{i-1}, \mathbf{x})}{1 - \gamma(t_{i-1}, \mathbf{x})} - \frac{\gamma(t_i, \mathbf{x})}{1 - \gamma(t_i, \mathbf{x})}\right)\|\mathbf{y} - \hat{\mathbf{y}}_\nu\|_2^2
$$

$$
= \frac{1}{2}\left(\frac{\gamma(t_{i-1}, \mathbf{x})}{\sigma(t_{i-1}, \mathbf{x})} - \frac{\gamma(t_i, \mathbf{x})}{\sigma(t_i, \mathbf{x})}\right)\|\mathbf{y} - \hat{\mathbf{y}}_\nu\|_2^2,
$$

where $\hat{\mathbf{y}}_\nu = \hat{\mathbf{y}}_\nu(\mathbf{z}_{t_i}, t_i, \mathbf{x})$ The fraction $\gamma(t, \mathbf{x})/\sigma(t, \mathbf{x})$ is what Kingma et al. (2023) call the *signal-to-noise ratio* and it should be a decreasing function of time, so the above expression makes sense as a nonnegative loss. Since we are in the variance-preserving case where $\sigma(t, \mathbf{x}) = 1 - \gamma(t, \mathbf{x})$, it is sufficient that $\gamma(t, \mathbf{x})$ is a decreasing function of time to guarantee that the above expression is nonnegative. Recapping, we have that

$$
\mathcal{L}_{t_i} = \frac{1}{2}\mathbb{E}_{\mathbf{z}_{t_i} \sim q(\mathbf{z}_{t_i}|\mathbf{z}_{t_0}, \mathbf{x})}\left[\left(\frac{\gamma(t_{i-1}, \mathbf{x})}{\sigma(t_{i-1}, \mathbf{x})} - \frac{\gamma(t_i, \mathbf{x})}{\sigma(t_i, \mathbf{x})}\right)\|\mathbf{y} - \hat{\mathbf{y}}_\nu(\mathbf{z}_{t_i}, t_i, \mathbf{x})\|_2^2\right].
$$

By the definition of the forward process $q(\mathbf{z}_{t_i}|\mathbf{z}_{t_0}, \mathbf{x})$ in Section 3.1, a variable $\mathbf{z}_{t_i} \sim q(\mathbf{z}_{t_i}|\mathbf{z}_{t_0}, \mathbf{x})$ can be reparametrized as $\mathbf{z}_{t_i}(\epsilon) = \sqrt{\gamma(t_i, \mathbf{x})}\mathbf{y} + \sqrt{\sigma(t_i, \mathbf{x})}\epsilon$ with $\epsilon \sim \mathcal{N}(\mathbf{0}, \mathbf{I})$. This means that we can write the $i$-th term of the diffusion loss as

$$
\mathcal{L}_{t_i} = \frac{1}{2}\mathbb{E}_{\epsilon \sim \mathcal{N}(0, \mathbf{I})}\left[\left(\frac{\gamma(t_{i-1}, \mathbf{x})}{\sigma(t_{i-1}, \mathbf{x})} - \frac{\gamma(t_i, \mathbf{x})}{\sigma(t_i, \mathbf{x})}\right)\|\mathbf{y} - \hat{\mathbf{y}}_\nu(\mathbf{z}_{t_i}(\epsilon), t_i, \mathbf{x})\|_2^2\right].
$$

Putting everything together, the diffusion loss is given by

$$
\mathcal{L}_{\mathrm{diffusion}} = \sum_{i=2}^{T}\mathcal{L}_{t_i}
$$

$$
= \frac{1}{2}\sum_{i=2}^{T}\mathbb{E}_{\epsilon \sim \mathcal{N}(0, \mathbf{I})}\left[\left(\frac{\gamma(t_{i-1}, \mathbf{x})}{\sigma(t_{i-1}, \mathbf{x})} - \frac{\gamma(t_i, \mathbf{x})}{\sigma(t_i, \mathbf{x})}\right)\|\mathbf{y} - \hat{\mathbf{y}}_\nu(\mathbf{z}_{t_i}(\epsilon), t_i, \mathbf{x})\|_2^2\right]
$$

$$
= \frac{1}{2}\sum_{i=2}^{T}\mathbb{E}_{\epsilon \sim \mathcal{N}(0, \mathbf{I})}\left[\left(\frac{\gamma(t_{i-1}, \mathbf{x})}{1 - \gamma(t_{i-1}, \mathbf{x})} - \frac{\gamma(t_i, \mathbf{x})}{1 - \gamma(t_i, \mathbf{x})}\right)\|\mathbf{y} - \hat{\mathbf{y}}_\nu(\mathbf{z}_{t_i}(\epsilon), t_i, \mathbf{x})\|_2^2\right].
$$

## E   ESTIMATORS FOR DIFFUSION LOSS

### E.1   MONTE CARLO ESTIMATOR FOR DIFFUSION LOSS

From Appendix D, we know that

$$
\mathcal{L}_{\mathrm{diffusion}} = \frac{1}{2}\sum_{i=2}^{T}\mathbb{E}_{\epsilon \sim \mathcal{N}(\mathbf{0}, \mathbf{I})}\left[\left(\frac{\gamma(t_{i-1}, \mathbf{x})}{\sigma(t_{i-1}, \mathbf{x})} - \frac{\gamma(t_i, \mathbf{x})}{\sigma(t_i, \mathbf{x})}\right)\|\mathbf{y} - \hat{\mathbf{y}}_\nu(\mathbf{z}_{t_i}(\epsilon), t_i, \mathbf{x})\|_2^2.\right]
$$

For large $T$, computing the sum becomes computationally expensive. By linearity of expectation:

$$= \frac{1}{2}\mathbb{E}_{\epsilon\sim\mathcal{N}(\mathbf{0},\mathbf{I})}\left[\sum_{i=2}^{T}\left(\frac{\gamma(t_{i-1},\mathbf{x})}{\sigma(t_{i-1},\mathbf{x})} - \frac{\gamma(t_i,\mathbf{x})}{\sigma(t_i,\mathbf{x})}\right)\|\mathbf{y} - \hat{\mathbf{y}}_\nu(\mathbf{z}_{t_i}(\epsilon),t_i,\mathbf{x})\|_2^2\right]$$

$$= \frac{T-1}{2}\mathbb{E}_{\epsilon\sim\mathcal{N}(\mathbf{0},\mathbf{I})}\left[\frac{1}{T-1}\sum_{i=2}^{T}\left(\frac{\gamma(t_{i-1},\mathbf{x})}{\sigma(t_{i-1},\mathbf{x})} - \frac{\gamma(t_i,\mathbf{x})}{\sigma(t_i,\mathbf{x})}\right)\|\mathbf{y} - \hat{\mathbf{y}}_\nu(\mathbf{z}_{t_i}(\epsilon),t_i,\mathbf{x})\|_2^2\right].$$

We can recognize the expression in brackets as an expected value, where $i$ is chosen uniformly at random from $\{2,\ldots,T\}$ with probability $1/(T-1)$. In other words, we can rewrite $\mathcal{L}_{\text{diffusion}}$ as:

$$\mathcal{L}_T(\mathbf{y},\mathbf{x}) := \frac{T-1}{2}\mathbb{E}_{\epsilon\sim\mathcal{N}(\mathbf{0},\mathbf{I}),i\sim U_{\{2,\ldots,T\}}}\left[\left(\frac{\gamma(t_{i-1},\mathbf{x})}{\sigma(t_{i-1},\mathbf{x})} - \frac{\gamma(t_i,\mathbf{x})}{\sigma(t_i,\mathbf{x})}\right)\|\mathbf{y} - \hat{\mathbf{y}}_\nu(\mathbf{z}_{t_i}(\epsilon),t_i,\mathbf{x})\|_2^2\right],$$

where $U_A$ represents the discrete uniform distribution over a finite set $A$. As mentioned in Section 3.3, the advantage of writing the loss term like this is that it provides a straightforward Monte Carlo estimator, by taking samples $\epsilon\sim\mathcal{N}(\mathbf{0},\mathbf{I})$ and $i\sim U_{\{2,\ldots,T\}}$.

### E.2 CONTINUOUS-TIME DIFFUSION LOSS

We mention again that throughout the appendices, we sometimes omit writing function parameters to simplify notation, mainly in the case of the loss terms. For instance, we write $\mathcal{L}_T$ instead of $\mathcal{L}_T(\mathbf{y},\mathbf{x})$.

Recall from Section 3.1 that $\mathbf{z}_{t_i} = i/T$, so intuitively the limit case $T\to\infty$ takes us into a continuous-time diffusion process, which can be described by a stochastic differential equation. This fact has been noticed before in the literature and it allows to present both diffusion models and score-based generative models as part of the same framework (Song et al., 2021).

In our case, we are mostly interested in how the loss term $\mathcal{L}_T$ changes when $T$ goes to infinity. Kingma et al. (2023) give the following result when taking $T\to\infty$:

$$\mathcal{L}_\infty := \lim_{T\to\infty}\mathcal{L}_\infty = -\frac{1}{2}\mathbb{E}_{\epsilon\sim\mathcal{N}(\mathbf{0},\mathbf{I})}\left[\int_0^1\text{SNR}'(t,\mathbf{x})\|\mathbf{y} - \hat{\mathbf{y}}_\nu(\mathbf{z}_t(\epsilon),t,\mathbf{x})\|_2^2 dt\right] \tag{7}$$

$$= -\frac{1}{2}\mathbb{E}_{\epsilon\sim\mathcal{N}(\mathbf{0},\mathbf{I}),t\sim U([0,1])}\left[\text{SNR}'(t,\mathbf{x})\|\mathbf{y} - \hat{\mathbf{y}}_\nu(\mathbf{z}_t(\epsilon),t,\mathbf{x})\|_2^2\right],$$

where $\text{SNR}(t,\mathbf{x})$ represents the *signal-to-noise ratio* at time $t$ and is defined as $\gamma(t,\mathbf{x})/\sigma(t,\mathbf{x})$. Throughout this section, for simplicity of notation we use $\text{SNR}'(t,\mathbf{x})$ to denote the partial derivative of $\text{SNR}(t,\mathbf{x})$ with respect to the time $t$, and analogously for $\text{SNR}''(t,\mathbf{x})$.

We are interested in better understanding this result, in particular regarding sufficient conditions under which the above equality holds, and its rate of convergence. This provides an interesting regularization idea for the training process, which we discuss in Section 3.4. Starting from the expressions for $\mathcal{L}_T$ and $\mathcal{L}_{\text{diffusion}}$ given in Appendices D and E.1, we know that

$$\mathcal{L}_T = \mathcal{L}_{\text{diffusion}}$$

$$= \frac{1}{2}\mathbb{E}_{\epsilon\sim\mathcal{N}(\mathbf{0},\mathbf{I})}\left[\sum_{i=2}^{T}\left(\frac{\gamma(t_{i-1},\mathbf{x})}{\sigma(t_{i-1},\mathbf{x})} - \frac{\gamma(t_i,\mathbf{x})}{\sigma(t_i,\mathbf{x})}\right)\|\mathbf{y} - \hat{\mathbf{y}}_\nu(\mathbf{z}_{t_i}(\epsilon),t_i,\mathbf{x})\|_2^2\right]$$

$$= \frac{1}{2}\mathbb{E}_{\epsilon\sim\mathcal{N}(\mathbf{0},\mathbf{I})}\left[\sum_{i=2}^{T}(\text{SNR}(t_{i-1},\mathbf{x}) - \text{SNR}(t_i,\mathbf{x}))\|\mathbf{y} - \hat{\mathbf{y}}_\nu(\mathbf{z}_{t_i}(\epsilon),t_i,\mathbf{x})\|_2^2\right]$$

$$= -\frac{1}{2}\mathbb{E}_{\epsilon\sim\mathcal{N}(\mathbf{0},\mathbf{I})}\left[\sum_{i=2}^{T}(\text{SNR}(t_i,\mathbf{x}) - \text{SNR}(t_{i-1},\mathbf{x}))\|\mathbf{y} - \hat{\mathbf{y}}_\nu(\mathbf{z}_{t_i}(\epsilon),t_i,\mathbf{x})\|_2^2\right]$$

$$= -\frac{1}{2}\mathbb{E}_{\epsilon\sim\mathcal{N}(\mathbf{0},\mathbf{I})}\left[S_T\right],$$

where we have denoted the sum inside the brackets as $S_T$. Now, we denote $h_T = 1/T$ and rewrite $S_T$ in the following way, such that a derivative-like expression appears:

$$
\begin{aligned}
S_T &= \sum_{i=2}^{T} (\text{SNR}(t_i, \mathbf{x}) - \text{SNR}(t_{i-1}, \mathbf{x})) \|\mathbf{y} - \hat{\mathbf{y}}_\nu(\mathbf{z}_{t_i}(\epsilon), t_i, \mathbf{x})\|_2^2 \\
&= \sum_{i=2}^{T} \frac{\text{SNR}(t_i, \mathbf{x}) - \text{SNR}(t_{i-1}, \mathbf{x})}{h_T} \|\mathbf{y} - \hat{\mathbf{y}}_\nu(\mathbf{z}_{t_i}(\epsilon), t_i, \mathbf{x})\|_2^2 h_T \\
&= \sum_{i=2}^{T} f^T(t_i) h_T,
\end{aligned}
$$

where the function $f^T$ is defined as

$$
f^T(t) = \frac{\text{SNR}(t, \mathbf{x}) - \text{SNR}(t - h_T, \mathbf{x})}{h_T} \|\mathbf{y} - \hat{\mathbf{y}}_\nu(\mathbf{z}_t(\epsilon), t, \mathbf{x})\|_2^2.
$$

Now, we need to understand the behaviour of $S_T$ as $T$ goes to infinity. Assuming that both SNR and $\hat{\mathbf{y}}_\nu$ are continuously differentiable, it is easy to see that $f^T(t)$ converges pointwise:

$$
f^T(t) \xrightarrow{T \to \infty} g(t) := \text{SNR}'(t, \mathbf{x}) \|\mathbf{y} - \hat{\mathbf{y}}_\nu(\mathbf{z}_t(\epsilon), t, \mathbf{x})\|_2^2.
$$

We have established pointwise convergence of $f^T$ to $g$, but what we are really interested in is to understand the convergence of $\mathcal{L}_T$ to $\mathcal{L}_\infty$, defined as in equation (7). Notice that

$$
\mathcal{L}_T = -\frac{1}{2} \mathbb{E}_{\epsilon \sim \mathcal{N}(\mathbf{0}, \mathbf{I})} [S_T], \quad \mathcal{L}_\infty = -\frac{1}{2} \mathbb{E}_{\epsilon \sim \mathcal{N}(\mathbf{0}, \mathbf{I})} \left[ \int_0^1 g(t) dt \right].
$$

Denote the integral $\int_0^1 g(t) dt$ by $I$. The above expression strongly suggests that we should understand the relation between $S_T$ and $I$ when $T$ goes to infinity, in order to establish the convergence of $\mathcal{L}_T$ to $\mathcal{L}_\infty$. With that in mind, for any given positive integer $T$, the difference between the sum $S_T$ and the integral $I$ can be bounded as follows:

$$
\begin{aligned}
|S_T - I| &= \left| \sum_{i=2}^{T} f^T(t_i) h_T - \int_0^1 g(t) dt \right| \\
&\leq \underbrace{\left| \sum_{i=2}^{T} f^T(t_i) h_T - \sum_{i=2}^{T} g(t_i) h_T \right|}_{E_1} + \underbrace{\left| \sum_{i=1}^{T} g(t_i) h_T - \int_0^1 g(t) dt \right|}_{E_2} + \underbrace{|g(t_1) h_T|}_{E_3}.
\end{aligned}
$$

We have assumed that both SNR and $\hat{\mathbf{y}}_\nu$ are continuously differentiable, which makes $g$ continuously differentiable too. Given the smoothness of $g$, for large $T$ we have

$$
E_3 \approx \frac{|g(0)|}{T}.
$$

On the other hand, $E_2$ is just the difference between a Riemann Sum for $g$ and its integral. For a uniform partition such as ours (i.e. $t_i = i/T$) the Riemann Sum converges as $O(1/T)$. In particular, since $g$ is differentiable, for large $T$ we have (Owens, 2014)

$$
E_2 \approx \frac{|g(1) - g(0)|}{T}.
$$

On the other hand, notice that

$$
\begin{aligned}
E_1 &= \left| \sum_{i=2}^{T} \left( f^T(t_i) - g(t_i) \right) h_T \right| \\
&= \left| \sum_{i=2}^{T} \underbrace{\left( \frac{\text{SNR}(t_i, \mathbf{x}) - \text{SNR}(t_i - h_T, \mathbf{x})}{h_T} - \text{SNR}'(t_i, \mathbf{x}) \right)}_{D_i} \|\mathbf{y} - \hat{\mathbf{y}}_\nu(\mathbf{z}_{t_i}(\epsilon), t_i, \mathbf{x})\|_2^2 h_T \right|.
\end{aligned}
$$

Assuming further that SNR is twice differentiable, the forward difference approximates the derivative:

$$D_i = \frac{\text{SNR}(t_i, \mathbf{x}) - \text{SNR}(t_i - h_T, \mathbf{x})}{h_T} - \text{SNR}'(t_i, \mathbf{x}) \leq \frac{1}{2T} \big\| \text{SNR}''(\cdot, \mathbf{x}) \big\|_{L^\infty([t_{i-1}, t_i])}.$$

Replacing in the bound for $E_1$, we get

$$E_1 = \left| \sum_{i=2}^{T} D_i \| \mathbf{y} - \hat{\mathbf{y}}_\nu(\mathbf{z}_{t_i}(\epsilon), t_i, \mathbf{x}) \|_2^2 h_T \right|$$

$$\leq \sum_{i=2}^{T} |D_i| \| \mathbf{y} - \hat{\mathbf{y}}_\nu(\mathbf{z}_{t_i}(\epsilon), t_i, \mathbf{x}) \|_2^2 h_T$$

$$\leq \sum_{i=2}^{T} \frac{1}{2T} \big\| \text{SNR}''(\cdot, \mathbf{x}) \big\|_{L^\infty([t_{i-1}, t_i])} \| \mathbf{y} - \hat{\mathbf{y}}_\nu(\mathbf{z}_{t_i}(\epsilon), t_i, \mathbf{x}) \|_2^2 h_T$$

$$= \frac{1}{2T} \sum_{i=2}^{T} \big\| \text{SNR}''(\cdot, \mathbf{x}) \big\|_{L^\infty([t_{i-1}, t_i])} \| \mathbf{y} - \hat{\mathbf{y}}_\nu(\mathbf{z}_{t_i}(\epsilon), t_i, \mathbf{x}) \|_2^2 h_T$$

Approximating the Riemann Sum by its integral

$$\approx \frac{1}{2T} \int_0^1 \big| \text{SNR}''(t, \mathbf{x}) \big| \| \mathbf{y} - \hat{\mathbf{y}}_\nu(\mathbf{z}_t(\epsilon), t, \mathbf{x}) \|_2^2 dt.$$

Putting everything together, we get

$$|S_T - I|$$
$$\lesssim E_3 + E_2 + E_1$$
$$\leq \frac{|g(0)|}{T} + \frac{|g(1) - g(0)|}{T} + \frac{1}{2T} \int_0^1 \big| \text{SNR}''(t, \mathbf{x}) \big| \| \mathbf{y} - \hat{\mathbf{y}}_\nu(\mathbf{z}_t(\epsilon), t, \mathbf{x}) \|_2^2 dt$$

Replacing $g$ and using Cauchy-Schwarz inequality:

$$\leq \frac{\left| \text{SNR}'(0, \mathbf{x}) \| \mathbf{y} - \hat{\mathbf{y}}_\nu(\mathbf{z}_0(\epsilon), 0, \mathbf{x}) \|_2^2 \right|}{T}$$
$$+ \frac{\left| \text{SNR}'(1, \mathbf{x}) \| \mathbf{y} - \hat{\mathbf{y}}_\nu(\mathbf{z}_1(\epsilon), 1, \mathbf{x}) \|_2^2 - \text{SNR}'(0, \mathbf{x}) \| \mathbf{y} - \hat{\mathbf{y}}_\nu(\mathbf{z}_0(\epsilon), 0, \mathbf{x}) \|_2^2 \right|}{T}$$
$$+ \frac{1}{2T} \left( \int_0^1 \big| \text{SNR}''(t, \mathbf{x}) \big|^2 dt \right)^{1/2} \left( \int_0^1 \| \mathbf{y} - \hat{\mathbf{y}}_\nu(\mathbf{z}_t(\epsilon), t, \mathbf{x}) \|_2^4 dt \right)^{1/2}.$$

If the model is good enough, we would expect $\| \mathbf{y} - \hat{\mathbf{y}}_\nu(\mathbf{z}_0(\epsilon), 0, \mathbf{x}) \|_2^2 \approx 0$:

$$\approx \frac{|\text{SNR}'(1, \mathbf{x})| \| \mathbf{y} - \hat{\mathbf{y}}_\nu(\mathbf{z}_1(\epsilon), 1, \mathbf{x}) \|_2^2}{T} + \frac{\| \text{SNR}''(\cdot, \mathbf{x}) \|_{L_2([0,1])}}{2T} \left( \int_0^1 \| \mathbf{y} - \hat{\mathbf{y}}_\nu(\mathbf{z}_t(\epsilon), t, \mathbf{x}) \|_2^4 dt \right)^{1/2}.$$

Summarizing, the main assumptions so far are that SNR is twice differentiable and that $\hat{\mathbf{y}}_\nu$ is continuously differentiable. With that in mind, we can use the last expression to prove that $S_T$ converges to $I$ when $T$ goes to infinity. Adding an extra condition of boundedness for $\hat{\mathbf{y}}_\nu$, we can use the Dominated Convergence Theorem to conclude that

$$\lim_{T \to \infty} \mathcal{L}_T = \lim_{T \to \infty} -\frac{1}{2} \mathbb{E}_{\epsilon \sim \mathcal{N}(\mathbf{0}, \mathbf{I})} [S_T] = -\frac{1}{2} \mathbb{E}_{\epsilon \sim \mathcal{N}(\mathbf{0}, \mathbf{I})} \left[ \lim_{T \to \infty} S_T \right] = -\frac{1}{2} \mathbb{E}_{\epsilon \sim \mathcal{N}(\mathbf{0}, \mathbf{I})} \left[ \int_0^1 g(t) dt \right].$$

We thus establish sufficient conditions for the convergence of $\mathcal{L}_T$ to $\mathcal{L}_\infty$. Our analysis shows that, in some way, this convergence is of order $O(1/T)$ and its speed depends on the magnitude of $\text{SNR}'$ and $\text{SNR}''$. Also, it depends on the approximation quality of the neural network model $\hat{\mathbf{y}}_\nu$, which is

consistent with the finding in Kingma et al. (2023) that increasing the number of steps $T$ decreases the error on the condition that the model is good enough.

This provides an interesting insight. Kingma et al. (2023) show that in the continuous-time limit, the diffusion loss is invariant to the choice of SNR function, as long as it fulfills a few basic conditions. However, in any computational implementation, continuous time cannot truly exist, so the rate of convergence to the continuous-time case does matter. This means that for choices of SNR with ill-behaved derivatives, or for a model $\hat{\mathbf{y}}_\nu$ that is not good enough, the "invariance under the choice of SNR" does not necessarily hold. This gives intuition for the regularization introduced in Section 3.4.

### E.3 NOISE PREDICTION MODEL

From Appendix E.2, we know that the continuous-time diffusion loss is given by

$$\mathcal{L}_\infty = -\frac{1}{2}\mathbb{E}_{\epsilon\sim\mathcal{N}(\mathbf{0},\mathbf{I}),t\sim U([0,1])}\left[\text{SNR}'(t,\mathbf{x})\|\mathbf{y} - \hat{\mathbf{y}}_\nu(\mathbf{z}_t(\epsilon),t,\mathbf{x})\|_2^2\right].$$

By the definition of the forward process $q(\mathbf{z}_t|\mathbf{y},\mathbf{x})$ in Section 3.1, a variable $\mathbf{z}_t \sim q(\mathbf{z}_t|\mathbf{y},\mathbf{x})$ can be reparametrized as $\mathbf{z}_t = \sqrt{\gamma(t,\mathbf{x})}\mathbf{y} + \sqrt{\sigma(t,\mathbf{x})}\epsilon$ with $\epsilon \sim \mathcal{N}(\mathbf{0},\mathbf{I})$. Rearranging the terms:

$$\mathbf{y} = \frac{1}{\sqrt{\gamma(t,\mathbf{x})}}\left(\mathbf{z}_t - \sqrt{\sigma(t,\mathbf{x})}\epsilon\right). \tag{8}$$

Our model for predicting $\mathbf{y}$ is $\hat{\mathbf{y}}_\nu(\mathbf{z}_t(\epsilon),t,\mathbf{x})$. From equation (8), notice that $\hat{\mathbf{y}}_\nu$ receives, as explicit input, all the values necessary to compute $\mathbf{y}$ excepting $\epsilon$. As a consequence, we can follow Ho et al. (2020) and repurpose $\hat{\mathbf{y}}_\nu$ into a *noise prediction model* $\hat{\epsilon}_\nu$ by parametrizing

$$\hat{\mathbf{y}}_\nu = \frac{1}{\sqrt{\gamma(t,\mathbf{x})}}\left(\mathbf{z}_t - \sqrt{\sigma(t,\mathbf{x})}\hat{\epsilon}_\nu\right).$$

This means that the diffusion loss takes the form

$$\begin{aligned}
\mathcal{L}_\infty &= -\frac{1}{2}\mathbb{E}_{\epsilon\sim\mathcal{N}(\mathbf{0},\mathbf{I}),t\sim U([0,1])}\left[\text{SNR}'(t,\mathbf{x})\|\mathbf{y} - \hat{\mathbf{y}}_\nu(\mathbf{z}_t(\epsilon),t,\mathbf{x})\|_2^2\right] \\
&= -\frac{1}{2}\mathbb{E}_{\epsilon\sim\mathcal{N}(\mathbf{0},\mathbf{I}),t\sim U([0,1])}\left[\text{SNR}'(t,\mathbf{x})\frac{\sigma(t,\mathbf{x})}{\gamma(t,\mathbf{x})}\|\epsilon - \hat{\epsilon}_\nu(\mathbf{z}_t(\epsilon),t,\mathbf{x})\|_2^2\right] \\
&= -\frac{1}{2}\mathbb{E}_{\epsilon\sim\mathcal{N}(\mathbf{0},\mathbf{I}),t\sim U([0,1])}\left[\frac{\text{SNR}'(t,\mathbf{x})}{\text{SNR}(t,\mathbf{x})}\|\epsilon - \hat{\epsilon}_\nu(\mathbf{z}_t(\epsilon),t,\mathbf{x})\|_2^2\right].
\end{aligned}$$

This is the form of the diffusion loss that we use as part of the loss function in the end (see Section 3.4). In experiments, we get better results by dropping the rational term $\text{SNR}'(t,\mathbf{x})/\text{SNR}(t,\mathbf{x})$, which is consistent with the approach in Ho et al. (2020). Otherwise, the training process can converge to trivial solutions that minimize $\mathcal{L}_\infty$ by making $\text{SNR}'(t,\mathbf{x}) \approx 0$ for all $t,\mathbf{x}$. In the end, the actual form of the diffusion loss for our method is given by

$$\hat{\mathcal{L}}_\infty = -\frac{1}{2}\mathbb{E}_{\epsilon\sim\mathcal{N}(\mathbf{0},\mathbf{I}),t\sim U([0,1])}\left[\|\epsilon - \hat{\epsilon}_\nu(\mathbf{z}_t(\epsilon),t,\mathbf{x})\|_2^2\right].$$

## F CONTINUOUS-TIME SCHEDULE

In Appendix B.1, for all $i \in \{1,\ldots,T\}$ we define

$$\hat{\beta}_T(t_i,\mathbf{x}) = \mathbf{1} - \frac{\gamma(t_i,\mathbf{x})}{\gamma(t_{i-1},\mathbf{x})}.$$

Now, we want to study the relationship between functions $\gamma$ and $\hat{\beta}_T$ when $T$ goes to infinity and we move into a continuous-time framework. Since we want a diffusion process to be smooth and free of sudden jumps, we require that $\gamma$ be least continuous on $[0,1]$ and continuously differentiable on $(0,1)$. By definition, notice that for any $i \in \{1,\ldots,T\}$

$$\gamma(t_i,\mathbf{x}) = \gamma(t_{i-1},\mathbf{x})(\mathbf{1} - \hat{\beta}_T(t_i,\mathbf{x})). \tag{9}$$

Recall that the discretization we are using assumes a number of steps $T \in \mathbb{N}$, and we use the uniform partition of $[0, 1]$ defined by $\{t_i\}_{i=0}^T$ where $t_i = i/T$. Now, take any $t \in [0, 1]$ and define $t' = \lceil tT \rceil / T$. Notice that $t'$ is an element of the partition $\{t_i\}$ defined above, and that $t' \to t$ when $T \to \infty$. Then, equation (9) implies

$$\gamma(t', \mathbf{x}) = \gamma(t' - h_T, \mathbf{x}) \left( \mathbf{1} - \hat{\beta}_T(t', \mathbf{x}) \right).$$

Now, assume there exists a continuous function $\beta(t, \mathbf{x})$ such that

$$\hat{\beta}_T(t', \mathbf{x}) = \beta(t', \mathbf{x})/T \tag{10}$$

for all $t', \mathbf{x}$. A function like this would allow us to calculate $\hat{\beta}_T$ for any discretization, so we are interested in learning more about $\beta$. Denoting $h_T = 1/T$, we have

$$\gamma(t', \mathbf{x}) = \gamma(t' - h_T, \mathbf{x}) \left( \mathbf{1} - \beta(t', \mathbf{x}) h_T \right)$$
$$= \gamma(t' - h_T, \mathbf{x}) - \gamma(t' - h_T, \mathbf{x}) \beta(t', \mathbf{x}) h_T$$
$$\implies \gamma(t', \mathbf{x}) - \gamma(t' - h_T, \mathbf{x}) = -\gamma(t' - h_T, \mathbf{x}) \beta(t', \mathbf{x}) h_T$$
$$\implies \frac{\gamma(t', \mathbf{x}) - \gamma(t' - h_T, \mathbf{x})}{h_T} = -\gamma(t' - h_T, \mathbf{x}) \beta(t', \mathbf{x}).$$

We can now take the limit $T \to \infty$ (which means $h_T \to 0$ and $t' \to t$). With our assumption that $\gamma$ is continuously differentiable, the left-hand side of the equation above converges to $\partial \gamma(t, \mathbf{x})/\partial t$. On the right-hand side, $\gamma(t' - h_T, \mathbf{x})$ converges to $\gamma(t, \mathbf{x})$ and, with our assumption that $\beta$ is continuous, $\beta(t', \mathbf{x})$ converges to $\beta(t, \mathbf{x})$. In conclusion, we get the equation

$$\frac{\partial \gamma(t, \mathbf{x})}{\partial t} = -\gamma(t, \mathbf{x}) \beta(t, \mathbf{x}).$$

During training, we learn both the $\gamma$ and $\beta$ functions, and we enforce the above differential equation by adding a corresponding term to the loss function. This ensures that the correct relation between $\gamma$ and $\beta$ is preserved during training. Afterwards, having learned these functions successfully, we can use equation (10) to discretize $\beta$ into $\hat{\beta}_T$, and then sample from the forward and the reverse process using the equations derived in Appendix B.

# G IMPLEMENTATION DETAILS

## G.1 METRICS DEFINITION

We use two main metrics to measure our results (Section 4). The first is MAE, defined as:

$$\text{MAE} := \frac{1}{|P|} \sum_{p \in P} |\mathbf{y}_p - \hat{\mathbf{y}}_p|,$$

where $p$ indexes the set of pixel positions $P$ in the images, $\mathbf{y}$ stands for the ground truth image, and $\hat{\mathbf{y}}$ is the predicted image. The second metric is MS-SSIM, which measures the structural similarity between images and is defined as:

$$\text{MS-SSIM}(\mathbf{y}, \hat{\mathbf{y}}) := [l_M(\mathbf{y}, \hat{\mathbf{y}})]^{\alpha_M} \prod_{j=1}^{M} [c_j(\mathbf{y}, \hat{\mathbf{y}})]^{\beta_j} [s_j(\mathbf{y}, \hat{\mathbf{y}})]^{\gamma_j},$$

where $l_j$, $c_j$, and $s_j$ are the measures of luminance, contrast, and structure corresponding to scale $j$. We use five scales for this evaluation and we set $\alpha_j = \beta_j = \gamma_j$ such that $\sum_{j=1}^{M} \gamma_j = 1$.

## G.2 ARCHITECTURAL DETAILS

Following Kingma et al. (2023), to learn the functions $\tau_\theta(t)$ and $\rho_\chi(t)$ we parametrize them using a monotonic neural network. This network is composed of a residual block with three convolutional layers. The first and third layers employ linear activation and are linked by a skip connection, while

the second layer uses a sigmoid activation and is equipped with 1024 filters. All layers adopt $1 \times 1$ convolutions. The decision to use convolutional layers over a dense network stems from the desire to facilitate the model's operation at various resolutions without retraining. Additionally, we constrain the weights of the network to be positive. To satisfy the condition $\beta(0, \mathbf{x}) = \mathbf{0}$, we multiply the network's output by $t$.

For $\lambda_\phi(\mathbf{x})$, we parametrize it using a U-Net architecture (Ronneberger et al., 2015), represented in Figure 3. The model incorporates 5 scales, doubling the number of filters at each scale while concurrently halving the resolution. Each block consists of two sequences: convolution, activation, and instance normalization. Upscaling is achieved using transposed convolutions to ensure differentiability. Softplus activations are used throughout the architecture. Mirrored filters are concatenated, and the output's positivity is guaranteed by a final softplus activation.

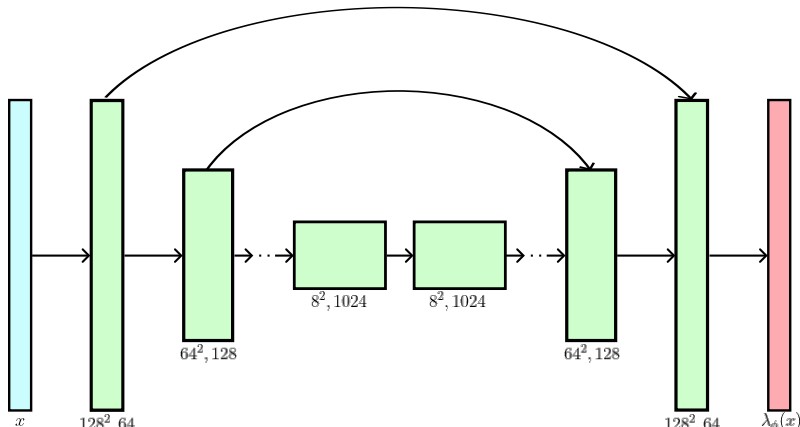

Figure 3: Architecture of $\lambda_\phi(\mathbf{x})$.

Our denoising model (Figure 4) closely mirrors this implementation with two notable distinctions: first, its input comprises the concatenation of $\mathbf{x}$, $\gamma(t, \mathbf{x})$, and $\mathbf{z}_t$, and the predicted $\mathbf{z}_{t-1}$ output has a linear rather than softplus activation. In line with common practices, the network predicts the noise $\epsilon$ at the corresponding timestep. We determine $\mathbf{z}_{t-1}$ by using the algorithm detailed in Appendix H.

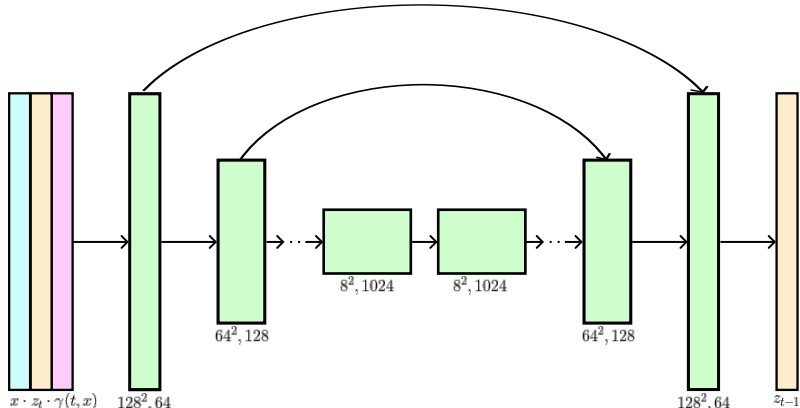

Figure 4: Architecture of the score predictor used in BioSR and QPI.

## H  TRAINING AND INFERENCE ALGORITHMS

Figure 5 presents a high-level overview of the training algorithm and the inference process for CVDM. Algorithm 1 describes the training of the denoiser using a learnable schedule, while Algorithm 2 demonstrates the inference process and how to use the learned schedule during this procedure.

---

**Algorithm 1:** Training of Denoising Model $\hat{\epsilon}_\nu(\mathbf{z}_t(\epsilon), t, \mathbf{x})$

---

**repeat**
$\quad (\mathbf{x}, \mathbf{y}) \sim p(\mathbf{x}, \mathbf{y})$
$\quad t \sim U([0, 1])$
$\quad \epsilon \sim \mathcal{N}(\mathbf{0}, \mathbf{I})$
$\quad$ Take a gradient descent step
$\quad\quad \nabla_\omega \mathcal{L}_{\text{CVDM}}(\gamma, \mathbf{x}, \mathbf{y}, \epsilon)$ where $\omega$ are the parameters of the joint model $\chi, \phi, \theta, \nu$. See equation (6).
**until** converged;

---

**Algorithm 2:** Inference in $T$ timesteps

---

**for** $t = 0$ *to* $T$ **do**
$\quad \beta_t \leftarrow \frac{\beta(t/T, \mathbf{x})}{T}$
$\quad \alpha_t \leftarrow 1 - \beta_t$
$\quad \gamma_t \leftarrow \prod_t \alpha_{<t}$
**for** $t = T$ *to* $1$ **do**
$\quad$ **if** $t = 1$ **then**
$\quad\quad \epsilon \leftarrow 0$
$\quad$ **else**
$\quad\quad \epsilon \sim \mathcal{N}(\mathbf{0}, \mathbf{I})$
$\quad \mathbf{z}_{t-1} \leftarrow \frac{1}{\sqrt{\alpha_t}}\left(\mathbf{z}_t - \frac{\beta_t}{\sqrt{1-\gamma_t}}\hat{\epsilon}_\nu(\mathbf{z}_t, t, \mathbf{x})\right) + \sqrt{\beta_t}\epsilon$

---

Figure 5: Training and Sampling algorithms for CVDM.

# I ADDITIONAL EXPERIMENTAL RESULTS FOR BIOSR AND QPI

The following figures further highlight results from our method in comparison to its counterparts in both the BioSR and QPI benchmarks. The reconstructions in Figures 7 and 8 depict the differences between the CDDPM benchmark and our approach. Similarly, Figure 9 contrasts an F-actin sample as reconstructed by our method and DFCAN.

Additionally, in Figure 10 we provide more examples of QPI evaluated in the synthetic benchmark.

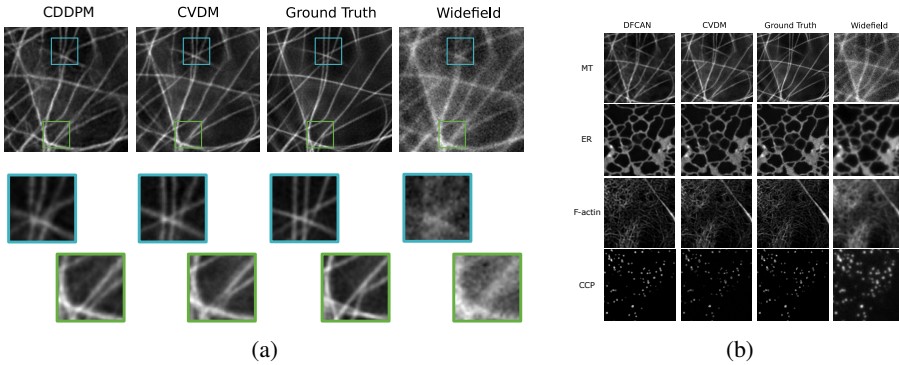

Figure 6: Results from super-resolution methods evaluated in BioSR. (a) Fine-grained comparison for CDDPM and CVDM on a microtubule (MT) sample. (b) Comparison of DFCAN and CVDM over different BioSR structures.

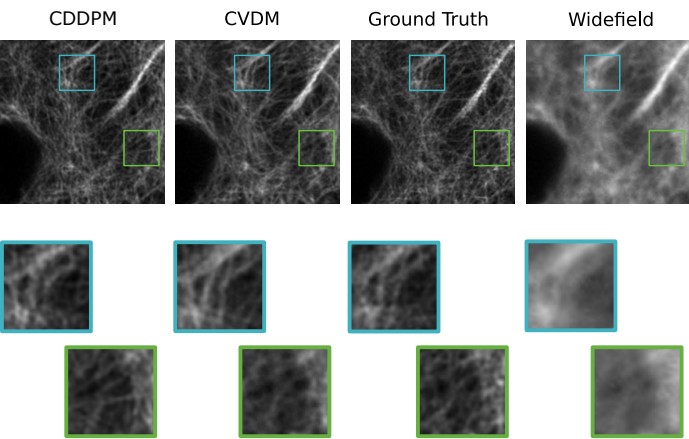

Figure 7: Comparison between a CDDPM reconstruction and our method on a F-actin sample.

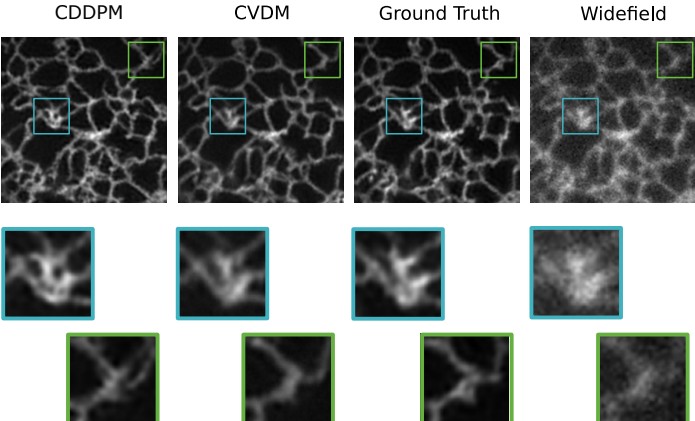

Figure 8: Comparison between a CDDPM reconstruction and our method on a ER sample.

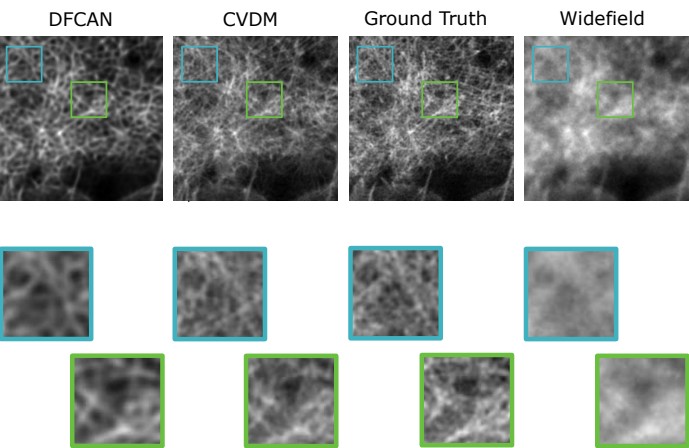

Figure 9: Comparison between a DFCAN reconstruction and our method on a F-actin sample.

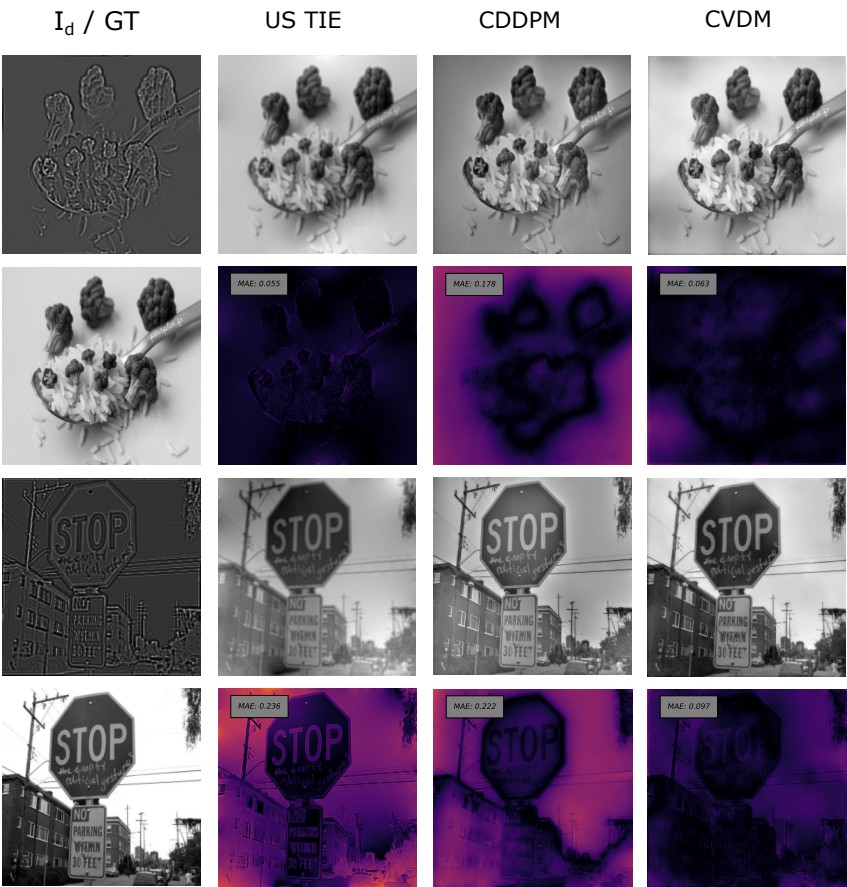

Figure 10: Comparison of Phase Retrieval Methods on simulated HCOCO. From left to right: the first column displays the defocused image at distance $d$ ($I_d$), with the respective ground truth (GT) situated directly below. The second, third and fourth columns represent each a different method, with the reconstruction on top and the error image at the bottom.

## J  THE SCHEDULE AND THE FORWARD PROCESS IN EXPERIMENTS

As described in Section 3.1, the forward process can be characterized by the function $\gamma$:

$$q(\mathbf{z}_t|\mathbf{y}, \mathbf{x}) = \mathcal{N}\left(\sqrt{\gamma(t, \mathbf{x})}\mathbf{y}, \sigma(t, \mathbf{x})\mathbf{I}\right).$$

There is an alternative, equivalent parametrization of the forward process in terms of the function $\beta$. From Section 3.2, equation (4), we know that the following relation holds:

$$\frac{\partial \gamma(t, \mathbf{x})}{\partial t} = -\beta(t, \mathbf{x})\gamma(t, \mathbf{x}).$$

This differential equation can be integrated:

$$-\beta(s, \mathbf{x}) = \frac{1}{\gamma(s, \mathbf{x})}\frac{\partial \gamma(s, \mathbf{x})}{\partial s} = \frac{\partial \log \gamma(s, \mathbf{x})}{\partial s} \implies -\int_0^t \beta(s, \mathbf{x})ds = \log \gamma(t, \mathbf{x}) - \log \gamma(0, \mathbf{x}).$$

As we describe in Section 3, the initial condition $\gamma(0, \mathbf{x}) = 1$ should hold for all $\mathbf{x}$ in a diffusion process. Replacing in the above equation and applying exponentiation, we get

$$\gamma(t, \mathbf{x}) = e^{-\int_0^t \beta(s, \mathbf{x})ds}. \tag{11}$$

Figures 11 and 17 show the averages of $\beta$ and $\gamma$ for different groups of pixels (structure and background) in different images. Notice that the shapes of $\beta$ and $\gamma$ are consistent with equation (11).

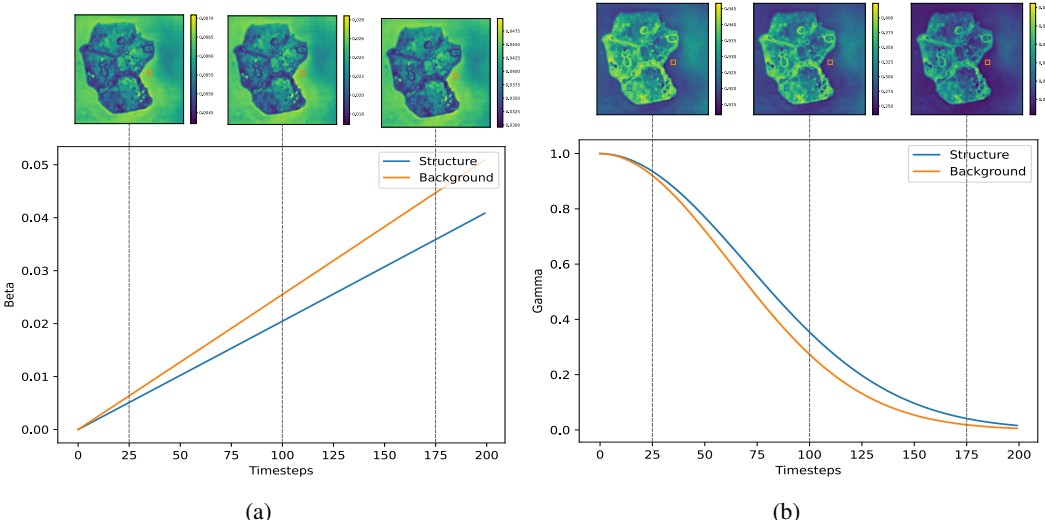

(a)                                                          (b)

Figure 11: Schedule for the represented image from the clinical brightfield dataset. The graph shows the average of the pixels in the respective region (structure and background). (a) Schedule function $\beta$ for this image. (b) Schedule function $\gamma$ for this image.

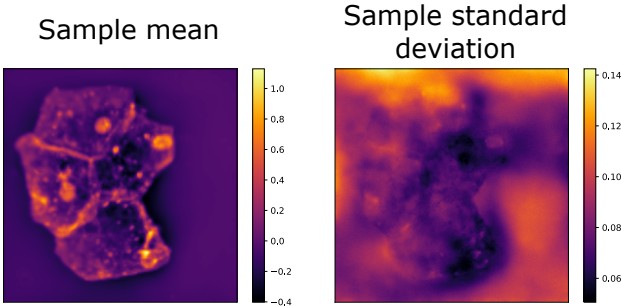

Figure 12: Mean and standard deviations for an image in one of the QPI datasets were reconstructed using 20 samples obtained with CVDM.

Moreover, notice that higher values of $\beta$ lead to values of $\gamma$ that decrease more rapidly to zero, as we would expect. Now, using the variance-preserving condition $\sigma(t, \mathbf{x}) = \mathbf{1} - \gamma(t, \mathbf{x})$, the distribution of the forward process takes the form

$$q(\mathbf{z}_t | \mathbf{y}, \mathbf{x}) = \mathcal{N}\left(e^{-\frac{1}{2} \int_0^t \beta(s, \mathbf{x}) ds} \mathbf{y}, \left(\mathbf{1} - e^{-\int_0^t \beta(s, \mathbf{x}) ds}\right) \mathbf{I}\right).$$

This is the relation we use in Section 5 to analyze the results of the BioSR experiments. We note that it is equivalent to do the analysis in terms of $\gamma$. If $\gamma$ decreases fast, the latent variable $\mathbf{z}_t$ gets rapidly close to a $\mathcal{N}(\mathbf{0}, \mathbf{I})$ distribution. This corresponds to pixels (or parts of the image) that are harder to invert, and is reflected in the variance of those pixels in the reconstructed images. The converse is true for pixels where $\gamma$ decreases more slowly. This relation between the learned schedule and the uncertainty (represented by the variance) is exemplified by Figures 11 and 12, which respectively show the schedule and the variance of the reconstruction for an image.

## K  ABLATION STUDY

First, we study the behavior of our method without the regularization strategy. Figure 13 shows the learned schedule $\gamma(t, \mathbf{x})$ (averaged over all pixels) for BioSR, learned with the same specifications as CVDM but removing the regularization term. As can be observed, under these conditions, the schedule is mostly meaningless. Under this schedule, the input is steeply transformed at the beginning

of the diffusion process, then remains mostly equal for most of the time, and experiences another abrupt change at the end. There is no gradual injection of noise.

As explained in Section 3.4, regularization is important to prevent this type of result. Recall that a variable $\mathbf{z}_t$ sampled from the distribution $q(\mathbf{z}_t|\mathbf{y},\mathbf{x})$ can be written as $\mathbf{z}_t = \sqrt{\gamma(t,\mathbf{x})}\mathbf{y} + \sqrt{\sigma(t,\mathbf{x})}\epsilon$, where $\epsilon$ follows a Gaussian distribution $\mathcal{N}(\mathbf{0},\mathbf{I})$. This reparametrization implies that setting $\gamma \equiv \mathbf{0}$ and $\sigma \equiv \mathbf{1}$ gives $\mathbf{z}_t = \epsilon$, allowing the noise prediction model $\hat{\epsilon}_\nu(\mathbf{z}_t(\epsilon),t,\mathbf{x})$ to perfectly predict $\epsilon$ and yield $\hat{\mathcal{L}}_\infty = 0$. It is true that $\gamma \equiv \mathbf{0}$ conflicts with the $\mathcal{L}_\beta$ loss term, but any function $\gamma$ that starts at $\mathbf{1}$ for $t = 0$ and then abruptly decreases to $\mathbf{0}$ is admissible. In Figure 13 we can see a similar behavior, with $\gamma$ starting at $\mathbf{1}$ and then abruptly dropping to a low value. The regularization term $\mathcal{L}_\gamma$ prevents undesired solutions of this type and ensures the gradual nature of the diffusion process.

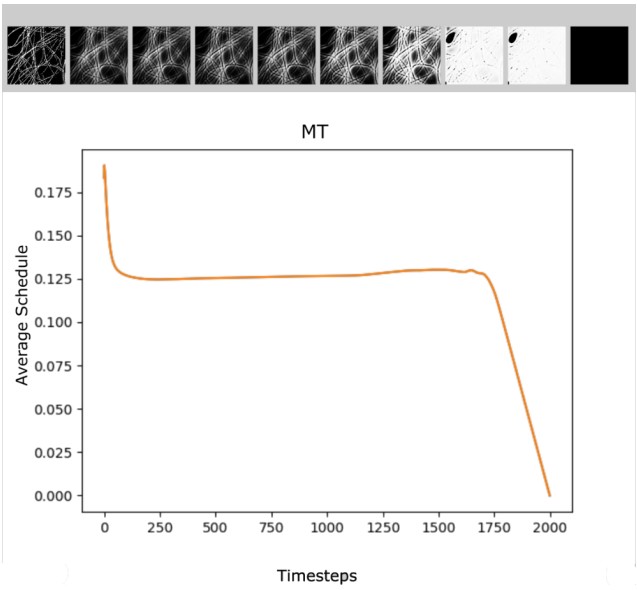

Figure 13: Value of $\gamma(t,\mathbf{x})$ for each timestep, averaged over all pixels.

Additionally, we perform an ablation study on the impact of learning a pixel-wise schedule instead of a single, global one. Learning a single schedule can be implemented as a special case of our method, which we denote as CVDM-simple. For the experiment we use the synthetic QPI dataset, introduced in Section 4. Table 3 summarizes the results. As can be seen, CVDM-simple shows improved performance when compared to CDDPM, supporting the idea that learning the schedule can yield better results than fine-tuning it. At the same time, the results are not on par with CVDM, showing there is value in learning a pixel-wise schedule. This supports the idea that some regions of the image are intrinsically more difficult to reconstruct than others, and the schedule captures part of that difficulty. Figure 14 allows for visual comparison of the reconstructions given by these methods.

| Metric / Model | CDDPM | CVDM-simple | CVDM |
|---|---|---|---|
| MS-SSIM | 0.881 | 0.915 | 0.943 |
| MAE | 0.134 | 0.094 | 0.073 |

Table 3: Comparison of diffusion methods on synthetic HCOCO.

## L    EXPERIMENTAL RESULTS FOR IMAGE SUPER-RESOLUTION

We showcase the versatility of our method by training the model described in Saharia et al. (2021) enhanced with our schedule learning mechanism, for the task of image super-resolution in ImageNet.

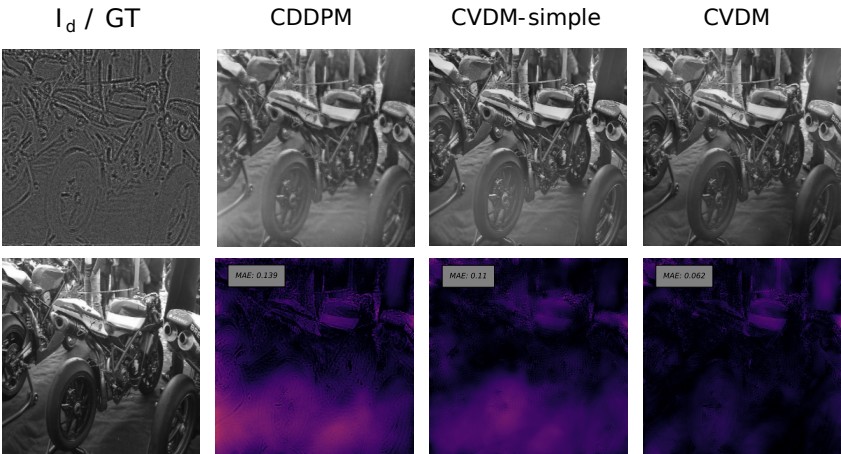

Figure 14: Reconstructions by different diffusion methods on a synthetic HCOCO sample. The first column displays the defocused image from left to right at a distance $d$ ($I_d$), with the ground truth (GT) directly below. The second, third, and fourth columns represent each a different method, with the reconstruction on top and the error image at the bottom.

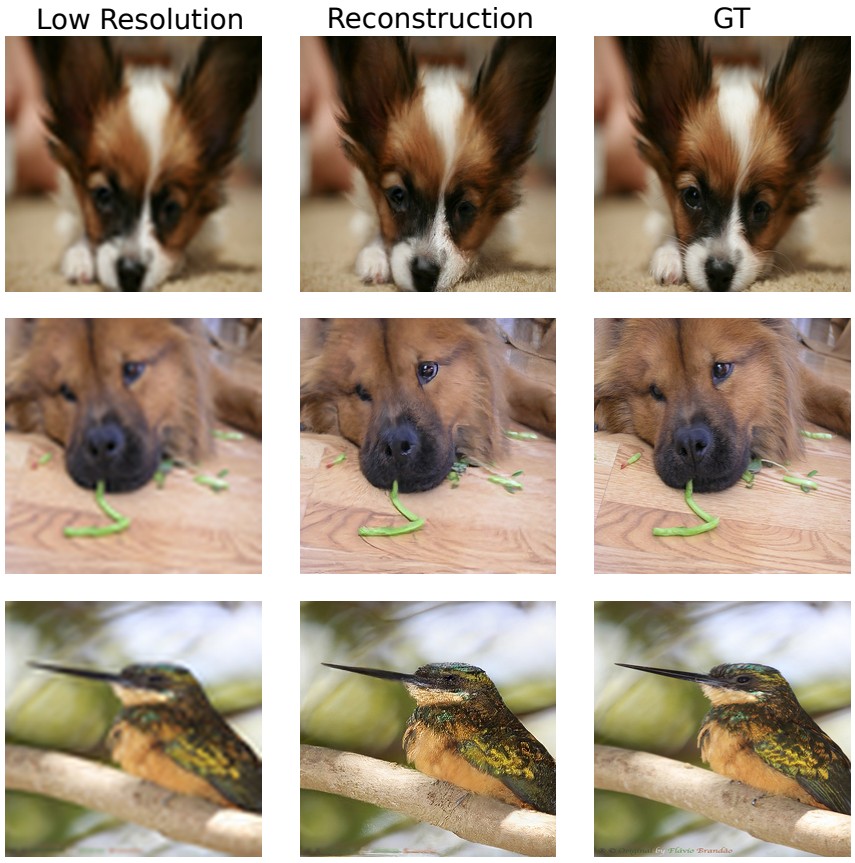

Figure 15: Super-resolution reconstructions in ImageNet using CVDM (our method).

For this task, we compare our method to SR3 (Saharia et al., 2021) and Denoising Diffusion Restoration Models or DDRMs (Kawar et al., 2022). The images are sampled with $T = 500$ timesteps, for our method. For SR3 and DDRM, the metrics are taken from their respective works.

Table 4 shows the results obtained by the three methods, using the peak signal-to-noise ratio (PSNR) and structural similarity index measure (SSIM) metrics. The results obtained by our method are comparable to both SR3 and DDRM. Similarly to the other applications described in Section 4, our method achieves competitive results without requiring any fine-tuning of the schedule. Figure 15 shows three examples from the ImageNet dataset, including the low-resolution image, the high-resolution ground truth, and a reconstruction sampled with our method.

| Method | PSNR | SSIM |
|---|---|---|
| CVDM | 26.38 | 0.76 |
| SR3 | 26.40 | 0.76 |
| DDRM | 26.55 | 0.74 |

Table 4: Comparison between methods for image super-resolution.

## M UNCERTAINTY ESTIMATION AND THE SCHEDULE

Our method implicitly learns a conditional probability distribution, from which conditioned sampling can be performed. This means that we can get different reconstructions for a given input, drawing attention to the question of how much uncertainty is there in the reconstruction. As a way of measuring uncertainty, several reconstructions can be sampled for a given input, and the element-wise (e.g., pixel-wise) sample variance can be computed. Theoretically, we expect the schedule function $\beta$ to be significantly linked to this sample variance. As described in Section 5, in continuous time, the reverse diffusion process can be characterized by a stochastic differential equation with a diffusion coefficient of $\sqrt{\beta(t, \mathbf{x})}$. In that sense, higher values of $\beta$ make the reverse process more diffusive and leads to more randomness in the reconstructions.

We illustrate these ideas using samples from both the BioSR and ImageNet datasets, corresponding respectively to the problems of super-resolution microscopy and image super-resolution. The top half of Figure 16 shows a widefield microscopy image from the BioSR dataset, along with the ground truth image and five sample reconstructions. The bottom half shows a low-resolution image from the ImageNet dataset, along with the ground truth super-resolution image and five sample reconstructions.

For each reconstruction, we include the absolute error with respect to the ground truth image, which can be seen at the bottom of the respective half. Also at the bottom, we include the sample variance over the five reconstructions for each pixel, besides an image showcasing the pixel-wise magnitude of the $\beta$ schedule function (as an integral with respect to $t$). Finally, we highlight zones of high sample variance, so that it is possible to visually appreciate how the reconstructions differ from each other.

From Figure 16 it is easy to visualize the ideas described at the beginning of this Section. The sample variance is reflected in the reconstructions, which clearly vary in some of the details. There is also a clear relation between the magnitude of $\beta$ and the sample variance, as expected theoretically. This is interesting, as it suggests that $\beta$ could be used to assess uncertainty in the absence of a readily available ground truth image, which would be the case in most real-world applications. We can also observe that the regions with high sample variance tend to exhibit a higher absolute error in the reconstruction. In that sense, the regions with larger uncertainty (as measured by the sample variance or the magnitude of $\beta$) correspond to details that are truly more difficult to reconstruct correctly.

In general, this highlights the importance of robust uncertainty estimation, even for reconstructions that look generally correct. For applications such as microscopy, a single pixel may contain relevant information, such as evidence of cell interaction. Something similar happens with MRI, where a few pixels can determine, for instance, the difference between the presence or absence of a tumor. Therefore, hallucinations in the reconstruction method can be easily misleading, which is in fact one of the challenges for the adoption of accelerated MRI in clinical environments (Bhadra et al., 2021).

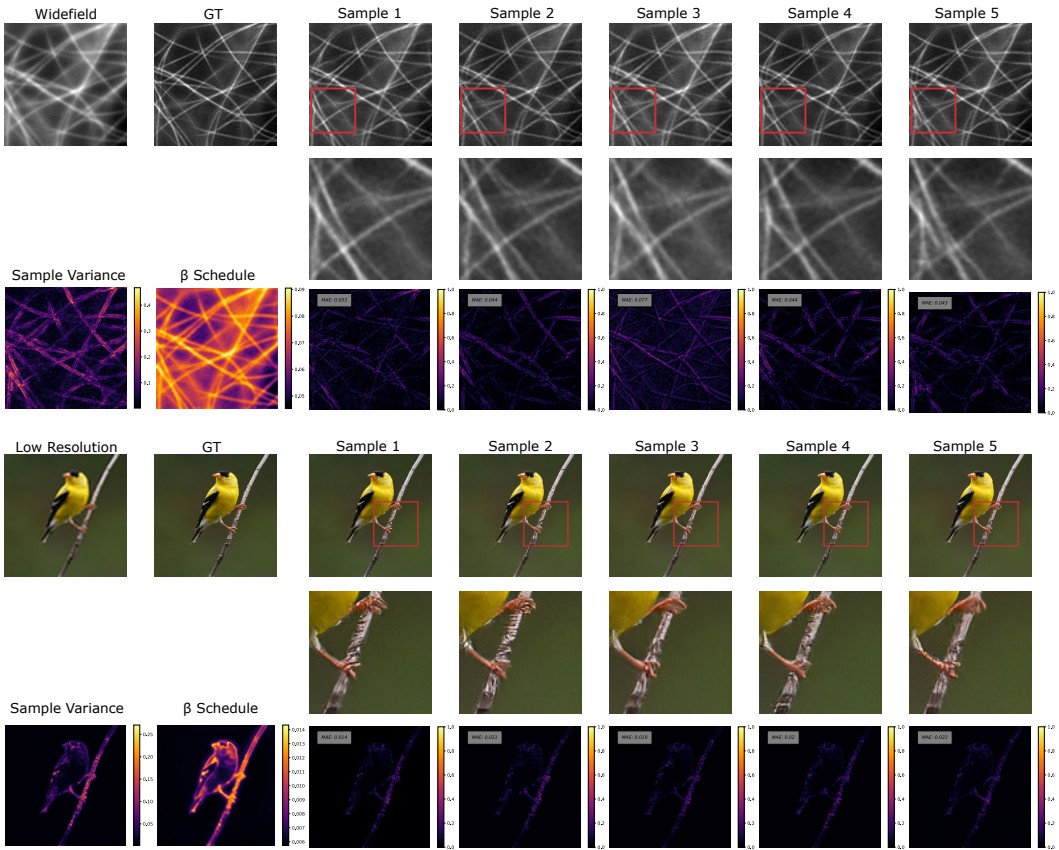

Figure 16: Input and ground truth images, along with five reconstructions, for samples in BioSR and ImageNet. For every reconstruction, absolute error with respect to the ground truth is included, and regions of high uncertainty (i.e., sample variance) are enlarged so that differences in the details can be appreciated. Pixel-wise sample variance over the five reconstructions is also shown, along with the pixel-wise intensity of the $\beta$ schedule function.

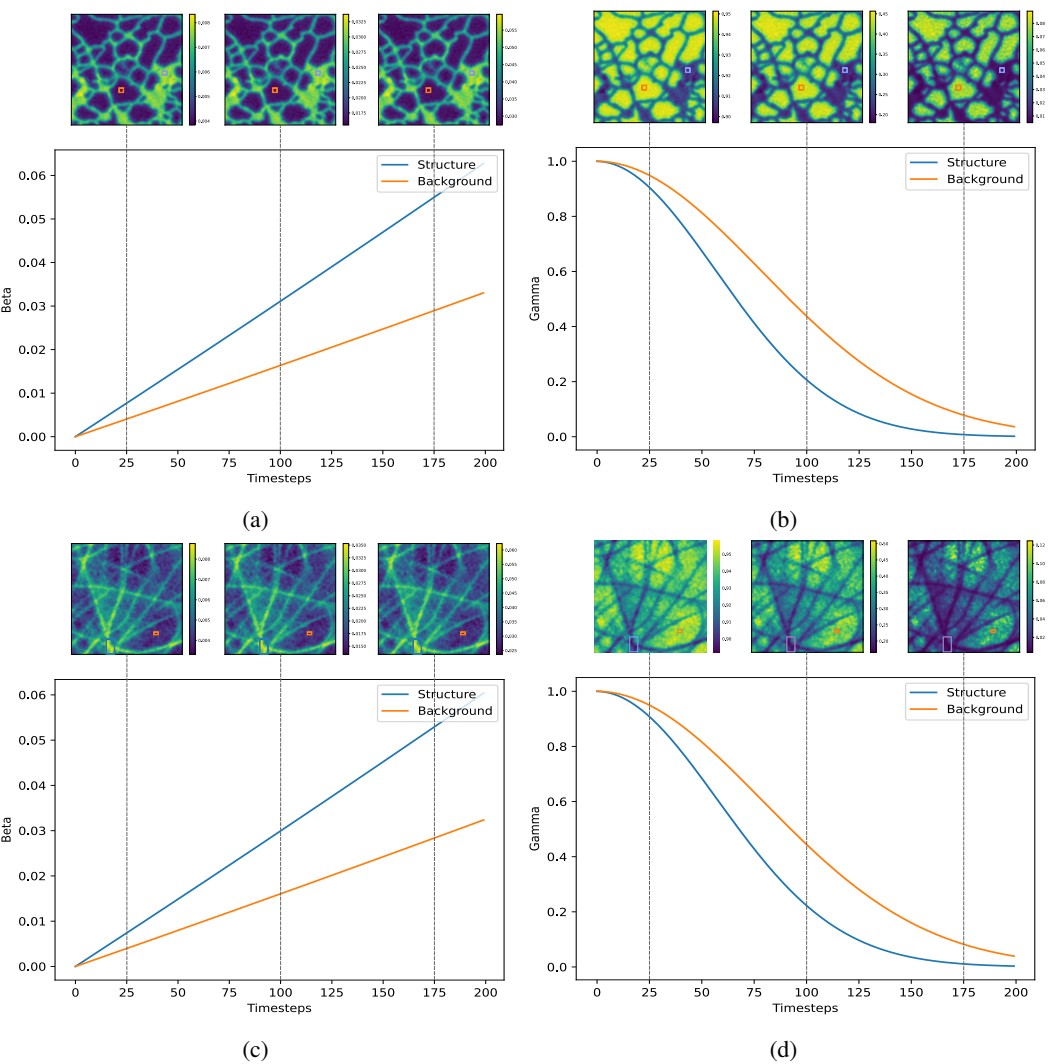

Figure 17: Schedule for the represented images from the BioSR dataset. The graph shows the average of the pixels in the respective region (structure and background). (a) Schedule function $\beta$ for an ER image. (b) Schedule function $\gamma$ for the same ER image. (c) Schedule function $\beta$ for an MT image. (d) Schedule function $\gamma$ for the same MT image.

