# OpenReview forum: "Conditional Variational Diffusion Models"
_ICLR.cc/2024/Conference — ICLR 2024 poster_

### Official Review · Reviewer_gCsZ · 2023-10-20

**Soundness:** 3 good
**Presentation:** 2 fair
**Contribution:** 2 fair
**Rating:** 5
**Confidence:** 4

**Summary:**

The paper presents an extension of variational diffusion models to the conditional setting. In particular, the paper makes the following novel contributions:
1. An element-wise learned variance schedule is used.
2. The learned variance schedule is factorized wrt. time and the conditioning variable.
3. The continuous time shedule is formulated in a different way than in Kingma et al. and new loss terms enforcing constraints is introduced.
The method is demonstrated on two image processing problems.

**Strengths:**

Developing variational diffusion models and making them practically useful is an important and timely problem.

The proposed methods appear sound and reasonable, and seem to work well in practice.

The paper includes supplementary material where ample technical details are given, allowing the reader to follow each step of the derivations.

The paper presents several novel contributions beyond the existing literature.

**Weaknesses:**

In the abstract, learning the variance schedule is stated as a main contribution, however it is not clear from the abstract in which way the proposed method differs from existing methods that learn the variance schedule.

In general, it is not clearly stated what the contributions are beyond previous work, particularly Kingma et al. 2023. It would be a strong improvement to have a clear list of technical contributions in the beginning of the paper, allowing the reader to have an overview from the beginning.

Before eq. 1 it is mentioned that a continuous time diffusion is used, however, the manuscripts proceeds with discrete time steps. Since in the end, the continuous time formulation is used, I wonder if the presentation could be more direct, building on the continuous time formulation in Kingma et al. 2023?

The technical novelty of the paper is fairly limited, but while the novel contributions might be minor, it could still be practically important. However, it is not absolutely clear from the paper, to what extent these contributions are important for performance. An ablation study or a direct comparison with Kingma et al. 2023 could have illuminated this more direcly.

**Questions:**

The introduction begins with a relatively high level discussion of inverse problems, which in my view is a bit removed from the specific contributions of the paper. Maybe it could be an idea to motivate the paper more directly by mentioning specific inverse problems that the method is suitable for, such as the super resolution microscopy problem?

"To achieve this, we adopt the framework proposed by Saharia et al. (2021), which focuses on training with the statistics of the noise at each timestep rather than direct timesteps, thus allowing flexible use of the model during inference." Could you clarify what this means?

In the final loss, you drop the term SNR'/SNR. A large portion of the paper is dedicated to the design of the variance schedule mechanism. Would it be possible to include some more discussion / insights into why this term can reasonably be dropped? I realize this if done in many other papers as well, but usually without much discussion.

---

> ### Author Response · Authors · 2023-11-14
> **Thanks for your detailed review. We have addressed your concerns in our response and in the text of the paper.**
>
> # Official Comment for Reviewer gCsZ
>
> Thank you for taking the time to review our work. We appreciate your detailed suggestions and criticisms, which we think will help us improve our paper. Please find below detailed clarification to your comments and questions. We have also included this material in the new version of the paper, highlighting the text in blue to make it easier to find.
>
> ## Weakness #1
>
> > *In the abstract, learning the variance schedule is stated as a main contribution, however, it is not clear from the abstract in which way the proposed method differs from existing methods that learn the variance schedule.*
> >
> > *In general, it is not clearly stated what the contributions are beyond previous work, particularly Kingma et al. 2023. It would be a strong improvement to have a clear list of technical contributions in the beginning of the paper, allowing the reader to have an overview from the beginning.*
>
> Thanks for pointing out this issue. We understand your concerns and agree with your suggestion. We have included a bullet list in the Introduction (Section 1) explicitly stating the contributions of our work. Also, in Related Work (Section 2), we have included a summary of Variational Diffusion Models (VDMs, Kingma et al. 2023) and explained some of the differences with our method. Moreover, throughout the paper, we have extended the discussions about our technical contributions and their importance for the success of the method. See especially Sections 3.1, 3.2 and 3.4. We also include here a relevant summary.
>
> VDMs (Kingma et al. 2023) constitute the main framework described in the literature for learning the variance schedule.  VDMs formulate a diffusion process for unconditioned distribution sampling, where the latent variables are indexed in continuous time, and the schedule is a learnable function that must satisfy a few minimal conditions. The parameters of the schedule are defined as a monotonic network, and the model is trained by minimizing a weighted version of the noise prediction loss. Their work also uses Fourier features to improve the prediction of high-frequency details. We introduce the Conditional Variational Diffusion Model (CVDM), which includes important generalizations and novelties with respect to VDMs:
>
> - Similarly to Kingma et al. (2023), we learn the schedule as part of the training. However, we extend their approach to the conditioned case. This turns out to be a highly non-trivial task, as detailed below. We also allow for learning a different schedule for each element in the output (e.g., a pixel-wise schedule for images). These extensions require several technical novelties, including a separation-of-variables approach when defining the schedule (details in Section 3.2).
> - We prove that the rate of convergence of the discrete-time diffusion loss to the continuous-time case depends strongly on the schedule functions. Kingma et al. (2023) show that diffusion models are invariant to the schedule selection in the continuous case. Nevertheless, any discretization used for model sampling will introduce the dependency shown in Appendix E.2 on the schedule's derivatives. Our finding motivates the introduction of a novel regularization term that proves to be critical for the performance of our method (details in Section 3.4).
> - We replace the architecture in Kingma et al. (2023) with two networks, one required to be positive (for the conditioning variable) and one monotonic-convolutional network (for the time variable). This allows us to test our model with inputs of different resolutions without retraining. Moreover, by incorporating some schedule properties in the loss function, our method does not need the post-processing of the schedule that Kingma et al. 2023 uses, nor the preprocessing of the input. This results in a cleaner and more straightforward method.
>
> Our approach results in a streamlined implementation that requires minimum fine-tuning for new domains of application, and can be readily used for conditioned problems.

---

> ### Author Response · Authors · 2023-11-14
> **We continue our response to your concerns and questions.**
>
> ## Weakness #2
>
> > *Before eq. 1 it is mentioned that a continuous time diffusion is used, however, the manuscripts proceeds with discrete time steps. Since in the end, the continuous time formulation is used, I wonder if the presentation could be more direct, building on the continuous time formulation in Kingma et al. 2023?*
>
> We appreciate your insight. We considered using a continuous-time formulation in the spirit of Kingma et al. 2023. In some ways, we still follow a continuous-time formulation. For instance, the schedule is represented by functions instead of finite sequences (in contrast to Ho et al. 2020 for instance). That being said, the forward process and its posterior are indeed first introduced using a time discretization. We chose this formulation for a few reasons:
>
> 1. We find it simpler for our case. In particular, given latent variables $\mathbf{z}\_s$ and $\mathbf{z}\_t$, Kingma et al. 2023 provide expressions for the forward process and the posterior distribution, for **any** values of $t,s$. In our (conditioned) case, that would correspond to $q(\mathbf{z}\_t|\mathbf{x}\_s,\mathbf{x})$ and $q(\mathbf{z}\_t|\mathbf{z}\_s,\mathbf{y},\mathbf{x})$ for any $t,s$. We don't need the full generality of these expressions for training or inference, so it is unnecessary to introduce them. The inference is performed for a time discretization anyway, so we find it simpler and more natural to introduce the diffusion process in that context.
> 2. Our formulation allows us to derive equation (4) from more elementary ideas, without relying on stochastic calculus.
> 3. As we highlight in Section 3.4, a result that holds for continuous time may not hold once time is discretized (which is the case for any practical implementation). In that sense, we prefer to introduce the diffusion process in discrete time, and formulate the continuous-time case as the limit when $T \rightarrow\infty$. Then, we can analyze how different properties change from one setting to the other.
>
> We agree that our introduction to the formulation sounds confusing before equation (1). We have rewritten that part of Section 3.1 to better represent our approach.
>
>
> ## Weakness #3
>
> > *The technical novelty of the paper is fairly limited, but while the novel contributions might be minor, it could still be practically important. However, it is not absolutely clear from the paper, to what extent these contributions are important for performance. An ablation study or a direct comparison with Kingma et al. 2023 could have illuminated this more direcly.*
>
> Thanks for this suggestion. We have clarified the role played by our specific contributions in the new version of the paper (see for instance Sections 3.2, 3.4 and 4.3). We are also performing an ablation study. Some important points:
>
> 1. The regularization term described in Section 3.4 is critical to keep the schedule from converging to an uninformative solution that trivially minimizes the diffusion loss. In this regard, this technical novelty is essential for the method. See details in Section 3.4.
> 2. It is non-trivial how to properly condition the schedule on the data $\mathbf{x}$, because the schedule function $\gamma$ needs to be monotonic in $t$, but not necessarily on $\mathbf{x}$. Our factorization of the function $\beta$ is in our opinion a clever way of solving this problem and is also crucial for our method. See details in Section 3.2.
> 3. We performed an ablation study on the effect of having a global unique schedule instead of a pixel-wise dedicated schedule. In our ablation study (Appendix K), we compare global vs. pixel-wise schedules (CVDM-simple). The latter outperforms CDDPM but does not match CVDM's results, highlighting the value of pixel-wise schedules in addressing unique image challenges. We also include an ablation study on the impact of our regularization strategy, which shows the importance of regularized learning for CVDM.
>
> A direct comparison with Kingma et al. 2023 is not really possible because their method is for unconditioned sampling, and extending the idea to the conditioned case is precisely one of the non-trivial contributions of our work, as explained above.
>
> In general, we think our approach significantly simplifies the use of variational diffusion models, especially for the conditioned case. As mentioned in your list of Strengths, this is important and timely. Our own experience running experiments was that our method is very straightforward to apply and exhibits good results and very robust convergence. In light of this, we think our technical contributions are critical for the success of the method.

---

> ### Author Response · Authors · 2023-11-14
> **We continue our response to your concerns and questions.**
>
> ## Questions
>
> > *The introduction begins with a relatively high level discussion of inverse problems, which in my view is a bit removed from the specific contributions of the paper. Maybe it could be an idea to motivate the paper more directly by mentioning specific inverse problems that the method is suitable for, such as the super resolution microscopy problem?*
>
> Thanks for the suggestion. Indeed, that would be a better way of motivating our work in the context of a computer science conference and given the contents of the paper. We have added image super-resolution as a motivating problem in the Introduction.
>
>
>
> > *"To achieve this, we adopt the framework proposed by Saharia et al. (2021), which focuses on training with the statistics of the noise at each timestep rather than direct timesteps, thus allowing flexible use of the model during inference." Could you clarify what this means?*
>
> Thanks for pointing this out. We have tried to clarify the text at the end of Section 2. Basically, Saharia et al. 2021 define a noise prediction model that takes a different input than the one in Ho et al. 2020, and we have used their approach.
>
>
>
> > *In the final loss, you drop the term SNR'/SNR. A large portion of the paper is dedicated to the design of the variance schedule mechanism. Would it be possible to include some more discussion / insights into why this term can reasonably be dropped? I realize this if done in many other papers as well, but usually without much discussion.*
>
> Thanks for the question. We can clarify this. In works that set the schedule as a hyperparameter (i.e Ho et al. 2020), we think this type of strategy is adopted mainly because it yields better results in experiments and maybe because some of the fractions involved can be numerically undesirable. Our case is a bit different. Since we are learning the schedule, there is a new consideration. Basically, during training, the schedule can converge to a solution such that $\text{SNR}'(t,\mathbf{x})\approx 0$ for all $t,\mathbf{x}$, and the diffusion loss is trivially minimized without learning anything useful. We have included this explanation in Appendix E.3, where the final form of the diffusion loss is derived.
>
>
>
> ### References
>
> Jonathan Ho, Ajay Jain, and Pieter Abbeel. Denoising Diffusion Probabilistic Models, December 2020. URL `http://arxiv.org/abs/2006.11239`. arXiv:2006.11239 [cs, stat].
>
> Diederik P. Kingma, Tim Salimans, Ben Poole, and Jonathan Ho. Variational Diffusion Models, April 2023. URL `http://arxiv.org/abs/2107.00630`. arXiv:2107.00630 [cs, stat].
>
> Chitwan Saharia, Jonathan Ho, William Chan, Tim Salimans, David J. Fleet, and Mohammad Norouzi. Image Super-Resolution via Iterative Refinement, June 2021. URL `http://arxiv.org/abs/2104.07636`. arXiv:2104.07636 [cs, eess].

---

> ### Author Response · Authors · 2023-11-23
>
> Thank you for all your comments. We were wondering if you had additional questions that we could answer for you?

---

### Official Review · Reviewer_jHYB · 2023-10-31

**Soundness:** 3 good
**Presentation:** 3 good
**Contribution:** 4 excellent
**Rating:** 8
**Confidence:** 4

**Summary:**

This papers proposes a learned scheduling approach for training conditioned diffusion model for solving imaging inverse problems. In effect, the paper takes the variational diffusion model developed by Kingma et al. and replaces its learned scheduling algorithm with one that is conditioned on observations. The paper further extends this approach by learning per-pixel variances. The proposed method is applied to super-resolution microscopy and quantitative phase imaging; it outperforms existing diffusion-based approaches in both contexts.

**Strengths:**

The proposed method is effective and, though relatively straightforward, novel.

Validation on real-world data is highly valuable and effectively demonstrates the utility of the proposed method.

The proposed method is general-purpose.

**Weaknesses:**

The paper could do a clearer job differentating itself from VDM. A bullet-pointed list of contributions would have been appreciated.

**Questions:**

## Minor comment
The typical convention in imaging inverse problems is y=A(x), rather than x=A(y).

---

> ### Author Response · Authors · 2023-11-14
> **We appreciate your review. We have commented on your questions in this response and in the text of the paper.**
>
> # Official Comment for Reviewer jHYB
>
> Thank you for taking the time to review our work, and for noticing the novel aspects of our method and its practical usefulness. Below, we clarify the specific concerns mentioned in the review. Also, note that we have included these ideas in the text of the new version of the paper, highlighting the changes in blue so that they are easy to spot.
>
> ## Weaknesses
>
> > *The paper could do a clearer job differentating itself from VDM. A bullet-pointed list of contributions would have been appreciated.*
>
> Thanks for pointing this out and for suggesting the use of a bullet list. We have answered this specific concern for other reviews too, but we also include an answer here to keep reviews and relevant comments organized.
>
> We agree with your suggestion. We have included a bullet list in the Introduction (Section 1) explicitly stating the contributions of our work. Also, in Related Work (Section 2), we have included a summary of Variational Diffusion Models (VDMs, Kingma et al. 2023) and explained some of the differences with our method. Moreover, throughout the paper, we have extended the discussions about our technical contributions and their importance for the success of the method. See especially Sections 3.1, 3.2 and 3.4. We also include here a relevant summary.
>
> VDMs (Kingma et al. 2023) constitute the main framework described in the literature for learning the variance schedule.  VDMs formulate a diffusion process for unconditioned distribution sampling, where the latent variables are indexed in continuous time, and the schedule is a learnable function that must satisfy a few minimal conditions. The parameters of the schedule are defined as a monotonic network, and the model is trained by minimizing a weighted version of the noise prediction loss. Their work also uses Fourier features to improve the prediction of high-frequency details. We introduce the Conditional Variational Diffusion Model (CVDM), which includes important generalizations and novelties with respect to VDMs:
>
> - Similarly to Kingma et al. (2023), we learn the schedule as part of the training. However, we extend their approach to the conditioned case. This turns out to be a highly non-trivial task, as detailed below. We also allow for learning a different schedule for each element in the output (e.g., a pixel-wise schedule for images). These extensions require several technical novelties, including a separation-of-variables approach when defining the schedule (details in Section 3.2).
> - We prove that the rate of convergence of the discrete-time diffusion loss to the continuous-time case depends strongly on the schedule functions. Kingma et al. (2023) show that diffusion models are invariant to the schedule selection in the continuous case. Nevertheless, any discretization used for model sampling will introduce the dependency shown in Appendix E.2 on the schedule's derivatives. Our finding motivates the introduction of a novel regularization term that proves to be critical for the performance of our method (details in Section 3.4).
> - We replace the architecture in Kingma et al. (2023) with two networks, one required to be positive (for the conditioning variable) and one monotonic-convolutional network (for the time variable). This allows us to test our model with inputs of different resolutions without retraining. Moreover, by incorporating some schedule properties in the loss function, our method does not need the post-processing of the schedule that Kingma et al. 2023 uses, nor the preprocessing of the input. This results in a cleaner and more straightforward method.
>
> ## Questions
>
> #### Minor comment
>
> > *The typical convention in imaging inverse problems is y=A(x), rather than x=A(y).*
>
> Thanks for pointing this out. We tried to present our method in the context of general inverse problems, so we used the $A(\mathbf{y}) = \mathbf{x}$ notation throughout. It is true that all the applications presented involve images, so the notation you suggest could have been used. However, we think it is not worth it to change it now. It could end up being more confusing for the reviewers, who already read the paper with the original notation.
>
> ### References
>
> Diederik P. Kingma, Tim Salimans, Ben Poole, and Jonathan Ho. Variational Diffusion Models, April 2023. URL `http://arxiv.org/abs/2107.00630`. arXiv:2107.00630 [cs, stat].

---

### Official Review · Reviewer_UQRa · 2023-10-31

**Soundness:** 2 fair
**Presentation:** 2 fair
**Contribution:** 2 fair
**Rating:** 8
**Confidence:** 3

**Summary:**

The authors aim to solve inverse problems, i.e. recovering an underlying true object from imperfect measurements. The authors propose a conditional variational diffusion model (CVDM) to learn the conditional distribution of the true object given the measurement while being able to report uncertainty unlike typical supervised deep learning methods for inverse problems. CVDMs extend *unconditional* variational diffusion models (VDM; Kingma et al., 2023) to the *conditional* case in order to solve inverse problems and avoid hand-tuning the variance schedule of the diffusion process.

The authors demonstrate their model by:
1. showing competitive performance on the BioSR dataset for super-resolution in microscopy
2. showing superior performance on quantitative phase imaging (phase retrieval) on both the HCOCO dataset and their own clinical dataset.

**Strengths:**

The authors present an extension of VDMs to the conditional case, which has not yet been demonstrated and would indeed eliminate some hand-tuning when training conditional diffusion models. This is also well motivated by an interesting application to inverse problems in optics where uncertainty estimates would be very useful. The authors demonstrate that their learned schedules correspond well to more structured regions of their input on the super-resolution task.

**Weaknesses:**

While the authors motivate their work well, ~I have some concerns with the experiments performed and their presentation. I would be willing to raise my score if the following concerns are addressed.~

**Update**: The authors have addressed the major concerns, so I have updated my score accordingly.

**Major concerns**:

**Uncertainty results**: One of the main motivations of the paper is the ability to report uncertainty in order to point out artifacts in the solutions. In Figure 2, the authors show error images between the reconstructions and the ground truth. Do the uncertainty estimates correspond well to the regions of the image with more error?

**Performance on other datasets**: The performance is not significantly better than other methods for the BioSR super-resolution dataset. If one of the major goals is to demonstrate the flexibility and robustness of this approach, it would be nice to see comparisons against more conditional generative model datasets, especially those in the CDDPM paper (though I understand that there is a focus on inverse problems).

**Flexibility of super-resolution results**: When applied to super-resolution, this method does not require information about the PSF. This means that the model is learning the PSF implicitly from the data, but also means that the model would need to be retrained in order to get accurate results on data from a new PSF, which might be impractical if too much data is required for training. Again, the flexibility and robustness of this method is highlighted, so it would be nice to see some experiments or discussion addressing this, e.g. comparison against other blind deconvolution methods on datasets with different PSFs.

**Estimating pixelwise schedule**: The schedules demonstrated for structured versus background regions appear to be linear with different slopes for the different regions. Would other/baseline methods perform better using an estimated linear pixelwise schedule with a higher slope on "structured" regions?

**Minor concerns**:

**Fourier features**: The original VDM paper showed that a significant amount of the performance was due to the use of Fourier features. It would be helpful to know whether anything like that is the case here.

**Differences from VDM**: It would be helpful to have an explicit list or section summarizing the differences/extensions from VDM.

**Figure design**: Many of the figures are very small and laid out in a way that makes comparing images and reading text difficult, e.g. in Figure 3 it is difficult to see the sections from which the schedules are being estimated.

**Questions:**

In addition to the previous concerns, I would appreciate if the authors could clarify the following:

1. Does the pattern of schedules for structured versus background regions shown in Figure 2 hold true across the whole sample and for all samples?
1. Are there significant differences in the inference time of this model versus the methods being compared against?
1. Is the "US" in "US-TIE" for universal solution? Only TIE is specified in the text.

---

> ### Author Response · Authors · 2023-11-17
> **Thanks for your detailed review. We have addressed your concerns in our response and in the text of the paper.**
>
> # Official Comment for Reviewer UQRa
>
> Thanks for your detailed assessment of our work and your helpful criticism and suggestions. We have made every effort to include all the constructive feedback we received, and the new revision of the paper reflects that. We have made a number of changes, improving clarity of exposition and including new discussions that were absent before. We have also conducted new experiments and performed an ablation study. In the new version of the paper, changes and new content are in blue text to make them easier to find. Please find below our response to your concerns and questions.
>
> ## Weaknesses
>
> > While the authors motivate their work well, I have some concerns with the experiments performed and their presentation. I would be willing to raise my score if the following concerns are addressed.
>
> ### Major concerns
>
> > **Uncertainty results:** One of the main motivations of the paper is the ability to report uncertainty in order to point out artifacts in the solutions. In Figure 2, the authors show error images between the reconstructions and the ground truth. Do the uncertainty estimates correspond well to the regions of the image with more error?
>
> Thanks for the question. It is indeed relevant to explore the connection between uncertainty and reconstruction error. As a general matter, the answer to your question is that, yes, regions with higher uncertainty exhibit larger reconstruction errors. In the new Appendix M, you can find extended material about uncertainty. We discuss the relationship between uncertainty estimates and the schedule and offer a clear illustration of the correlation between uncertainty and reconstruction error. We have also improved the discussion about uncertainty in Section 5. In summary, even though uncertainty estimation is not the primary aspect of the paper, we agree it is important, so we have extended and improved the parts related to this topic, including a discussion of the point you raise here.
>
> > **Performance on other datasets:** The performance is not significantly better than other methods for the BioSR super-resolution dataset. If one of the major goals is to demonstrate the flexibility and robustness of this approach, it would be nice to see comparisons against more conditional generative model datasets, especially those in the CDDPM paper (though I understand that there is a focus on inverse problems).
>
> Thanks for the suggestion. We agree that the method's robustness is emphasized in the paper as a feature of CVDM, and more experiments for conditioned sampling would further support this claim. To better showcase the flexibility of our method, we have conducted additional experiments. In particular, we have used our method for an image super-resolution task using ImageNet 1K (Deng et al. 2009). We compare our results against two diffusion-based methods, SR3 (Saharia et al. 2021) and DDRM (Kawar et al. 2022), using two different metrics (SSIM and PSNR). Using a learned schedule (and no fine-tuning), our method yields comparable results to these two methods, providing further evidence of the flexibility and robustness of CVDM.
>
> > **Flexibility of super-resolution results:** When applied to super-resolution, this method does not require information about the PSF. This means that the model is learning the PSF implicitly from the data, but also means that the model would need to be retrained in order to get accurate results on data from a new PSF, which might be impractical if too much data is required for training. Again, the flexibility and robustness of this method is highlighted, so it would be nice to see some experiments or discussion addressing this, e.g. comparison against other blind deconvolution methods on datasets with different PSFs.
>
> The BioSR dataset for super-resolution microscopy is actually a good benchmark in this regard. It contains images acquired under varied conditions (i.e., different hardware, wavelength, etc.), and each variation corresponds to a different point spread function (PSF). The aim of the dataset is to help in training models that can be generalized to any fluorescence microscopy system. Therefore, since the model was trained on a wide array of PSFs, certain generalization capabilities are expected. In other words, we would expect the model to perform well in the general domain of fluorescence microscopy. The results obtained in our experiments show that the model was able to learn these different PSFs simultaneously. However, if the problem settings change too much with respect to the training dataset, there is in fact a need for retraining. Even though this is not something desirable, the lack of hyperparameter fine-tuning means that the method can be readily trained on new data, which shows the flexibility of our schedule-learning approach.

---

> > ### Author Response · Authors · 2023-11-17
> > **We continue our response to your concerns and questions.**
> >
> > ## Weaknesses
> >
> > ### Major concerns (continued)
> >
> > > **Estimating pixelwise schedule:** The schedules demonstrated for structured versus background regions appear to be linear with different slopes for the different regions. Would other/baseline methods perform better using an estimated linear pixelwise schedule with a higher slope on "structured" regions?
> >
> > Thanks for mentioning this idea. It is indeed an interesting concept. That type of implementation, if done carefully, would probably improve the performance of a method like CDDPM. However, there would still be some limitations. First, we find cases where the schedule is nonlinear, which suggests that linear schedules may not be a good model for all problems or all instances. Second, and more importantly, there would be a need for fine-tuning the slopes of the linear schedules and the exact boundary of the "structured" regions. Since our model is capable of learning different schedules by itself, we think it is better to avoid that type of work and adopt a learned-schedule approach.
> >
> > ### Minor concerns
> >
> > > **Fourier features:** The original VDM paper showed that a significant amount of the performance was due to the use of Fourier features. It would be helpful to know whether anything like that is the case here.
> >
> > Thanks for pointing this out. Conceivably, the use of Fourier features could help with performance. However, one of the aims of the paper is to simplify the training process of conditioned diffusion models as much as possible, so we deliberately omit this type of preprocessing. In that sense, we are more interested in developing a robust, flexible, and easy-to-use framework to learn the schedule in conditioned problems than obtaining the best possible results for any specific task. In this light, getting comparable-to-better results to the state of the art in several tasks, with a general method that needs no fine-tuning, is strong evidence for our hypothesis that setting the schedule as a hyperparameter is an unnecessary effort.
> >
> > > **Differences from VDM:** It would be helpful to have an explicit list or section summarizing the differences/extensions from VDM.
> >
> > Thanks for the advice. We have addressed this suggestion in several ways. In Section 2 (Related Work), we have included a summary of Variational Diffusion Models (VDMs, Kingma et al. (2023)) to provide more background before starting the Methods section. Also, in Section 2, we have explained some of the differences between VDMs and our method, and we have included a bullet list in Section 1 (Introduction) to state the specific contributions of our work more clearly. Moreover, throughout the paper, we have extended the discussions about our technical contributions and their impact on the performance of the method. See especially Sections 3.1, 3.2, 3.4 and 4.3.
> >
> > > **Figure design:** Many of the figures are very small and laid out in a way that makes comparing images and reading text difficult, e.g. in Figure 3 it is difficult to see the sections from which the schedules are being estimated.
> >
> > Regrettably, due to the space constraints inherent to the publication format, we were unable to present the figures in a larger size and still retain all the textual exposition we found important. We acknowledge that larger figures could potentially provide a more detailed and clearer representation of our results. However, we have to adhere to the space limitations imposed by ICLR. We have made every effort to ensure that the figures remain legible and informative at their current size by using vectorial figures, and we have enlarged the ones in the Appendix.
> >
> > ## Questions
> >
> > > *In addition to the previous concerns, I would appreciate if the authors could clarify the following:*
> >
> > 1. Does the pattern of schedules for structured versus background regions shown in Figure 2 hold true across the whole sample and for all samples?
> >
> >  The pattern does hold in general. For instance, for the super-resolution microscopy problem, we observe higher values of $\beta$ for structure than background pixels. However, the exact shapes and slopes can vary between pixels according to the difficulty of the local inverse problem.
> >
> > 2. Are there significant differences in the inference time of this model versus the methods being compared against?
> >
> >  Inference time is the same for our method and CDDPM.  We have clarified this in Sections 4.1 and 4.2, adding a corresponding text. In that sense, this method could probably benefit from consistency models to reduce the number of function evaluations. DFCAN is faster as it is not a diffusion model but rather a regression model.
> >
> > 3. Is the "US" in "US-TIE" for universal solution? Only TIE is specified in the text.
> >
> >  Thanks for noticing this issue. We have clarified in the text (Section 4.2) that US-TIE corresponds to the method described in Zhang et al. (2020) for solving the transport intensity equation. "US" does stand for "universal solution" in that paper.

---

> > > ### Author Response · Authors · 2023-11-17
> > > **References.**
> > >
> > > ### References
> > >
> > > Kingma, Diederik, Tim Salimans, Ben Poole, and Jonathan Ho. "Variational diffusion models." Advances in neural information processing systems 34 (2021): 21696-21707.
> > >
> > > Zhang, Jialin, Qian Chen, Jiasong Sun, Long Tian, and Chao Zuo. "On a universal solution to the transport-of-intensity equation." Optics Letters 45, no. 13 (2020): 3649-3652.
> > >
> > > Deng, Jia, Wei Dong, Richard Socher, Li-Jia Li, Kai Li, and Li Fei-Fei. "Imagenet: A large-scale hierarchical image database." In 2009 IEEE conference on computer vision and pattern recognition, pp. 248-255. Ieee, 2009.
> > >
> > > Saharia, Chitwan, Jonathan Ho, William Chan, Tim Salimans, David J. Fleet, and Mohammad Norouzi. "Image super-resolution via iterative refinement." IEEE Transactions on Pattern Analysis and Machine Intelligence 45, no. 4 (2022): 4713-4726.
> > >
> > > Kawar, Bahjat, Michael Elad, Stefano Ermon, and Jiaming Song. "Denoising diffusion restoration models." Advances in Neural Information Processing Systems 35 (2022): 23593-23606.

---

> > > > ### Comment · Reviewer_tcMU · 2023-11-18
> > > > **Follow-up question about uncertainty computation**
> > > >
> > > > I would like to thank the authors for their detailed response. Most of my concerns have been well-addressed. I have one follow-up question about the uncertainty computation:
> > > >
> > > > It seems like the uncertainty is evaluated by the sample variance. Do you compute the sample variance over e.g. 5 reconstruction examples, as in Figure 17, or you can read that from the beta schedule? Also there is a discussion on how the uncertainty is linked to $\sqrt{\beta (t,x)}$. I wonder that $t$ do you use in this case, is it $t=0$? This question also applies to the beta-schedule panel in Figure17.

---

> > > > > ### Author Response · Authors · 2023-11-18
> > > > > **Answer to follow-up question about uncertainty computation**
> > > > >
> > > > > **Answer to follow-up question.**
> > > > >
> > > > > Thanks for the question. We are happy to clarify the matter of uncertainty estimation. As you indicate, we measured uncertainty using the sample variance throughout the paper. In any case, we do not claim this is the only or necessarily the best way to do it. We use it because it allows for clear visual presentation and emphasizes that uncertainty directly impacts the reconstructed image (i.e., different reconstructions can vary in some of the details). Regarding your specific questions:
> > > > >
> > > > > > *Do you compute the sample variance over e.g. 5 reconstruction examples, as in Figure 17, or you can read that from the beta schedule?*
> > > > >
> > > > > The former. That is, all the graphs depicting sample variance are constructed by calculating the pixel-wise variance over a certain number of reconstructions. For instance, for Figure 17, the sample variance is calculated over the 5 reconstructions that we include in the Figure. In other cases, where the reconstructions are not shown in the figure, we have used more samples (e.g., 20 reconstructions for Figure 12) to have a more robust estimator.
> > > > >
> > > > > > *Also there is a discussion on how the uncertainty is linked to
> > > > > $\sqrt{\beta(t,x)}$. I wonder that $t$ do you use in this case, is it $t=0$? This question also applies to the beta-schedule panel in Figure17.*
> > > > >
> > > > > Thanks for the question. It is mentioned briefly in Appendix M, but we can certainly provide more detail about this. Conceptually, the value used is the integral of $\beta$ with respect to $t$, that is,
> > > > >
> > > > > $$I=\int_0^1 \beta(t,x)dt.$$
> > > > >
> > > > > In that sense, $I$ is the integral with respect to $t$ of the square of the diffusion term $\sqrt{\beta(t,x)}$. In terms of actual implementation, this is estimated as
> > > > >
> > > > > $$S=\frac{1}{T}\sum_{i=1}^T \beta(t_i,x).$$

---

> > > > > > ### Comment · Reviewer_UQRa · 2023-11-22
> > > > > > **Response to author comments**
> > > > > >
> > > > > > Thank you for the responses, I think you've addressed all my major concerns. I've updated my score to reflect that.

---

### Official Review · Reviewer_tcMU · 2023-10-31

**Soundness:** 2 fair
**Presentation:** 2 fair
**Contribution:** 2 fair
**Rating:** 5
**Confidence:** 3

**Summary:**

This paper presents a conditional extension to the prior work of Variational Diffusion Models, to allow learning the variance schedule which eliminates the need to fine-tune this choice. This goal is achieved by incorporating the conditioning information as an additional input to the diffusion denoising network and as the input to one of the component in the decomposition of the learnable variance schedule. Specifically,  the paper discusses a regularized learning approach that keeps the scale of the SNR curvatures with respect to time steps to be low. Empirically, the method shows promising results on super-resolution microscopy tasks and quantitative phase imaging tasks.

**Strengths:**

This paper's primary strength lies in its practical extension of variational diffusion models (VDM) to the conditioning case, including several technical improvements such as the incorporation of a regularization term on the signal-to-noise ratio.

In addition, by adopting the VDM framework, the paper eliminates the need of prior works that fine tunes the variance schedule.

Another strength of the paper is on the experiment study which includes two practical downstream benchmarks assessed with meaningful metrics. It also pays attention on the uncertainty quantification, which is rarely highlighted in prior works.

**Weaknesses:**

The biggest weakness is the paper's technical novelty. The proposed approach seems like a straightforward conditional extension to the of variational diffusion models. In addition, the paradigm of turning an unconditional model to a conditional version has been largely explored and established, e.g.  [1]. I can see the decomposition of the learnable variance schedule, and the regularized learning are novel. In Besides that, can the authors comment on the technical non-triviality of this extension?

Another weakness is lack of ablation study. For example, how does the method perform without using the regularized learning approach in Section 3.4?

Finally, there is a room for improvement in terms of clarity, especially in the experiment section. See my detailed comments in the below Questions section.


[1] Chitwan Saharia, Jonathan Ho, William Chan, Tim Salimans, David J. Fleet, and Mohammad Norouzi. Image Super-Resolution via Iterative Refinement, June 2021. URL http://arxiv.org/abs/ 2104.07636. arXiv:2104.07636 [cs, eess].

**Questions:**

1. In abstract and intro, what does "causal factors" mean?

2. The condition $A(x) = y$ is brought up in the introducation, but is not used in the method section. Does this condition specification matter?

3. Missing a background of Variational Diffusion Models (VDM) before starting the method section. And I would appreciate a more straightforward discussion to how this version extends beyond and distinguishes from VDM.

4. I find the notation $SNR''$ on page 5 not clear enough; it might be worth out writing $SNR''(t,x)$ when it is first introduced.

5. Missing (the reference to) the details on how the base model architecture and other training details for the competing methods. This question is important to understand whether the empirical compairison is a fair one.

6. Missing a description on competing methods, i.e. DFCAN, CDDPM, and  a discussion on how is the proposed method different from them.

7. For the CDDPM, did you fine-tune the hyperparameters, namely the variance schedule?

8. Limited methods used for comparison. There are definitely other qualified competing methods in the diffusion space, e.g. [2], [3], [4]

9. Table1: Maybe I missed this point but what does the "resolution" metric mean?

10. Table 1: why there is only one underlined result?

11. Table 1: Why some rows do not have underlines and bolded results?

12. Table 1: How do you define statistically significant? Do you repeat the experiments for multiple times and compute the average results with standard errors? If so, please describe this detail in table or the text.

13. Table 2a: I would suggest to add the "synthetic" description to the HCOCO dataset in the caption.

14. Section 4.2.2, what is the "US-TIE method"?

15. Figure 3a: maybe I missed this but what do the structure and background in the legend box mean? Are they fixed values instead of learned?

16. Figure 3a: Where does the learned uncertainty reflect in the figure, or it is not? I was thinking of a figure like Figure 4a in [1]. And how do you draw the conclusion on "We briefly point out that our uncertainty estimation is consistent with the values of β as described above" if there is no comparison between the uncertainty estimation and the values of $\beta$?

17. I wonder how the uncertainty quantification around the reconstruction can be useful?

18. Appendix A, second paragraph, first sentence, should be "Ay=x" instead of "Ax=y"?

[1] Chitwan Saharia, Jonathan Ho, William Chan, Tim Salimans, David J. Fleet, and Mohammad Norouzi. Image Super-Resolution via Iterative Refinement, June 2021. URL http://arxiv.org/abs/ 2104.07636. arXiv:2104.07636 [cs, eess].
[2] Denoising Diffusion Restoration Models. Bahjat Kawar, Michael Elad, Stefano Ermon, Jiaming Song. https://arxiv.org/abs/2201.11793
[3] Diffusion Models Beat GANs on Image Synthesis. Prafulla Dhariwal, Alex Nichol. https://arxiv.org/abs/2105.05233
[4] Pseudoinverse-Guided Diffusion Models for Inverse Problems. Jiaming Song, Arash Vahdat, Morteza Mardani, Jan Kautz. https://openreview.net/forum?id=9_gsMA8MRKQ

---

> ### Author Response · Authors · 2023-11-16
> **Thanks for your in-depth review. We have addressed your concerns in our response and in the text of the paper.**
>
> # Official Comment for Reviewer tcMU
>
> Thank you for taking the time to review our work with such great detail. We appreciate your comments about the paper's Strengths, which we think are a good showcase of our main objectives. We also very much appreciate your detailed criticism, which have helped us improve our paper. Please find below a detailed clarification of your comments and questions. We have also included this material in the new version of the paper, highlighting the new text in blue to make it easier to find.
>
>
> ## Weaknesses
>
> > *The biggest weakness is the paper's technical novelty. The proposed approach seems like a straightforward conditional extension to the of variational diffusion models. In addition, the paradigm of turning an unconditional model to a conditional version has been largely explored and established, e.g. [1]. I can see the decomposition of the learnable variance schedule, and the regularized learning are novel. In Besides that, can the authors comment on the technical non-triviality of this extension?*
>
> We can comment about the technical non-triviality of our contributions in more detail. In general, our hypothesis is that the schedule can be learned, eliminating the need for fine-tuning the schedule while exhibiting better results. Kingma et al. (2023) explored this idea in [6], but only for the unconditioned case. We are interested in the case of conditioned sampling, which opens up many more opportunities for application, but a straightforward application of Saharia et al. (2021) does not work at all in this case (the schedule does not converge to anything useful or meaningful).
>
> So, in fact, learning the schedule for conditioned sampling is a non-trivial challenge and is far from a direct combination of [1] and [6]. As you mention, two of the key ideas that address this problem are the decomposition/factorization of the variance schedule (Section 3.2) and the regularization strategy (Section 3.4), both of which are essential:
>
> - It is non-trivial how to properly condition the schedule on the data $\mathbf{x}$, because the schedule function needs to be monotonic in $t$, but not necessarily on $\mathbf{x}$. Our decomposition/factorization of the function $\beta$ is in our opinion a clever way of solving this problem and is absolutely key for our method. See comments in Sections 3.2 and 4.3.
> - We prove that the rate of convergence of the discrete-time diffusion loss to the continuous-time case depends strongly on the schedule functions. Kingma et al. (2023) show that diffusion models are invariant under schedule selection in the continuous case. Nevertheless, any discretization used for model sampling will introduce the dependency shown in Appendix E.2 on the schedule's derivatives. Our finding motivates the introduction of a novel regularization term that proves to be critical for the performance of our method (details in Section 3.4 and 4.3, and Appendix K).
>
> Regarding implementation details, we replace the architecture in Kingma et al. (2023) with two networks, one required to be positive (for the conditioning variable) and one monotonic-convolutional network (for the time variable). This allows us to test our model with inputs of different resolutions without retraining. Moreover, by incorporating schedule properties in the loss function, our method does not need the post-processing of the schedule that Kingma et al. 2023 uses, nor the preprocessing of the input (addition of Fourier features). These design decisions are arguably less novel, but they result in a cleaner and more straightforward method compared to Kingma et al. (2023).

---

> ### Author Response · Authors · 2023-11-16
> **We continue our response to your concerns and questions.**
>
> ## Weaknesses (continued)
>
> > *Another weakness is lack of ablation study. For example, how does the method perform without using the regularized learning approach in Section 3.4?*
>
> Thanks for this suggestion. We agree that an ablation study would help clarify our technical contributions' impact on the performance. We have answered something similar for a different reviewer, but we also respond here in order to keep the reviews and specific comments organized.
>
> We have clarified the role played by our specific contributions in the new version of the paper (see Sections 3.2, 3.4 and 4.3). We have also performed an ablation study. Some important points:
>
> 1. The regularization term described in Section 3.4, which you mention as an example, is critical to keep the schedule from converging to an uninformative solution that trivially minimizes the diffusion loss. In this regard, this technical novelty is essential for the method. See details in Section 3.4. We have also ran experiments without using this regularization term, with results described in Section 4.3.
> 2. Our factorization of the function $\beta$ is also crucial for our method. Recall that the schedule function $\gamma$ needs to be monotonic in $t$, but not necessarily on $\mathbf{x}$, so it is non-trivial how to properly condition the schedule on the data $\mathbf{x}$. See details in Section 3.2. While trying these changes out, the method simply did not converge.
> 3. We performed an ablation study on the effect of having a global unique schedule instead of a pixel-wise dedicated schedule. In our ablation study (Appendix K), we compare global vs. pixel-wise schedules (CVDM-simple). The latter outperforms CDDPM but does not match CVDM's results, highlighting the value of pixel-wise schedules in addressing unique image challenges. We also include an ablation study on the impact of our regularization strategy, which shows the importance of regularized learning for CVDM.
>
> >*Finally, there is a room for improvement in terms of clarity, especially in the experiment section. See my detailed comments in the below Questions section.*
>
> Thanks a lot for going into so much detail regarding the experimental section (and the rest of the paper). We include an answer to all your questions below.
>
>
>
> ## Questions
>
> 1. In abstract and intro, what does "causal factors" mean?
>
>     Thanks for pointing this out. We have changed "causal factors" to "parameters" in the new version of the paper so as not to generate confusion with e.g. causal inference. In general, it means we lack some key information (which we want to recover) about the process that generated the data.
>
> 2. The condition $A(x)=y$ is brought up in the introduction but is not used in the method section. Does this condition specification matter?
>
>     This condition specification does not matter in Methods section (our method is general and agnostic about the specific mapping $A$). It is only mentioned in the Introduction to further clarify what is meant by an Inverse Problem, and to introduce the relation between $\mathbf{x}$ and $\mathbf{y}$, which is used throughout the paper.
>
> 3. Missing a background of Variational Diffusion Models (VDM) before starting the method section. And I would appreciate a more straightforward discussion to how this version extends beyond and distinguishes from VDM.
>
>    Thanks for the suggestion. We have answered similar concerns for other reviewers, too. We also include an answer here. We have addressed your suggestion in several ways. In Related Work (Section 2), we have included a summary of Variational Diffusion Models (VDMs, [6]), to provide more background before starting the Methods section, as you suggest. In Section 2, we have also explained some of the differences with our method, and we have included a bullet list in the Introduction (Section 1) to explain the contributions of our work more clearly. Moreover, throughout the paper, we have extended the discussions about our technical contributions and their importance for the success of the method. See especially Sections 3.1, 3.2, 3.4 and 4.3. We also include here a relevant summary.
> VDMs (Kingma et al. 2023) constitute the main framework described in the literature for learning the variance schedule. VDMs formulate a diffusion process for unconditioned distribution sampling, where the latent variables are indexed in continuous time, and the schedule is a learnable function that must satisfy a few minimal conditions. The parameters of the schedule are defined as a monotonic network, and the model is trained by minimizing a weighted version of the noise prediction loss. Their work also uses Fourier features to improve the prediction of high-frequency details. We introduce the Conditional Variational Diffusion Model (CVDM), which includes important generalizations and novelties with respect to VDMs, as described in detail above (see our answers to Weaknesses).

---

> ### Author Response · Authors · 2023-11-16
> **We continue our response to your concerns and questions.**
>
> ## Questions (continued)
>
> 4. I find the notation $SNR''$ on page 5 not clear enough; it might be worth out writing $SNR''(t,x)$ when it is first introduced.
>
>     We have followed the suggestion of using SNR$''(t,\mathbf{x})$ when introducing the term in Section 3.4. Also, we have slightly rewritten that paragraph to further clarify the notation.
>
> 6. Missing (the reference to) the details on how the base model architecture and other training details for the competing methods. This question is important to understand whether the empirical compairison is a fair one.
>
>     Thanks for this insight. The architecture details for our method are provided in Appendix G. Since one of our aims is to prove that learning a schedule can improve the results over fine-tuning it, we base both our architecture and that of CDDPM on [1], and reduce the number of parameters to avoid overfitting. To fine-tune CDDPM, we choose a linear schedule as in [1] and perform a grid search to find the best schedule. In that sense, both diffusion models share the same base architecture, with the addition of the schedule network for our method. For non-diffusion models, DFCAN is implemented as in [5] (referenced in Section 4.1) and US-TIE is implemented as in [4] (referenced in Section 4.2).
>
> 7. Missing a description on competing methods, i.e. DFCAN, CDDPM, and a discussion on how is the proposed method different from them.
>
>     Thanks for pointing this out. We have added a brief description of these methods in Section 4. Specifically, DFCAN is a regression-based method that introduces the Fourier attention channel mechanism, currently state of the art in super-resolution microscopy. We implemented DFCAN as in [5], and CDDPM is the conditioned DDPM implemented and trained as described in [1].
>
> 8. For the CDDPM, did you fine-tune the hyperparameters, namely the variance schedule?
>
>     Yes, we fine-tuned them. We changed our text to emphasize better that the CDDPM schedule was fine-tuned (see Sections 4.1 and 4.2).
>
> 9. Limited methods used for comparison. There are definitely other qualified competing methods in the diffusion space, e.g. [2], [3], [4].
>
>     Both DFCAN and US-TIE were chosen for being the state of the art for super-resolution microscopy and QPI, respectively. Regarding diffusion models, there are certainly a number of methods available. The main reason we chose CDDPM is that we wanted to test the effect of learning the schedule, as opposed to fine-tuning it, while controlling for many of the implementation details of the methods. In that sense, CDDPM is sufficiently similar to CVDM to allow us to assess better the impact of learning the schedule.
>
>     To a considerable extent, we have verified our hypothesis that treating the schedule as a hyperparameter is unnecessary because it takes effort and may lead to suboptimal performance. This is of special importance to the users of the method since reducing the number of hyperparameters provides a way of applying this problem to a larger array of problems without the overhead of fine-tuning for the specific case. We find that learning the schedule in this way is straightforward and leads to comparable-to-better performance. Aditionally, we also wanted a way to quantify uncertainty for these methods.
>
>     To provide a more comprehensive understanding of our method's performance, we have conducted further experiments for an image super-resolution task using ImageNet. The results are described in Section 4.3 and Appendix L. For this task, we compare against [1] and one of the methods referenced in your question (Denoising Diffusion Restoration Models, or DDRMs). We get comparable results to [1] and slighlty improve on DDRMs. In any case, we want to clarify that we are not claiming a superiority in results for all tasks and compared to all other diffusion-based methods. To our mind, one of the great strengths of CVDM is that it provides a method for learning the schedule that is robust, provides a way to quantify the uncertainty of the reconstruction, requires no fine-tuning, yields good results (comparable or superior to state-of-the-art methods for the problems we tried), and which can be used to enhance other architectures.

---

> > ### Author Response · Authors · 2023-11-16
> > **We continue our response to your concerns and questions.**
> >
> > ## Questions (continued)
> >
> > 9. Table 1: Maybe I missed this point but what does the "resolution" metric mean?
> >
> > Spatial resolution, also known as lateral resolution, describes the ability to distinguish fine details in the X and Y plane of an image. It determines how closely spaced two distinct points or objects can be positioned before they appear as a single merged entity in the image.
> > Spatial resolution is related to the cutoff frequency in the k-space (Fourier space). In super-resolution, the aim of the problem is to restore the frequencies that were lost due to the low-pass effect of the optical system. Therefore, the resolution metric also measures the extent of the k-space. Blurry images will have a lower resolution metric, a larger distance is required to tell objects apart because their Fourier transform will lack high frequencies (the cutoff frequency will be at a lower frequency), and higher resolution means that the image has more information in the higher frequencies. Thus, it will require a shorter distance to tell objects apart.
> >
> > 10. Table 1: why there is only one underlined result?
> >
> > Thanks for noticing this, it was an error on our part. We had missed a few results that needed to be underlined, we have now fixed Table 1.
> >
> > 11. Table 1: Why some rows do not have underlines and bolded results?
> >
> > For those rows, there are no statistically significant differences between methods.
> >
> > 12. Table 1: How do you define statistically significant? Do you repeat the experiments for multiple times and compute the average results with standard errors? If so, please describe this detail in table or the text.
> >
> > The idea is the following. Take a testing dataset with N samples. For each method, we calculate the mean error over the dataset using 1 reconstruction for each input. Then, we compare the sample error means between two different methods, by performing a hypothesis test to determine if the two population means are statistically significant. We have included a clarification in the caption of Table 1 to explain this.
> >
> > 13. Table 2a: I would suggest to add the "synthetic" description to the HCOCO dataset in the caption.
> >
> > We have followed your suggestion and added the "synthetic" description for the HCOCO dataset.
> >
> > 14. Section 4.2.2, what is the "US-TIE method"?
> >
> > Thanks for this comment. We have clarified in the text (Section 4.2) that US-TIE corresponds to the method described in [4] for solving the transport intensity equation.
> >
> > 15. Figure 3a: maybe I missed this but what do the structure and background in the legend box mean? Are they fixed values instead of learned?
> >
> > It is nomenclature we use to distinguish between classes of pixels. In general, "structure" corresponds to regions of the image with high-frequency information (e.g., edges and fine details), and background corresponds to regions with low-frequency information (e.g., large-scale patterns or gradients). We have included this explanation in Section 5. The BioSR dataset is particularly interesting in this classification, since the widefield microscope captures the low frequencies and the high frequencies remain to be reconstructed.
> >
> > 16. Figure 3a: Where does the learned uncertainty reflect in the figure, or it is not? I was thinking of a figure like Figure 4a in [1]. And how do you draw the conclusion on "We briefly point out that our uncertainty estimation is consistent with the values of $\beta$ as described above" if there is no comparison between the uncertainty estimation and the values of $\beta$?
> >
> > Thanks for pointing this out. We have added an illustrative Figure in Appendix M. The Figure shows several reconstructions for the same input, and it illustrates how different reconstructions can vary in the regions with higher sample variance. It also shows the relationship between sample variance and $\beta$, and how regions of higher sample variance generally exhibit higher reconstruction error.
> >
> > The uncertainty (i.e., higher variance) in the reconstruction is theoretically tied to $\beta$. As described in [3], the reverse diffusion process can be characterized by a stochastic differential equation with a diffusion coefficient of $\sqrt{\beta(t,\mathbf{x})}$ (in the variance-preserving case). Hence, higher values of $\beta$ introduce more diffusion into the reverse process over time, leading to more varied reconstructions. We have added this explanation at the end of Section 5, and in the new Figure 16 (Appendix M) it is also possible to appreciate that $\beta$ is correlated to the sample variance and the reconstruction error.

---

> > > ### Author Response · Authors · 2023-11-16
> > > **We continue our response to your concerns and questions.**
> > >
> > > ## Questions (continued)
> > >
> > > 17. I wonder how the uncertainty quantification around the reconstruction can be useful?
> > >
> > > In applications such as microscopy, relevant information may sometimes be contained in single pixels (for instance, cell interaction). Therefore, a single hallucination in the reconstruction method can lead to totally misleading conclusions. The same is the case for MRI, where a few pixels can determine the difference between the presence or absence of a tumor. In fact, this is one of the challenges for the adoption of accelerated MRI in clinical environments, for instance, in [2]. That's why an estimation of certainty for different parts of the reconstruction can be a valuable tool.
> > >
> > > 18. Appendix A, second paragraph, first sentence, should be "Ay=x" instead of "Ax=y"?
> > >
> > > Thanks for pointing out this error. We have fixed it in the text.
> > >
> > >
> > > ### References
> > >
> > > [1] Chitwan Saharia, Jonathan Ho, William Chan, Tim Salimans, David J. Fleet, and Mohammad Norouzi. Image Super-Resolution via Iterative Refinement, June 2021. URL http://arxiv.org/abs/ 2104.07636. arXiv:2104.07636 [cs, eess].
> > >
> > > [2] Bhadra S, Kelkar VA, Brooks FJ, Anastasio MA. On Hallucinations in Tomographic Image Reconstruction. IEEE Trans Med Imaging. 2021 Nov;40(11):3249-3260. doi: 10.1109/TMI.2021.3077857. Epub 2021 Oct 27. PMID: 33950837; PMCID: PMC8673588.
> > >
> > > [3] Song, Y., Sohl-Dickstein, J., Kingma, D.P., Kumar, A., Ermon, S. and Poole, B., 2020. Score-based generative modeling through stochastic differential equations. arXiv preprint arXiv:2011.13456.
> > >
> > > [4] Zhang, Jialin, Qian Chen, Jiasong Sun, Long Tian, and Chao Zuo. "On a universal solution to the transport-of-intensity equation." Optics Letters 45, no. 13 (2020): 3649-3652.
> > >
> > > [5] Qiao, Chang, Di Li, Yuting Guo, Chong Liu, Tao Jiang, Qionghai Dai, and Dong Li. "Evaluation and development of deep neural networks for image super-resolution in optical microscopy." Nature Methods 18, no. 2 (2021): 194-202.
> > >
> > > [6] Kingma, Diederik, Tim Salimans, Ben Poole, and Jonathan Ho. "Variational diffusion models." Advances in neural information processing systems 34 (2021): 21696-21707.

---

> > > > ### Comment · Reviewer_tcMU · 2023-11-19
> > > > **follow-up question regarding uncertainty quantification**
> > > >
> > > > (I meant to ask about this question in my thread but ended up in another reviewer's thread) Thank you for clarifying my questions on uncertainty quantification. I have one more question: have you compared the uncertainty quantification results of the proposed method to other methods in the paper? I was looking for evidence of the proposed approach can have better uncertainty quantification.

---

> > > > > ### Author Response · Authors · 2023-11-19
> > > > > **Answer to follow-up question regarding uncertainty quantification**
> > > > >
> > > > > We have not compared our uncertainty quantification results with other methods. An important reason for this is that, to the best of our knowledge, uncertainty quantification for diffusion models remains largely unexplored. In works such as Ho et al. (2020), the evaluation is more focused on the sample quality or realism, as evidenced by e.g. the FID score. Even for diffusion models applied to inverse problems (e.g. Saharia et al. 2021, Kawar et al. 2022), uncertainty quantification for the reconstructions does not seem to be addressed. Since uncertainty estimation is not the primary aspect of our paper, in experiments we have devoted more work to evaluating the reconstruction error of CVDM, and comparing it with other methods.
> > > > >
> > > > > That said, we agree that uncertainty quantification is an important topic, and we can offer a few insights regarding your question:
> > > > >
> > > > > - First, there are other measures of uncertainty quantification that could be used. For example, the size of the 95% confidence interval for each pixel. There is also a more sophisticated notion introduced this year in Belhasin et al. (2023). As we mentioned in our previous answer, we use the sample variance as it has a simple visualization and interpretation, but there are certainly other ways to do it.
> > > > > - Second, since the variance of the reconstruction is related to the schedule, having a pixel-wise schedule, as in our method, should offer more precise uncertainty quantification than using a single schedule. Our hypothesis is that, during training, CVDM tries to minimize the variance of regions that are easier to reconstruct, and in that sense, a pixel-wise schedule offers more fine-grained control.
> > > > >
> > > > > As we mentioned, to the best of our knowledge this topic has not received too much attention, so the exact relation between variance, reconstruction error, and the schedule is an interesting area for future research. This revision stage for our paper has certainly offered some very intereting questions that we plan to explore.
> > > > >
> > > > >
> > > > > ### References
> > > > >
> > > > > Ho, Jonathan, Ajay Jain, and Pieter Abbeel. "Denoising diffusion probabilistic models." Advances in neural information processing systems 33 (2020): 6840-6851.
> > > > >
> > > > > Saharia, Chitwan, Jonathan Ho, William Chan, Tim Salimans, David J. Fleet, and Mohammad Norouzi. "Image super-resolution via iterative refinement." IEEE Transactions on Pattern Analysis and Machine Intelligence 45, no. 4 (2022): 4713-4726.
> > > > >
> > > > > Kawar, Bahjat, Michael Elad, Stefano Ermon, and Jiaming Song. "Denoising diffusion restoration models." Advances in Neural Information Processing Systems 35 (2022): 23593-23606.
> > > > >
> > > > > Belhasin, Omer, Yaniv Romano, Daniel Freedman, Ehud Rivlin, and Michael Elad. "Volume-Oriented Uncertainty for Inverse Problems." In NeurIPS 2023 Workshop on Deep Learning and Inverse Problems. 2023.

---

> > > > > > ### Author Response · Authors · 2023-11-23
> > > > > >
> > > > > > We thank you for your questions and carefully addressed comments.  We were wondering if you had additional questions that we could answer for you?

---

### Official Review · Reviewer_qRoi · 2023-11-05

**Soundness:** 3 good
**Presentation:** 4 excellent
**Contribution:** 2 fair
**Rating:** 3
**Confidence:** 4

**Summary:**

In this paper the authors propose to learn the schedule of the forward process for conditional diffusion models, building up on the work of   [1]. This is an interesting topic since fine-tuning the schedule can be quite time consuming. The main contribution of this paper is to make the schedule dependent on the conditioning variable and to devise an appropriate learning procedure.


[1] Kingma, Diederik, et al. "Variational diffusion models." Advances in neural information processing systems 34 (2021): 21696-21707.

**Strengths:**

The paper is well written and quite easy to understand, which is great. The numerical experiments/applications are interesting and show that the proposed method provides comparable or superior performance to existing methods that fine-tune the schedule.

**Weaknesses:**

I think that the contribution of this paper is quite marginal and is not suited for ICLR. The main difference with the methodology developed in [1] is the replacement of the unstable SNR term. While this yields good results in practice, I am not sure if this is enough novelty for a conference like ICLR.

**Questions:**

I have no further questions.

---

> ### Author Response · Authors · 2023-11-20
>
> # Official Comment for Reviewer qRoi
>
> Thanks for taking the time to read our work. We include a response below. If you could provide more details and evidence, we could answer more specifically your concerns. References below allude to the revised version of our paper, in which we have clarified some points, included additional experiments, and stated our contributions more explicitly.
>
> ## Weaknesses
>
> > *I think that the contribution of this paper is quite marginal and is not suited for ICLR.*
>
> We have to respectfully disagree with your evaluation of our work, at least as it is stated right now. We see no reason why our work would not be suited for ICLR. On the contrary, the topics discussed in our paper are active areas of research in machine learning. See for instance these ICLR 2023 papers:
>
> - Deep learning applied to inverse problems [2], [3], [4].
> - Diffusion models [5], [6], [7].
> - Variational inference [8], [9].
> - Uncertainty quantification [10], [11].
>
> In our paper, we have used novel technical ideas to tackle a relevant problem in current machine learning research. We have derived analytical insights to solve this problem (see Section 3 and Appendix E, for instance), and we have conducted thorough experimental work to provide compelling evidence for our ideas (see Section 4 and discussion in Section 5). We can answer more specifically if you provide details as to why you consider our paper not suited for ICLR.
>
> > *The main difference with the methodology developed in [1] is the replacement of the unstable SNR term. While this yields good results in practice, I am not sure if this is enough novelty for a conference like ICLR.*
>
> The differences with [1] go far beyond the replacement of the SNR term. Their framework is for unconditioned sampling only, which we extend in two ways. First, we allow for conditioned sampling, which opens up many more opportunities for application, including the case of inverse problems with which we motivate our work in Section 1. Second, we allow for learning a pixel-wise variance schedule, which further improves performance (see Ablations in Section 4.3).
>
> Moreover, these extensions are highly non-trivial. Learning the schedule for conditioned sampling is a difficult challenge and is far from a direct application of [1]. There are at least two new ideas we use to make it work: the decomposition/factorization of the variance schedule (Section 3.2) and the regularization strategy (Section 3.4), both of which are essential:
>
> - It is non-trivial how to properly condition the schedule on the data $\mathbf{x}$, because the schedule function needs to be monotonic in $t$, but not necessarily on $\mathbf{x}$. Our decomposition/factorization of the function $\beta$ is, in our opinion, a clever way of solving this problem and is absolutely key for our method. See comments in Sections 3.2 and 4.3.
> - We prove that the rate of convergence of the discrete-time diffusion loss to the continuous-time case depends strongly on the schedule functions. [1] show that diffusion models are invariant under schedule selection in the continuous case. Nevertheless, any discretization used for model sampling will introduce the dependency shown in Appendix E.2 on the schedule's derivatives. Our finding motivates the introduction of a novel regularization term that proves to be critical for the performance of our method (details in Section 3.4 and 4.3, and Appendix K).
>
> Regarding implementation details, we replace the architecture in [1] with two networks, one required to be positive (for the conditioning variable) and one monotonic-convolutional network (for the time variable). This allows us to test our model with inputs of different resolutions without retraining. Moreover, by incorporating schedule properties in the loss function, our method does not need the post-processing of the schedule that [1] uses, nor the preprocessing of the input (addition of Fourier features). These two last design decisions are arguably less novel, but they result in a cleaner and more straightforward method compared to [1].
>
> So we think it is clear that our work has substantial differences with [1], and there is a good deal of technical novelty. The "replacement of the unstable SNR term" that you mention is indeed a part of our work, but it is not the most novel or important part. The revised version of the paper makes this more clear. In Section 2 (Related Work), we have included a summary of Variational Diffusion Models (VDMs) [1] to provide more background before starting the Methods section. Also, in Section 2, we have explained some of the differences between VDMs and our method, and we have included a bullet list in Section 1 (Introduction) to state the specific contributions of our work more clearly. Moreover, throughout the paper, we have extended the discussions about our technical contributions and their impact on the performance of the method. See especially Sections 3.1, 3.2, 3.4 and 4.3.

---

> > ### Author Response · Authors · 2023-11-20
> >
> > ### References
> >
> > [1] Kingma, Diederik, et al. "Variational diffusion models." Advances in neural information processing systems 34 (2021): 21696-21707.
> >
> > [2] Song, Jiaming, et al. "Pseudoinverse-guided diffusion models for inverse problems." International Conference on Learning Representations. 2022.
> >
> > [3] Li, Dongzhuo. "Differentiable Gaussianization layers for inverse problems regularized by deep generative models." arXiv preprint arXiv:2112.03860 (2021).
> >
> > [5] Xu, Yilun, Shangyuan Tong, and Tommi Jaakkola. "Stable target field for reduced variance score estimation in diffusion models." arXiv preprint arXiv:2302.00670 (2023).
> >
> > [6] Reid, Machel, Vincent Josua Hellendoorn, and Graham Neubig. "Diffuser: Diffusion via edit-based reconstruction." The Eleventh International Conference on Learning Representations. 2022.
> >
> > [7] Vignac, Clement, et al. "Digress: Discrete denoising diffusion for graph generation." arXiv preprint arXiv:2209.14734 (2022).
> >
> > [8] Zhao, Jianan, et al. "Learning on large-scale text-attributed graphs via variational inference." arXiv preprint arXiv:2210.14709 (2022).
> >
> > [9] Malkin, Nikolay, et al. "GFlowNets and variational inference." arXiv preprint arXiv:2210.00580 (2022).
> >
> > [10] Lin, Zi, et al. "On Compositional Uncertainty Quantification for Seq2seq Graph Parsing." The Eleventh International Conference on Learning Representations. 2022.
> >
> > [11] Kuhn, Lorenz, Yarin Gal, and Sebastian Farquhar. "Semantic uncertainty: Linguistic invariances for uncertainty estimation in natural language generation." arXiv preprint arXiv:2302.09664 (2023).

---

> ### Author Response · Authors · 2023-11-23
>
> Before the rebuttal period is over. Do you have any questions we can help with?

---

### Author Response · Authors · 2023-11-22
**Summary of our work and of the main changes made during revision.**

## Summary

In our work, we propose a method for learning a diffusion model's variance schedule in conditioned sampling. This allows for the use of diffusion models without the need to fine-tune the schedule, which is time-consuming and not necessarily optimal. In that sense, we extend the idea of Variational Diffusion Models (VDMs) [1] to the conditional setting. This extension is highly non-trivial and requires several technical novelties. Our method can be used to solve inverse problems, exhibiting high accuracy and having the added benefit of providing uncertainty quantification for the solution.

In particular, our paper makes the following technical contributions:

1. Our method can learn an element-wise (e.g., pixel-wise) variance schedule. This improves performance and allows for fine-grained uncertainty estimation (e.g., at the pixel level). It is non-trivial how to properly condition the schedule on the data $\mathbf{x}$ because the schedule function needs to be monotonic in $t$, but not necessarily in $\mathbf{x}$. Our factorization of the function $\beta$ solves this problem (Sections 3.2 and 4.3).
3. [1] showed the diffusion loss is invariant under schedule choice in continuous time. Nevertheless, we prove that any time discretization (inevitable for computational implementation) introduces a strong dependence on the schedule's derivatives. Our finding motivates the introduction of a novel regularization term, which is key for performance (Sections 3.4 and 4.3, and Appendix K).
4. In terms of implementation, we replace the architecture in [1] with two networks, one required to be positive (for the conditioning variable) and one monotonic-convolutional network (for the time variable). This allows us to test our model with inputs of different resolutions without retraining. Moreover, our method does not need the post-processing that [1] uses, nor the input preprocessing.

These changes and extensions result in a clean and general method for learning the variance schedule, which can be applied to different problems in a straightforward manner. Originally, we tested our method on two distinct applications:

1. For super-resolution microscopy, we get similar results to the state-of-the-art methods in reconstruction metrics and much improved results for the resolution metric. No fine-tuning of the schedule is needed (Section 4.1).
2. For quantitative phase imaging, we significantly improve over the current state of the art in all metrics. No fine-tuning of the schedule is needed (Section 4.2).

In both cases, we can also provide uncertainty quantification for the solutions, which some of the state-of-the-art methods cannot do (e.g., some regression models).

## Revision

Based on the reviewers' feedback, we have made a number of improvements and clarifications (highlighted in blue in the paper for ease of finding). These changes do not alter the core of the paper or our main contributions. Nevertheless, they have been useful in clarifying our specific contributions, improving the writing in a number of places, and also making the experimental analysis more complete and robust. Some of the main changes:

1. To clarify our specific contributions, we have made several changes. In Section 2, we have included a summary of VDMs [1] to provide more background before starting the Methods section. In Section 2, we have also explained the differences between VDMs and our method, and we have included a bullet list in Section 1 to explain the contributions of our work more clearly. Moreover, throughout the paper, we have extended the discussions about our technical contributions and their importance for the success of the method. See especially Sections 3.1, 3.2, 3.4 and 4.3. As a result, our contributions and the novelty of our work have been more clearly laid out.
2. We have included an ablation study on the impact of our design decisions (Section 4.3 and Appendix K). First, we analyze the effect of having a global unique schedule instead of an element-wise dedicated schedule. Experimental results show that the element-wise schedule improves performance (besides allowing for more fine-grained uncertainty quantification). Second, we include an ablation study on the impact of our regularization strategy, highlighting its importance for learning a meaningful schedule.
3. To provide a more comprehensive understanding of our method's performance, we have conducted further experiments for an image super-resolution task over ImageNet. The results are described in Section 4.3 and Appendix L. These experiments provide further support for our hypothesis: with minimum fine-tuning, we can apply our method to a new problem and get comparable results to the state of the art. The method proves to be robust and easy to adapt.

### References

[1] Kingma, Diederik, et al. "Variational diffusion models." Advances in neural information processing systems 34 (2021): 21696-21707.

---

### Meta-Review · Area_Chair_X2nK · 2023-12-13

**Metareview:**

The authors propose a conditional variational diffusion model for inverse problems in computational imaging, with several practical improvements. The paper is a borderline paper, with some reviewers questioning the limited novelty of the proposed variance schedule and limited exploration of the proposed uncertainty quantification. The authors responded with a more thorough experimental evaluation and ablation studies. The method is clearly presented and evaluated, with clearly demonstrated improvements for important inverse problems in computational imaging, and a useful analysis of the posterior uncertainty is now presented.

**Justification For Why Not Higher Score:**

Concerns of limited novelty.

**Justification For Why Not Lower Score:**

Clearly described method with clear utility and good evaluation of the method. The clear utility of the method for two important inverse problems in computational imaging is a win.

---

### Decision · Program_Chairs · 2024-01-16

Accept (poster)